# Towards Understanding How Transformers Learn In-context Through a Representation Learning Lens

**Ruifeng Ren**
Gaoling School of Artificial Intelligence
Renmin University of China
Beijing, China
renruifeng920@ruc.edu.cn

**Yong Liu**[*]
Gaoling School of Artificial Intelligence
Renmin University of China
Beijing, China
liuyonggsai@ruc.edu.cn

## Abstract

Pre-trained large language models based on Transformers have demonstrated remarkable in-context learning (ICL) abilities. With just a few demonstration examples, the models can implement new tasks without any parameter updates. However, it is still an open question to understand the mechanism of ICL. In this paper, we attempt to explore the ICL process in Transformers through a lens of representation learning. Initially, leveraging kernel methods, we figure out a dual model for one softmax attention layer. The ICL inference process of the attention layer aligns with the training procedure of its dual model, generating token representation predictions that are equivalent to the dual model's test outputs. We delve into the training process of this dual model from a representation learning standpoint and further derive a generalization error bound related to the quantity of demonstration tokens. Subsequently, we extend our theoretical conclusions to more complicated scenarios, including one Transformer layer and multiple attention layers. Furthermore, drawing inspiration from existing representation learning methods especially contrastive learning, we propose potential modifications for the attention layer. Finally, experiments are designed to support our findings.

## 1 Introduction

Recently, large language models (LLMs) based on the Transformer architectures [Vaswani et al., 2017] has shown surprising in-context learning (ICL) capabilities [Brown et al., 2020, Wei et al., 2022, Dong et al., 2022, Liu et al., 2023]. By prepending several training examples before query inputs without labels, the models can make predictions for the queries and achieve excellent performance without any parameter updates. This excellent capability enables pre-trained LLMs such as GPT models to be used in general downstream tasks conveniently. Despite the good performance of the ICL capabilities, the mechanism of ICL still remains an open question.

In order to better understand the ICL capabilities, many works began to give explanations from different aspects. Xie et al. [2021] propose a Bayesian inference framework to explain how ICL occurs between pretraining and test time, where the LLMs infers a shared latent concept among the demonstration examples. Garg et al. [2022] demonstrate through experiments that pre-trained Transformer-based models can learn new functions from in-context examples, including (sparse) linear functions, two-layer neural networks, and decision trees. Zhang et al. [2023b] adopt a Bayesian perspective and show that ICL implicitly performs the Bayesian model averaging algorithm, which is approximated by the attention mechanism. Li et al. [2023] define ICL as an algorithm learning problem where a transformer model implicitly builds a hypothesis function at inference-time and derive generalization bounds for ICL. Han et al. [2023] suggest that LLMs can emulate

---

[*]Corresponding Author

kernel regression algorithms and exhibit similar behaviors during ICL. These works have provided significant insights into the interpretation of ICL capabilities from various perspectives.

In addition to the above explorations, there are also some attempts to relate ICL capabilities to gradient descent. Inspired by the dual form of linear attention proposed in Aiserman et al. [1964] and Irie et al. [2022], the ICL process is interpreted as implicit fine-tuning in the setting of linear attention by Dai et al. [2022]. However, there is still a certain noticeable gap between linear attention and the widely used softmax attention. Additionally, this comparison is more of a formal resemblance and the specific details of gradient descent, including the form of the loss function and training data, require a more fine-grained exploration. Akyürek et al. [2022] show that by constructing specific weights, Transformer layers can perform fundamental operations (mov, mul, div, aff), which can be combined to execute gradient descent. Von Oswald et al. [2023a] adopt another construction, such that the inference process on a single or multiple linear attention layers can be equivalently seen as taking one or multiple steps of gradient descent on linear regression tasks. Building upon this weight construction method, subsequent work has conducted a more in-depth exploration of the capabilities of ICL under a causal setting, noticing that the inference of such attention layers is akin to performing online gradient descent [Ding et al., 2023, Von Oswald et al., 2023b]. However, these analyses are still conducted under the assumption of linear attention and primarily focus on linear regression tasks, adopting specific constructions for the input tokens (concatenated from features and labels) and model weights. This limits the explanation of the Transformer's ICL capabilities in more general settings. Thus, the question arises: *Can we relate ICL to gradient descent under the softmax attention setting, rather than the linear attention setting, without assuming specific constructions for model weights and input tokens?*

Motivated by the aforementioned challenges and following these works that connect ICL with gradient descent, we explore the ICL inference process from a representation learning lens. First, by incorporating kernel methods, we establish a connection between the ICL inference process of one softmax attention layer and the gradient descent process of its dual model. The test prediction of the trained dual model will be equivalent to the ICL inference result. We analyze the training process of this dual model from the perspective of representation learning and compare it with existing representation learning methods. Then, we derive a generalization error bound of this process, which is related to the number of demonstration tokens. Our conclusions can be easily extended to more complex scenarios, including a single Transformer layer and multiple attention layers. Furthermore, inspired by existing representation learning methods especially contrastive learning, we propose potential modifications to the attention layer and experiments are designed to support our findings.

## 2 Preliminaries

### 2.1 In-context Learning with Transformers

The model we consider is composed of many stacked Transformer decoder layers, each of which is composed of an attention layer and a FFN layer. For simplicity, we have omitted structures such as residual connections and layer normalization, retaining only the most essential parts. We consider the standard ICL scenario, where the model's input consists of demonstrations followed by query inputs, that is, the input can be represented as $\boldsymbol{X} = [\boldsymbol{X}_D, \boldsymbol{X}_T] \in \mathbb{R}^{d_i \times (N+T)}$, where $\boldsymbol{X}_D = [\boldsymbol{x}_1, \boldsymbol{x}_2, ..., \boldsymbol{x}_N]$ denotes $N$ demonstration tokens, and $\boldsymbol{X}_T = [\boldsymbol{x}'_1, \boldsymbol{x}'_2, ..., \boldsymbol{x}'_T]$ denotes $T$ query tokens. Here, we focus more on how tokens interact during model inference while ignoring the internal structure of demonstration tokens. For the query input at position $T + 1$, its output after one layer of Transformer can be represented as

$$\boldsymbol{h}'_{T+1} = \boldsymbol{W}_V \boldsymbol{X} \operatorname{softmax}\left((\boldsymbol{W}_K \boldsymbol{X})^T \boldsymbol{W}_Q \boldsymbol{x}'_{T+1} / \sqrt{d_o}\right), \tag{1}$$

$$\widehat{\boldsymbol{x}}'_{T+1} = \boldsymbol{W}_2 \operatorname{ReLu}(\boldsymbol{W}_1 \boldsymbol{h}'_{T+1} + \boldsymbol{b}_1) + \boldsymbol{b}_2, \tag{2}$$

where $\boldsymbol{W}_K, \boldsymbol{W}_Q, \boldsymbol{W}_V \in \mathbb{R}^{d_o \times d_i}$ are parameters for key, query, value projections and $\boldsymbol{W}_1 \in \mathbb{R}^{d_h \times d_o}, \boldsymbol{W}_2 \in \mathbb{R}^{d_o \times d_h}, \boldsymbol{b}_1 \in \mathbb{R}^{d_h}, \boldsymbol{b}_2 \times \mathbb{R}^{d_o}$ are FFN parameters. Our concern is how the query token $\boldsymbol{x}'_{T+1}$ learns in-context information from demonstrations. Unlike previous work [Von Oswald et al., 2023a, Zhang et al., 2023a, Bai et al., 2023], here we do not make additional assumptions about the structure of input matrix $\boldsymbol{X}$ and parameters to study the Transformer's ability to implement some specific algorithms. Instead, we adopt the same setting as [Dai et al., 2022] to study more general cases.

## 2.2 Self-Supervised Representation Learning Using Contrastive Loss Functions

Representation learning aims to learn embeddings of data to preserve useful information for downstream tasks. One class of methods most relevant to our work is probably contrastive learning methods without negative samples [Chen and He, 2021, Grill et al., 2020, Caron et al., 2020, Tian et al., 2021]. Contrastive learning is a significant approach of self-supervised learning (SSL) which aims at learning representations by minimizing the distance between the augmentations of the same data point (positive samples) while maximizing the distance from different data points (negative samples) [He et al., 2020, Chen et al., 2020b, Oord et al., 2018, Oh Song et al., 2016]. To alleviate the burden of constructing a sufficient number of negative samples while avoiding representational collapse, some works propose architectures for contrastive learning without negative samples, which mainly use weight-sharing network known as Siamese networks [Chen and He, 2021, Grill et al., 2020, Caron et al., 2020, Tian et al., 2021]. The architecture takes two augmentations $\boldsymbol{x}_1, \boldsymbol{x}_2$ from the same data $\boldsymbol{x}$ as inputs, which will be processed by online network and target network respectively to obtain the corresponding representations, that is, $\hat{\boldsymbol{x}}_1 = f_{\text{online}}(\boldsymbol{x}_1), \hat{\boldsymbol{x}}_2 = f_{\text{target}}(\boldsymbol{x}_2)$. The two encoder networks share weights directly or using Exponential Moving Average (EMA). Then, $\hat{\boldsymbol{x}}_1$ will be input into a predictor head to obtain the predictive representation $\boldsymbol{z}_1 = g(\hat{\boldsymbol{x}}_1)$. Finally, we minimize the distance between the predictive representation and target representation, that is, $\mathcal{L}(\boldsymbol{z}_1, \text{StopGrad}(\hat{\boldsymbol{x}}_2))$ where $\text{StopGrad}(\cdot)$ means $\hat{\boldsymbol{x}}_2$ is treated as a constant during backpropagation. For $\mathcal{L}(\cdot)$, we often choose the cosine similarity or the $l_2$-norm as a measure of distance, although they are equivalent when the vector is normalized. Another class similar to our work is kernel contrastive learning [Esser et al., 2024]. Given an anchor $\boldsymbol{x}$ and its positive and negative samples $\boldsymbol{x}^+, \boldsymbol{x}^-$, it aims to optimize the loss function $\mathcal{L} = f(\boldsymbol{x})^T (f(\boldsymbol{x}^-) - f(\boldsymbol{x}^+))$, where $f(\boldsymbol{x}) = \boldsymbol{W}\phi(\boldsymbol{x})$ and $\phi(\boldsymbol{x})$ is the feature mapping for some kernel. We will consider the gradient descent process corresponding to the inference process of ICL from the perspective of representation learning and compare it with the two aforementioned representation learning patterns.

## 2.3 Gradient Descent on Linear Layer is the Dual Form of Linear Attention

It has been found that the linear attention can be connected to the linear layer optimized by gradient descent [Aiserman et al., 1964, Irie et al., 2022, Dai et al., 2022], that is, the gradient descent on linear layer can be seen as the dual form [2] of linear attention. A simple linear layer can be defined as $f_L(\boldsymbol{x}) = \boldsymbol{W}\boldsymbol{x}$, where $\boldsymbol{W} \in \mathbb{R}^{d_o \times d_i}$ is the projection matrix. Given training inputs $[\boldsymbol{x}_i]_{i=1}^N \in \mathbb{R}^{d_i}$ with their labels $[\boldsymbol{y}_i]_{i=1}^N \in \mathbb{R}^{d_o}$, a linear layer can output the predictions $[\hat{\boldsymbol{y}}_i]_{i=1}^N$ where $\hat{\boldsymbol{y}}_i = \boldsymbol{W}\boldsymbol{x}_i$ and then compute certain loss $\mathcal{L}(\hat{\boldsymbol{y}}_i, \boldsymbol{y}_i)$ for training. Backpropagation signals $[\boldsymbol{e}_i]_{i=1}^N \in \mathbb{R}^{d_o}$ will be produced to update $\boldsymbol{W}$ in gradient descent process where $\boldsymbol{e}_i = -\eta (\nabla_{\hat{\boldsymbol{y}}_i}\mathcal{L})$ if we set $\eta$ as the learning rate. During test time, the trained weight matrix $\widehat{\boldsymbol{W}}$ can be represented by its initialization $\boldsymbol{W}_0$ and the updated part $\Delta\boldsymbol{W}$, that is,

$$\widehat{\boldsymbol{W}} = \boldsymbol{W}_0 + \Delta\boldsymbol{W} = \boldsymbol{W}_0 + \sum_{i=1}^N \boldsymbol{e}_i \otimes \boldsymbol{x}_i, \tag{3}$$

where $\otimes$ denotes the outer product according to the chain rule of differentiation. On the other hand, this process can be viewed from the perspective of linear attention. Let $[\boldsymbol{k}_i]_{i=1}^N, [\boldsymbol{v}_i]_{i=1}^N \in \mathbb{R}^{d_i}$ denote the $N$ key and value vectors constituting matrices $\boldsymbol{K}, \boldsymbol{V} \in \mathbb{R}^{d_i \times N}$ respectively. For a given query input $\boldsymbol{q} \in \mathbb{R}^{d_i}$, linear attention is typically defined as the weighted sum of these value vectors

$$\text{LA}(\boldsymbol{V}, \boldsymbol{K}, \boldsymbol{q}) = \boldsymbol{V}\boldsymbol{K}^T\boldsymbol{q} = \sum_{i=1}^N \boldsymbol{v}_i\boldsymbol{k}_i^T\boldsymbol{q} = \left(\sum_{i=1}^N \boldsymbol{v}_i \otimes \boldsymbol{k}_i\right)\boldsymbol{q}.$$

Then, we can rewrite the output of a linear layer during test time as

$$f_L(\boldsymbol{x}_{test}) = \widehat{\boldsymbol{W}}\boldsymbol{x}_{test} = \boldsymbol{W}_0\boldsymbol{x}_{test} + \left(\sum_{i=1}^N \boldsymbol{e}_i \otimes \boldsymbol{x}_i\right)\boldsymbol{x}_{test} = \boldsymbol{W}_0\boldsymbol{x}_{test} + \text{LA}(\boldsymbol{E}, \boldsymbol{X}, \boldsymbol{x}_{test}),$$

$$\tag{4}$$

---

[2]It should be clarified that the term "dual" here is different from the one in mathematical optimization theory. Instead, it follows the terminology used in previous works [Irie et al., 2022, Dai et al., 2022], where the forward process of the attention layer and backward process on some model are referred to as a form of "dual".

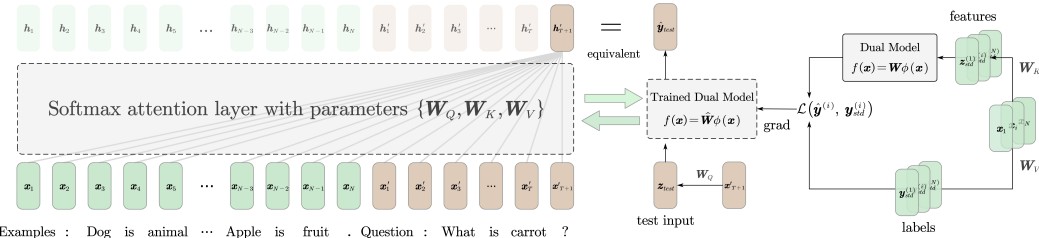

Figure 1: The ICL output $\boldsymbol{h}'_{N+1}$ of one softmax attention layer is equivalent to the test prediction $\hat{\boldsymbol{y}}_{test}$ of its trained dual model $f(\boldsymbol{x}) = \widehat{\boldsymbol{W}}\phi(\boldsymbol{x})$. The training data and test input can be obtained by linear transformations of demonstration and query tokens, respectively.

where $\boldsymbol{E} \in \mathbb{R}^{d_o \times N}$ and $\boldsymbol{X} \in \mathbb{R}^{d_i \times N}$ are stacked by backpropagation signals $[e_i]_{i=1}^N$ and training inputs $[\boldsymbol{x}_i]_{i=1}^N$ respectively. We can find from Eq (4) that the trained weight $\widehat{\boldsymbol{W}}$ records all training datapoints and the test prediction of the linear layer indicates which training datapoints are chosen to activate using $\text{LA}(\cdot)$ where $[e_i]_{i=1}^N$ can be considered as values while $[\boldsymbol{x}_i]_{i=1}^N$ as keys and $\boldsymbol{x}_{test}$ as the query. This interpretation uses gradient descent as a bridge to connect predictions of linear layers with linear attention, which can be seen as a simplified softmax attention used in Transformers.

Inspired by this relationship, Dai et al. [2022] understand ICL as implicit fine-tuning. However, this interpretation based on linear attention deviates from the softmax attention used in practical Transformers. Furthermore, this alignment is also ambiguous as the specific details of the gradient descent process, including the form of loss function and dataset, have not been explicitly addressed. In addition, Von Oswald et al. [2023a], Ding et al. [2023] also connect ICL with gradient descent for linear regression tasks using weight construction methods, where parameters $\boldsymbol{W}_K$, $\boldsymbol{W}_Q$ and $\boldsymbol{W}_V$ of the self-attention layer need to roughly adhere to a specific constructed form. However, these analyses rely on the setting of linear regression tasks and assumptions about the form of input tokens (concatenated with features and labels), which limits the interpretability of ICL capabilities from the perspective of gradient descent. Thus, we attempt to address these issues in the following sections.

## 3 Connecting ICL with Gradient Descent

In this section, we will address two questions discussed above: (i) *Without assuming specific constructions for model weights and input tokens, how to relate ICL to gradient descent in the setting of softmax attention instead of linear attention?* (2) *What are the specific forms of the training data and loss function in the gradient descent process corresponding to ICL?* In addressing these two questions, we will explore the gradient descent process corresponding to ICL from the perspective of representation learning.

### 3.1 Connecting Softmax Attention with Kernels

Before we begin establishing the connection between ICL and gradient descent, we need to firstly rethink softmax attention with kernel methods. Dai et al. [2022] connect ICL with gradient descent under the linear attention setting. In fact, it is completely feasible to interpret ICL under softmax attention with the help of kernel methods. We define the attention block as

$$\boldsymbol{A} = \text{softmax}\left((\boldsymbol{W}_K\boldsymbol{X})^T\boldsymbol{W}_Q\boldsymbol{X}/\sqrt{d_o}\right), \tag{5}$$

which can be viewed as the product of an unnormalized part $\boldsymbol{A}_u$ and a normalizing multiplier $\boldsymbol{D}$, that is,

$$\boldsymbol{A} = \boldsymbol{A}_u\boldsymbol{D}^{-1}, \ \ \boldsymbol{A}_u = \exp\left((\boldsymbol{W}_K\boldsymbol{X})^T\boldsymbol{W}_Q\boldsymbol{X}/\sqrt{d_o}\right), \ \ \boldsymbol{D} = \text{diag}(\mathbf{1}_N^T\boldsymbol{A}_u), \tag{6}$$

where $\exp(\cdot)$ is element-wise. Similar in [Choromanski et al., 2020], we define softmax kernel $K_{sm} :$ $\mathbb{R}^{d_o} \times \mathbb{R}^{d_o} \to \mathbb{R}_+$ as $K_{sm}(\boldsymbol{x}, \boldsymbol{y}) = e^{\boldsymbol{x}^T\boldsymbol{y}} = e^{\frac{\|\boldsymbol{x}\|^2 + \|\boldsymbol{y}\|^2}{2}}K_{guass}(\boldsymbol{x}, \boldsymbol{y})$ where $K_{guass} = e^{-\|\boldsymbol{x}-\boldsymbol{y}\|^2/2}$ is the guassian kernel when the variance $\sigma^2 = 1$. According to Mercer's theorem [Mercer, 1909], there exists some mapping function $\phi : \mathbb{R}^{d_o} \to \mathbb{R}^{d_r}$ satisfying that $K_{sm}(\boldsymbol{x}, \boldsymbol{y}) = \phi(\boldsymbol{x})^T\phi(\boldsymbol{y})$. Thus,

noting that when omitting the $\sqrt{d_o}$-renormalization and equivalently normalize key and value vectors in Eq (6), every entry in the unnormalized part $\boldsymbol{A}_u$ can be seen as the output of softmax kernel $K_{sm}$ defined for the mapping $\phi$, which can be formulated as:

$$\boldsymbol{A}_u(i,j) = \exp\left((\boldsymbol{W}_K\boldsymbol{x}_i)^T\boldsymbol{W}_Q\boldsymbol{x}_j\right) = K_{sm}(\boldsymbol{W}_K\boldsymbol{x}_i, \boldsymbol{W}_Q\boldsymbol{x}_j) = \phi(\boldsymbol{W}_K\boldsymbol{x}_i)^T\phi(\boldsymbol{W}_Q\boldsymbol{x}_j). \quad (7)$$

There have been many forms of mapping function $\phi(\cdot)$ used in linear Transformers research to approximate this non-negative kernel [Choromanski et al., 2020, Katharopoulos et al., 2020, Peng et al., 2021, Lu et al., 2021]. For example, we can choose $\phi(\cdot)$ as positive random features which has the form $\phi(\boldsymbol{x}) = e^{\boldsymbol{w}^T\boldsymbol{x} - \|\boldsymbol{x}\|^2/2}$ to achieve unbiased approximation [Choromanski et al., 2020]. Alternatively, we can also choose $\phi(\boldsymbol{x}) = \mathrm{elu}(\boldsymbol{x}) + 1$ proposed by Katharopoulos et al. [2020].

### 3.2 The Gradient Descent Process of ICL

Now, we begin to establish the connection between the ICL inference process of a softmax attention layer and gradient descent. We focus on a softmax attention layer in a trained Transformer model, where the parameters $\{\boldsymbol{W}_Q, \boldsymbol{W}_K, \boldsymbol{W}_V\}$ have been determined and the input $\boldsymbol{X} = [\boldsymbol{X}_D, \boldsymbol{X}_T]$ has the form introduced in Section 2.1. Then, after the inference by one attention layer, the query token at position $T+1$ will have the form $\boldsymbol{h}'_{T+1}$ formulated by Eq (1).

On the other hand, given a specific softmax kernel mapping function $\phi(\boldsymbol{x})$ that satisfies Eq (7), we can define the dual model for the softmax attention layer as

$$f(\boldsymbol{x}) = \boldsymbol{W}\phi(\boldsymbol{x}), \quad (8)$$

where $\boldsymbol{W} \in \mathbb{R}^{d_o \times d_r}$ is parameters. We assume that the dual model obtains its updated weights $\widehat{\boldsymbol{W}}$ after undergoing one step of gradient descent with some loss function $\mathcal{L}$. Subsequently, when we take $\boldsymbol{z}_{test} = \boldsymbol{W}_Q\boldsymbol{x}'_{T+1}$ as the test input, we can obtain its test prediction as

$$\hat{\boldsymbol{y}}_{test} = f(\boldsymbol{z}_{test}) = f\left(\boldsymbol{W}_Q\boldsymbol{x}'_{T+1}\right) = \widehat{\boldsymbol{W}}\phi\left(\boldsymbol{W}_Q\boldsymbol{x}'_{T+1}\right).$$

We will show that $\boldsymbol{h}'_{T+1}$ in Eq (1), is strictly equivalent to the above test prediction $\hat{\boldsymbol{y}}_{test}$, which implies that the inference process of ICL involves a gradient descent step on the dual model. This can be illustrated by the following theorem:

**Theorem 3.1.** *The query token $\boldsymbol{h}'_{T+1}$ obtained through ICL inference process with one softmax attention layer, is equivalent to the test prediction $\hat{\boldsymbol{y}}_{test}$ obtained by performing one step of gradient descent on the dual model $f(\boldsymbol{x}) = \boldsymbol{W}\phi(\boldsymbol{x})$. The form of the loss function $\mathcal{L}$ is:*

$$\mathcal{L} = -\frac{1}{\eta D}\sum_{i=1}^{N}\left(\boldsymbol{W}_V\boldsymbol{x}_i\right)^T\boldsymbol{W}\phi(\boldsymbol{W}_K\boldsymbol{x}_i), \quad (9)$$

*where $\eta$ is the learning rate and $D$ is a constant.*

Proof can be found in Appendix A. Theorem 3.1 demonstrates the equivalence between the ICL inference process and gradient descent. Below, we delve into more detailed discussions:

**Training Set and Test Input:** In fact, once the attention layer has already been trained, that is, $\boldsymbol{W}_K, \boldsymbol{W}_Q, \boldsymbol{W}_V$ has been determined, the demonstration tokens $[\boldsymbol{x}_i]_{i=1}^{N}$ will be used to construct a training set for the dual model. Specifically, the training data has the form $\{\boldsymbol{z}_{std}^{(i)}, \boldsymbol{y}_{std}^{(i)}\}_{i=1}^{N}$ where $\boldsymbol{z}_{std}^{(i)} = \boldsymbol{W}_K\boldsymbol{x}_i$ as inputs and $\boldsymbol{y}_{std}^{(i)} = \boldsymbol{W}_V\boldsymbol{x}_i$ as their labels. During training stage, for each input $\boldsymbol{z}_{std}^{(i)}$, the dual model outputs its prediction $\hat{\boldsymbol{y}}^{(i)} = f\left(\boldsymbol{z}_{std}^{(i)}\right) = \boldsymbol{W}\phi\left(\boldsymbol{z}_{std}^{(i)}\right) = \boldsymbol{W}\phi(\boldsymbol{W}_K\boldsymbol{x}_i)$. Then, the loss function Eq (9) can be rewritten as $\mathcal{L} = -\frac{1}{\eta D}\sum_{i=1}^{N}(\boldsymbol{y}_{std}^{(i)})^T\hat{\boldsymbol{y}}^{(i)}$, which can be regarded as the cosine similarity. Then, using this loss function and the training data, we can perform one step of Stochastic Gradient Descent (SGD) on the dual model and obtain the updated $\widehat{\boldsymbol{W}}$. Finally, during the testing stage, we take $\boldsymbol{z}_{test} = \boldsymbol{W}_Q\boldsymbol{x}'_{T+1}$ as the test input to get its prediction which will be consistent with the ICL result $\boldsymbol{h}'_{T+1}$, that is, $\hat{\boldsymbol{y}}_{test} = f(\boldsymbol{z}_{test}) = \widehat{\boldsymbol{W}}\phi\left(\boldsymbol{W}_Q\boldsymbol{x}'_{T+1}\right) = \boldsymbol{h}'_{T+1}$. This process can be illustrated in Figure 1. Demonstration tokens provide information about the training data points and the weight matrix $\widehat{\boldsymbol{W}}$ is optimized to learn sufficient knowledge about demonstrations. This

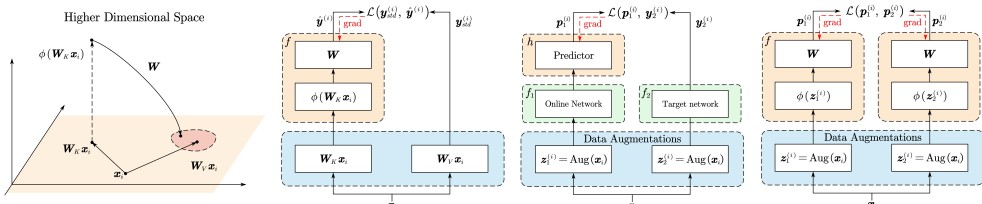

Figure 2: **Left Part:** The representation learning process for the ICL inference by one attention layer. **Remaining Part:** Comparison of the ICL Representation Learning Process (Center Left), Contrastive Learning without Negative Samples (Center Right), and Contrastive Kernel Learning (Right).

gradient descent process using the loss function $\mathcal{L}$ applied to $f(\boldsymbol{x})$ can be seen as the dual form of the ICL inference process of the attention layer.

**Representation Learning Lens:** Even though we have now clarified the details of the gradient descent process of ICL, what does this process more profoundly reveal to us? In fact, for a encoded demonstration token $\boldsymbol{x}_i$, the key and value mapping will generate a pair of features $\boldsymbol{W}_K \boldsymbol{x}_i$ and $\boldsymbol{W}_V \boldsymbol{x}_i$ that exhibit a certain distance from each other, akin to positive samples in contrastive learning. And then, $\phi(\boldsymbol{x})$ projects $\boldsymbol{W}_K \boldsymbol{x}_i$ into a higher-dimensional space to capture deeper features. Finally, the weight matrix $\boldsymbol{W}$, which maps $\phi(\boldsymbol{W}_K \boldsymbol{x}_i)$ back to the original space, is trained to make the mapped vector as close as possible to $\boldsymbol{W}_V \boldsymbol{x}_i$. This process is illustrated in Figure 2. Below, we attempt to understand this process from the perspective of existing representation learning methods introduced in Section 2.2, although we emphasize that there are certain differences between them.

**Comparison with Contrastive Learning without Negative Samples:** If we consider the key and value mapping as two types of data augmentation, then from the perspective of contrastive learning without negative samples, this process can be similarly formalized as

$$\min_{\boldsymbol{W}} \ \mathcal{L}\left(\hat{\boldsymbol{y}}^{(i)}, \boldsymbol{y}_{std}^{(i)}\right) = \mathcal{L}\left(\hat{\boldsymbol{y}}^{(i)}, \mathrm{StopGrad}(\boldsymbol{y}_{std}^{(i)})\right),$$

where $\mathrm{StopGrad}(\cdot)$ is naturally applicable because there are no learning parameters involved in the generation process of the representation $\boldsymbol{y}_{std}^{(i)}$. However, it's important to note that the representation learning process of ICL is much simpler: Firstly, the online and target networks are absent while the augmentations $\boldsymbol{W}_K \boldsymbol{x}_i, \boldsymbol{W}_V \boldsymbol{x}_i$ are directly used as online and target representations respectively. Secondly, the predictor head is useful and not discarded, which is then used during test stage.

**Comparison with Contrastive Kernel Learning:** Given an anchor data $\boldsymbol{x}$ and its positive and negative samples $\boldsymbol{x}^+$, $\boldsymbol{x}^-$, contrastive kernel learning aims to optimize the loss function $\mathcal{L} = f(\boldsymbol{x})(f(\boldsymbol{x}^-) - f(\boldsymbol{x}^+))$ where $f(\boldsymbol{x}) = \boldsymbol{W}\phi(\boldsymbol{x})$. There are significant differences in the representation learning process of ICL: Firstly, it does not involve negative samples. Secondly, there is no corresponding processing for positive samples, leading to parameter updates being solely dependent on the processing of the anchor.

**Extension to More Complicated Scenarios:** Theorem 3.1 can be naturally extended to one single Transformer layer and multiple attention layers. As for one Transformer layer formed in Section 2.1, its dual model $f^+(\boldsymbol{x}) = \boldsymbol{W}\phi(\boldsymbol{x}) + \boldsymbol{b}$ introduces an additional bias $\boldsymbol{b}$ and only $\boldsymbol{W}$ is trained while $\boldsymbol{b}$ remains fixed. In addition, the labels of training set will be $\boldsymbol{y}_{std}^{(i)} = \boldsymbol{W}_F \boldsymbol{W}_K \boldsymbol{x}_i$ where $\boldsymbol{W}_F$ has potential low-rankness property induced by $\mathrm{Relu}(\cdot)$. As for multiple attention layers, the ICL inference process will be equivalent to sequentially performing gradient descent and making predictions on the dual model sequence. We provide more details in Appendix B.

Compared to Dai et al. [2022] considering the connection under linear attention setting, Theorem 3.1 gives explanation for more generally used softmax attention and offers a more detailed exploration of the training process. Additionally, unlike Von Oswald et al. [2023a,b], Ding et al. [2023]'s focus on particular linear regression task and specific configurations of token and parameters, we aim to explain the process of token interactions during ICL inference in a more general setting.

## 3.3 Generalization Bound of the dual gradient descent process for ICL

In this part, we are interested in the generalization bound of the ICL gradient process. When ICL inference is performed for some task $\mathcal{T}$, we cannot provide all demonstrations related to task $\mathcal{T}$ limited by the length of input tokens. We denote $\mathcal{S}_\mathcal{T} \subseteq \mathbb{R}^{d_i}$ as all possible tokens for the task $\mathcal{T}$ and assume that these tokens will be selected according to the distribution $\mathcal{D}_\mathcal{T}$. During a particular instance of ICL inference, let $\mathcal{S} = \{\boldsymbol{x}_i\}_{i=1}^N \subseteq \mathcal{S}_\mathcal{T}$ represent the example tokens we selected. We define the function class as $\mathcal{F} := \{f(\boldsymbol{x}) = \boldsymbol{W}\phi(\boldsymbol{W}_K\boldsymbol{x}) \mid \|\boldsymbol{W}\| \leq w\}$ where $\|\cdot\|$ denotes the Frobenius norm. Generally, ignoring constant term in Eq (9), we consider the representation learning loss as

$$\mathcal{L}(f) = \mathbb{E}_{\boldsymbol{x}\sim\mathcal{D}_\mathcal{T}}\left[-(\boldsymbol{W}_V\boldsymbol{x})^T f(\boldsymbol{x})\right] = \mathbb{E}_{\boldsymbol{x}\sim\mathcal{D}_\mathcal{T}}\left[-(\boldsymbol{W}_V\boldsymbol{x})^T \boldsymbol{W}\phi(\boldsymbol{W}_K\boldsymbol{x})\right], \tag{10}$$

where $f \in \mathcal{F}$ and $\mathcal{D}_\mathcal{T}$ is the distribution for some ICL task $\mathcal{T}$. Correspondingly, the empirical loss will be formulated as $\hat{\mathcal{L}}(f) = -\frac{1}{N}\sum_{i=1}^N (\boldsymbol{W}_V\boldsymbol{x}_i)^T f(\boldsymbol{x}_i)$ and we have $\hat{f} = \arg\min_{f\in\mathcal{F}}\hat{L}(f)$. In addition, we denote the kernel matrix of demonstration tokens $\mathcal{S}$ as $\boldsymbol{K}_\mathcal{S} \in \mathbb{R}^{N\times N}$ where $(\boldsymbol{K}_\mathcal{S})_{i,j} = \langle\phi(\boldsymbol{W}_K\boldsymbol{x}_i), \phi(\boldsymbol{W}_K\boldsymbol{x}_j)\rangle$, that is, the inner product of the feature maps after $\boldsymbol{W}_K$ projection between the $i$-th token and $j$-th token. We state our theorem as follows:

**Theorem 3.2.** *Define the function class as $\mathcal{F} := \{f(\boldsymbol{x}) = \boldsymbol{W}\phi(\boldsymbol{W}_K\boldsymbol{x}) \mid \|\boldsymbol{W}\| \leq w\}$ and let the loss function defined as Eq (10). Consider the given demonstration set as $\mathcal{S} = \{\boldsymbol{x}_i\}_{i=1}^N$ where $\mathcal{S} \subseteq \mathcal{S}_\mathcal{T}$ and $\mathcal{S}_\mathcal{T}$ is all possible demonstration tokens for some task $\mathcal{T}$. With the assumption that $\|\boldsymbol{W}_V\boldsymbol{x}_i\|, \|\boldsymbol{W}\phi(\boldsymbol{W}_K\boldsymbol{x}_i)\| \leq \rho$, then for any $\delta > 0$, the following statement holds with probability at least $1 - \delta$ for any $f \in \mathcal{F}$*

$$\mathcal{L}(\hat{f}) \leq \mathcal{L}(f) + O\left(\frac{w\rho d_o\sqrt{\mathrm{Tr}(\boldsymbol{K}_\mathcal{S})}}{N} + \sqrt{\frac{\log\frac{1}{\delta}}{N}}\right). \tag{11}$$

Proof of 3.2 can be found in Appendix C. Theorem 3.2 provides the generalization bound of the optimal dual model trained on a finite selected demonstration set under a mild assumption that $\|\boldsymbol{W}\|$ is bounded. Intuitively, as the number of demonstration (and therefore the number of demonstration tokens) increases, the generalization error decreases, which is consistent with existing experimental observations [Xie et al., 2021, Garg et al., 2022, Wang et al., 2024].

## 4 Attention Modification Inspired by the Representation Learning Lens

Analyzing the dual gradient descent process of ICL from the perspective of representation learning inspires us to consider that: *Do existing representation learning methods, especially contrastive learning methods, also involve a dual attention inference process? Alternatively, can we modify the attention mechanism by drawing on existing methods?* In fact, since there are lots of mature works in representation learning especially contrastive learning, it is possible for us to achieve this by drawing on these works [He et al., 2020, Chen et al., 2020c, Wu et al., 2018, Chen et al., 2020a, Chen and He, 2021]. We will provide some simple perspectives from the loss function, data augmentations and negative samples to try to adjust attention mechanism. It is worth noting that these modifications are also applicable to the self-attention mechanism, and we will explore these variants in experiments. More details can be seen in Appendix D.

**Attention Modification inspired by the Contrastive Loss:** It can be observed that the unnormalized similarity in Eq (9) allows $\|\boldsymbol{W}\|$ to be optimized to infinity if we ignore the Layer Normalization (LN) layer to prevent this. As for one single attention layer without LN layer, to address this issue, we can introduce regularization term to constrain the norm of $\boldsymbol{W}$, specifically by

$$\mathcal{L} = -\frac{1}{\eta D}\sum_{i=1}^N (\boldsymbol{W}_V\boldsymbol{x}_i)^T \boldsymbol{W}\phi(\boldsymbol{W}_K\boldsymbol{x}_i) + \frac{\alpha}{2\eta}\|\boldsymbol{W}\|_F^2, \tag{12}$$

where $\alpha$ is a hyperparameter. Equivalently, the attention output Eq (1) will be modified as

$$\boldsymbol{h}'_{T+1} = \boldsymbol{W}_V\left[\boldsymbol{X}_D, (1-\alpha)\boldsymbol{X}_T\right]\mathrm{softmax}\left((\boldsymbol{W}_K\boldsymbol{X})^T\boldsymbol{W}_Q\boldsymbol{x}'_{T+1}/\sqrt{d_o}\right). \tag{13}$$

This modification is equivalent to retaining less prompt information for query token during aggregation and relatively more demonstration information will be attended to.

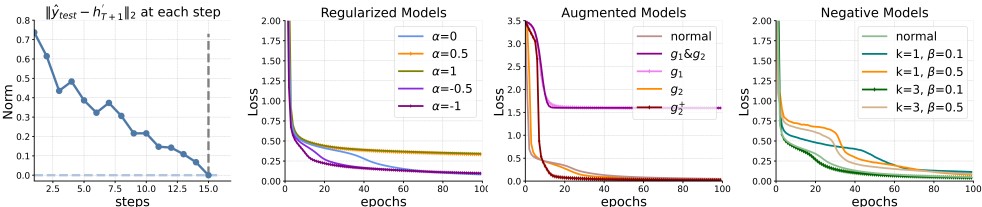

Figure 3: The equivalence between ICL of one softmax attention layer and gradient descent, along with analysis on different model modifications. **Left Part:** $\|\hat{\boldsymbol{y}}_{test} - \boldsymbol{h}'_{T+1}\|_2$ as the gradient descent proceeds under setting $N = 15$; **Remaining Part:** the performance for regularized models (Center Left), augmented models (Center Right) and negative models (Right) with different settings.

**Attention Modification inspired by the Data Augmentation:** If we analogize the key and value mappings to data augmentations in contrastive learning, then for the representation learning process of ICL, these overly simple linear augmentations may limit the model's ability to learn deeper representations. Thus, more complicated augmentations can be considered. Denoting these two augmentations as $g_1$ and $g_2$, the loss function will be modified as

$$\mathcal{L} = -\frac{1}{\eta D} \sum_{i=1}^{N} [g_1(\boldsymbol{W}_V \boldsymbol{x}_i)]^T \boldsymbol{W} \phi(g_2(\boldsymbol{W}_K \boldsymbol{x}_i)).$$

Correspondingly, the attention layer can be adjusted as,

$$\boldsymbol{h}'_{T+1} = g_1(\boldsymbol{W}_V \boldsymbol{X}) \text{softmax} \left( [g_2(\boldsymbol{W}_K \boldsymbol{X})]^T \boldsymbol{W}_Q \boldsymbol{x}'_{T+1} / \sqrt{d_o} \right), \tag{14}$$

where $g_1(\cdot)$ and $g_2(\cdot)$ will be column-wise here. Here we add augmentations for all tokens instead of only demonstration ones to maintain uniformity in the semantic space. In experiments, we simply select MLP for $g_1$ and $g_2$. It's worth noting that here we only propose the framework, and for different tasks, the augmentation approach should be specifically designed to adapt them.

**Attention Modification inspired by the Negative Samples:** Negative samples play a crucial role in preventing feature collapse in contrastive learning methods while the representation learning process of ICL only brings a single pair of features closer, lacking the modeling of what should be pushed apart, which could potentially limit the model's ability to learn representations effectively. Therefore, we can introduce negative samples to address this:

$$\mathcal{L} = -\frac{1}{\eta D} \sum_{i=1}^{N} (\boldsymbol{W}_V \tilde{\boldsymbol{x}}_i)^T \boldsymbol{W} \phi(\boldsymbol{W}_K \boldsymbol{x}_i), \quad \tilde{\boldsymbol{x}}_i = \boldsymbol{x}_i - \frac{\beta}{|\mathcal{N}(i)|} \sum_{j \in \mathcal{N}(i)} \boldsymbol{x}_j,$$

where $\mathcal{N}(i)$ is the set of the negative samples for $\boldsymbol{x}_i$ and $\beta$ is a hyperparameter. Correspondingly, the attention layer is modified as

$$\boldsymbol{h}'_{T+1} = \boldsymbol{W}_V \left[ \tilde{\boldsymbol{X}}_D, \boldsymbol{X}_T \right] \text{softmax} \left( (\boldsymbol{W}_K \boldsymbol{X})^T \boldsymbol{W}_Q \boldsymbol{x}_{T+1} / \sqrt{d_o} \right), \tag{15}$$

where $\tilde{\boldsymbol{X}}_D = [\tilde{\boldsymbol{x}}_1, \tilde{\boldsymbol{x}}_2, ..., \tilde{\boldsymbol{x}}_N]$. Here we simply use other tokens as negative samples and we emphasize that for specific tasks, an appropriate design of negative samples will be more effective.

## 5 Experiments

In this section, we design experiments on synthetic tasks to support our findings and more experiments including on more realistic tasks can be seen in Appendix E. The questions of interest are: *(i) Is the result of ICL inference equivalent to the test prediction of the trained dual model? (ii) Is it potential to improve the attention mechanism from the perspective of representation learning?*

**Linear Task Setting:** Inspired by Von Oswald et al. [2023a], to validate the equivalence and demonstrate the effectiveness of the modifications, we firstly train one softmax self-attention layer using linear regression tasks. We generate the task by $\boldsymbol{s} = \boldsymbol{W}\boldsymbol{t}$ where every element of $\boldsymbol{W} \in \mathbb{R}^{d_s \times d_t}$ is sampled from a normal distribution $\boldsymbol{W}_{ij} \sim \mathcal{N}(0, 1)$ and $\boldsymbol{t}$ from uniform distribution $\boldsymbol{t} \sim$

$U(-1,1)^{d_t}$. We set $d_t = 11$ and $d_s = 1$. Then, at each step, we use generated $\{\boldsymbol{x}_i = [\boldsymbol{t}_i; s_i]\}_{i=1}^{N+1}$ to form the input matrix $\boldsymbol{X}$ where the last token will be used as the query token and the label part will be masked, that is, $\boldsymbol{x}_{N+1} = [\boldsymbol{t}_i; 0]$. Here we consider only one query token ($T = 0$) and we denote $\boldsymbol{x}'_{T+1} = \boldsymbol{x}_{N+1}$ to maintain consistency of notation in Section 2.1. Finally, the attention layer is trained to predict $\hat{s}_{N+1}$ to approximate the true label $s_{N+1}$ using mean square error (MSE) loss.

**Model Setting:** It is worth noting that to facilitate direct access to the dual model, we use positive random features as kernel mapping functions (Performer architecture [Choromanski et al., 2020]) to approximate the standard softmax attention, that is, $\phi(\boldsymbol{x}) = e^{\boldsymbol{w}^T \boldsymbol{x} - \|\boldsymbol{x}\|^2/2}$ where $\boldsymbol{w} \sim \mathcal{N}(0, I)$. We set the dimension of the random features as $d_r = 100(d_t + d_s) = 1200$ to obtain relatively accurate estimation. After training, the weights of the attention layer have been determined. Thus, given specified input $\boldsymbol{X}$, we can construct the dual model $f(\boldsymbol{x}) = \boldsymbol{W}\phi(\boldsymbol{x})$ and its corresponding training data and test input according to Theorem 3.1.

We perform three experiments under different random seeds for linear regression tasks with the results of one presented in Figure 3. In addition, we also conduct more experiments including these on trigonometric, exponential synthetic regression tasks and more realistic tasks. More details of experiments setting and results can be found in Appendix E. We mainly discuss the results on the linear regression task as follows.

**Equivalence Between ICL and Gradient Descent:** To answer the first question, we generate the test input $\boldsymbol{X}_{test}$ using the same method as training and obtain the ICL result of the query token $\boldsymbol{h}'_{T+1}$. On the other hand, we use $\boldsymbol{X}_{test}$ to train the dual model according to Theorem 3.1 and get the test prediction $\hat{\boldsymbol{y}}_{test}$. The result is shown in the left part part of Figure 3. It can be observed that after $N = 15$ epochs training on the dual model, the test prediction $\hat{\boldsymbol{y}}_{test}$ is exactly equivalent to the ICL inference result $\boldsymbol{h}'_{T+1}$ by one softmax attention layer, which aligns with our analysis in Theorem 3.1. More detailed experiments can be seen in Appendix E.1.

**Analysis on the Modifications:** In Section 4, we discussed different modifications to the attention mechanism from perspectives of contrastive loss, data augmentation and negative samples. Here we call these modifications regularized models, augmented models and negative models respectively. More details of modifications for self-attention mechanism can be seen in Appendix D.

For regularized models, we vary different $\alpha$ to investigate the impact on pretraining performance under the same setting, as shown in the center left part of Figure 3. It can be observed that when $\alpha > 0$, the regularized models converges to a poorer result while when $\alpha < 0$, the model converges faster and achieves final results comparable to the normal model without regularization ($\alpha = 0$). At least for this setting, this is a little contrary to our initial intention of applying regularization to the contrastive loss where $\alpha$ should be positive. We explain it that the appropriate $\alpha$ contributes to achieving a full-rank attention matrix as stated in Appendix D, preserving information and accelerating convergence.

For augmented models, we simply choose a single-layer MLP for $g_1(\cdot)$ and $g_2(\cdot)$ as data augmentations to enhance the value and key embeddings respectively in Eq (14) and we choose GELU [Hendrycks and Gimpel, 2016] as the activation function. It can be observed in the center right part of Figure 3 that when we only use $g_2$, that is, only provide augmentation for keys, the model actually shows slightly faster convergence than other cases. Furthermore, when we use two-layer MLP as $g_2^+(\boldsymbol{x})$ as a more complicated augmentation function, the result indicates that although the model initially converges slightly slower due to the increased number of parameters, it eventually accelerates convergence and achieves a better solution. This indicates that appropriate data augmentation indeed have the potential to enhance the capabilities of the attention layer.

For negative models, we select the $k$ tokens with the lowest attention scores as negative samples for each token. From Eq (15), we can see that it is equivalent to subtracting a certain value from the attention scores corresponding to those negative samples. We vary the number of negative samples $k$ and $\beta$ in Eq (15) and the results are shown in the right part of Figure 3. It can be found that the model has the potential to achieve slightly faster convergence with appropriate settings ($k = 3$ and $\beta = 0.1$). In fact, it can be noted that in the original attention mechanism, attention scores are always non-negative, indicating that some irrelevant information will always be preserved to some extent. However, in the modified structure, attention scores can potentially become negative, which makes the model more flexible to utilize information. Certainly, as we discussed in Section 4, for different tasks, more refined methods of selecting augmentations and constructing negative samples may be more effective and we also leave these aspects for future.

# 6 Related Work

Since Transformers have shown remarkable ICL abilities [Brown et al., 2020], many works have aimed to analyze the underlying mechanisms [Garg et al., 2022, Wang et al., 2023]. To explain how Transformers can learn new tasks without parameter updates given few demonstrations, an intuitive idea is to link ICL with (implicit) gradient updates. The most relevant work to ours is that of Dai et al. [2022], which utilizes the dual form to understand ICL as an implicit fine-tuning (gradient descent) of the original model under a linear attention setting [Aiserman et al., 1964, Irie et al., 2022]. They design a specific fine-tuning setting where only the parameters for the key and value projection are updated and the causal language modeling objective is adopted. In this context, they find ICL will have common properties with fine-tuning. Based on this, Deutch et al. [2024] investigate potential shortcomings in the evaluation metrics used by Dai et al. [2022] in real model assessments and propose a layer-causal GD variant that performs better in simulating ICL. As a comparison, our research also uses the dual form to analyze the nonlinear attention layer and explores the specific form of the loss used in the training process. However, we link ICL to the gradient descent performed on the dual model rather than fine-tuning the original model. The former process utilizes a self-supervised representation learning loss formalized as Eq (9) determined by the attention structure itself while performing supervised fine-tuning on the original model is often determined by task-specific training objectives (or manually specified causal language modeling objective Dai et al. [2022]). A more formal and detailed comparison can be found in Appendix F.

Additionally, many other works also link ICL with gradient descent, aiming to explore the Transformer's ability to perform gradient descent algorithms to achieve ICL [Bai et al., 2023, Schlag et al., 2021]. Akyürek et al. [2022] reveal that under certain constructions, Transformer can implement simple basic operations (mov, mul, div and aff), which can be combined to further perform gradient descent. Von Oswald et al. [2023a] provide a simple and appealing construction for solving least squares solutions in the linear attention setting. Subsequently, Zhang et al. [2023a], Ahn et al. [2023], Mahankali et al. [2023] provide theoretical evidence showing that the local or global minima will have a form similar to this specific construction proposed by Von Oswald et al. [2023a] under certain assumptions. These works, both experimentally and theoretically, often focus on specific linear regression tasks ($y = \boldsymbol{w}^T \boldsymbol{x}$) and specific structured input format where each token takes the form $[\boldsymbol{x}, y]$ consisting of the input part $\boldsymbol{x}$ and the label part $y$. In addition, the label part of the final query to be predicted is masked, represented as $[\boldsymbol{x}, 0]$. Subsequent works have expanded this exploration under more complicated setups, including examining nonlinear attention instead of linear attention[Cheng et al., 2023, Collins et al., 2024], using unstructured inputs rather than structured ones[Xing et al., 2024], and considering casual or autoregressive setting[Ding et al., 2023, Von Oswald et al., 2023b]. As a comparison to these works, our work does not target specific tasks like linear regression; therefore, we do not make detailed assumptions about the model weights (simply treated as weights after pre-training) or specific input forms. Instead, we aim to view the ICL inference process from the perspective of representation learning in the dual model. However, we would like to point out that under these specific weight and input settings, an intuitive explanation can also be provided from a representation learning perspective (see Appendix F). We also notice that Shen et al. [2023] experimentally show that there may exist differences between ICL inference in LLMs and the fine-tuned models in real-world scenarios from various perspectives and assumptions used in previous works may be strong. As mentioned earlier, our analysis primarily focus on linking ICL with gradient descent on the dual model of a simplified Transformer rather than fine-tuning the original model. Analyzing more realistic models will also be our future directions.

# 7 Conclusion and Impact Statements

In this paper, we establish a connection between the ICL process of Transformers and gradient descent of the dual model, offering novel insights from a representation learning lens. Based on this, we propose modifications for the attention layer and experiments under our setup demonstrate their potential. Although we have made efforts in understanding ICL, there are still some limitations in our analysis: (1) our work primarily focuses on the simplified Transformer and the impact of structures like layer normalization, residual connections, and others requires more nuanced analysis; (2) for more tasks and settings, the proposed model modifications may require more nuanced design and validation. We leave these aspects for future exploration. And we believe that this work mainly studies the theory of in-context learning, which does not present any foreseeable societal consequence.

# 8 Acknowledgements

We sincerely appreciate the anonymous reviewers for their helpful suggestions and constructive comments. This research was supported by National Natural Science Foundation of China (No.62476277, No.6207623), Beijing Natural Science Foundation (No.4222029), CCF-ALIMAMA TECH Kangaroo Fund (No.CCF-ALIMAMA OF 2024008), and Huawei-Renmin University joint program on Information Retrieval. We also acknowledge the support provided by the fund for building worldclass universities (disciplines) of Renmin University of China and by the funds from Beijing Key Laboratory of Big Data Management and Analysis Methods, Gaoling School of Artificial Intelligence, Renmin University of China, from Engineering Research Center of Next-Generation Intelligent Search and Recommendation, Ministry of Education, from Intelligent Social Governance Interdisciplinary Platform, Major Innovation & Planning Interdisciplinary Platform for the "DoubleFirst Class" Initiative, Renmin University of China, from Public Policy and Decision-making Research Lab of Renmin University of China, and from Public Computing Cloud, Renmin University of China.

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

# A Details of Theorem 3.1

We repeat Theorem 3.1 as follows and provide proof and more discussion for it.

**Theorem A.1.** *The query token $\boldsymbol{h}'_{T+1}$ obtained through ICL inference process with one softmax attention layer, is equivalent to the test prediction $\hat{\boldsymbol{y}}_{test}$ obtained by performing one step of gradient descent on the dual model $f(\boldsymbol{x}) = \boldsymbol{W}\phi(\boldsymbol{x})$. The form of the loss function $\mathcal{L}$ is:*

$$\mathcal{L} = -\frac{1}{\eta D} \sum_{i=1}^{N} (\boldsymbol{W}_V \boldsymbol{x}_i)^T \boldsymbol{W}\phi(\boldsymbol{W}_K \boldsymbol{x}_i), \tag{16}$$

*where $\eta$ is the learning rate and $D$ is a constant.*

*Proof.* The derivative of $\mathcal{L}$ with respect to $\boldsymbol{W}$ should be:

$$\frac{\partial \mathcal{L}}{\partial \boldsymbol{W}} = -\left[\sum_{i=1}^{N} \frac{1}{\eta D} \boldsymbol{W}_V \boldsymbol{x}_i \otimes \phi(\boldsymbol{W}_K \boldsymbol{x}_i)\right].$$

Thus, after one step of gradient descent, the learned $\widehat{\boldsymbol{W}}$ will be

$$\widehat{\boldsymbol{W}} = \boldsymbol{W}_0 - \eta \frac{\partial \mathcal{L}}{\partial \boldsymbol{W}} = \boldsymbol{W}_0 + \left[\sum_{i=1}^{N} \frac{1}{D} \boldsymbol{W}_V \boldsymbol{x}_i \otimes \phi(\boldsymbol{W}_K \boldsymbol{x}_i)\right], \tag{17}$$

where $\boldsymbol{W}_0$ is the initialization of the reference model and $\eta$ is the learning rate. So the test prediction will be

$$\hat{\boldsymbol{y}}_{test} = \boldsymbol{W}_0 \phi\left(\boldsymbol{W}_Q \boldsymbol{x}'_{T+1}\right) + \left[\sum_{i=1}^{N} \frac{1}{D} \boldsymbol{W}_V \boldsymbol{x}_i \otimes \phi(\boldsymbol{W}_K \boldsymbol{x}_i)\right] \phi\left(\boldsymbol{W}_Q \boldsymbol{x}'_{T+1}\right). \tag{18}$$

On the other hand, from the perspective of ICL process with one attention layer, with Eq (7) in our mind, we can rewrite Eq (1) as

$$\boldsymbol{h}'_{T+1} = \boldsymbol{W}_V \boldsymbol{X} \text{softmax}\left(\frac{(\boldsymbol{W}_K \boldsymbol{X})^T \boldsymbol{W}_Q \boldsymbol{x}'_{T+1}}{\sqrt{d_o}}\right)$$

$$= \frac{1}{D'} \boldsymbol{W}_V [\boldsymbol{X}_D, \boldsymbol{X}_T] [\phi(\boldsymbol{W}_K \boldsymbol{X}_D), \phi(\boldsymbol{W}_K \boldsymbol{X}_T)]^T \phi(\boldsymbol{W}_Q \boldsymbol{x}'_{T+1})$$

$$= \frac{1}{D'} [\boldsymbol{V}_D, \boldsymbol{V}_T] [\phi(\boldsymbol{K}_D), \phi(\boldsymbol{K}_T)]^T \phi(\boldsymbol{q}),$$

where we use $[\boldsymbol{V}_D, \boldsymbol{V}_T] = \boldsymbol{W}_V [\boldsymbol{X}_D, \boldsymbol{X}_T]$, $[\boldsymbol{K}_D, \boldsymbol{K}_T] = \boldsymbol{W}_K [\boldsymbol{X}_D, \boldsymbol{X}_T]$, $\boldsymbol{q} = \boldsymbol{W}_Q \boldsymbol{x}'_{T+1}$ for simplify and $D' = \boldsymbol{1}_N^T \phi(\boldsymbol{K}_D)^T \phi(\boldsymbol{q}) + \boldsymbol{1}_T^T \phi(\boldsymbol{K}_T)^T \phi(\boldsymbol{q})$ is a constant to normalize the equivalent attention block. Further, we expand the above equation to connect the inference process of ICL using softmax attention with the gradient descent as follows

$$\boldsymbol{h}'_{T+1} = \frac{1}{D'} \boldsymbol{V}_T \phi(\boldsymbol{K}_T)^T \phi(\boldsymbol{q}) + \frac{1}{D'} \boldsymbol{V}_D \phi(\boldsymbol{K}_D)^T \phi(\boldsymbol{q})$$

$$= \boldsymbol{W}'_0 \phi(\boldsymbol{q}) + \frac{1}{D'}\left[\sum_{i=1}^{N} \boldsymbol{V}_D^{(i)} \otimes \phi(\boldsymbol{K}_D^{(i)})\right] \phi(\boldsymbol{q})$$

where $\boldsymbol{W}'_0 = \frac{1}{D'} \boldsymbol{V}_T \phi(\boldsymbol{K}_T)^T$ and $\boldsymbol{V}_D^{(i)}, \boldsymbol{K}_D^{(i)}$ are the $i$-th column vetors respectively.

Then, in Eq (18), when setting the initialization $\boldsymbol{W}_0 = \boldsymbol{W}'_0$ and the constant $D = D'$, we will find that

$$\hat{\boldsymbol{y}}_{test} = \boldsymbol{W}_0 \phi(\boldsymbol{q}) + \frac{1}{D}\left[\sum_{i=1}^{N} \boldsymbol{V}_D^{(i)} \otimes \phi(\boldsymbol{K}_D^{(i)})\right] \phi(\boldsymbol{q}) = \boldsymbol{h}'_{T+1}, \tag{19}$$

which means $\hat{\boldsymbol{y}}_{test}$ is strictly equivalent to $\boldsymbol{h}'_{T+1}$. Thus, we have completed our proof. $\square$

Given a reference model $f(\boldsymbol{x}) = \boldsymbol{W}\phi(\boldsymbol{x})$, by comparing Eq (19) and Eq (4), we can easily observe that the gradient descent on the loss function $\mathcal{L}$ applied to $f(\boldsymbol{x})$ is the dual form of the inference process of ICL, where $\boldsymbol{V}_D^{(i)}$, $\phi(\boldsymbol{K}_D^{(i)})$ and $\phi(\boldsymbol{q})$ play the roles of backpropagation signals, training inputs and test inputs respectively. Recalling the form of Eq (4), we can interpret the $\boldsymbol{W}_0$ as the initialization of the weight matrix which provide the information under the zero-shot case while the second part in Eq (19) shows that the demonstration examples in ICL acts as the training samples in gradient descent. The reference model $f(\boldsymbol{x}) = \boldsymbol{W}\phi(\boldsymbol{x})$, initialized with $\boldsymbol{W}_0$, will have test prediction $\hat{\boldsymbol{y}}_{test} = \boldsymbol{h}'_{T+1}$ after training. This is also why we refer to it as the dual model of the softmax attention layer. We also note that for different demonstrations, even though the model has the same query input, the different given demonstrations will result in different output results. This is equivalent to the dual model performing gradient descent in different directions from the same initialization.

## B  Extensions to more complex scenarios

In Theorem 3.1, we provided the dual form of gradient descent for the ICL of one softmax attention layer. Here, we extend the conclusion to more complex scenarios, including one Transformer layer (attention layer plus one FFN layer) and multiple attention layers.

### B.1  Extension to one Transformer Layer

As for one Transformer layer introduced in Section 2.1, we define the new dual model as
$$f^+(\boldsymbol{x}) = \boldsymbol{W}\phi(\boldsymbol{x}) + \boldsymbol{b}. \tag{20}$$
We will show that after performing gradient descent on $\boldsymbol{W}$, the test output $\hat{\boldsymbol{y}}_{test} = f^+(\boldsymbol{W}_Q\boldsymbol{x}'_{T+1})$ will be equivalent to $\hat{\boldsymbol{x}}'_{T+1}$. Our theorem is given as follows.

**Theorem B.1.** *The output $\hat{\boldsymbol{x}}'_{N+1}$ of ICL inference process with one Transformer layer, is strictly equivalent to the test prediction of its dual model $f^+(\boldsymbol{x}) = \boldsymbol{W}\phi(\boldsymbol{x}) + \boldsymbol{b}$, where $f(\boldsymbol{x})$ is trained under the loss function $\mathcal{L}$ formed as*
$$\mathcal{L} = -\frac{1}{\eta D}\sum_{i=1}^{N}(\boldsymbol{W}_F\boldsymbol{W}_V\boldsymbol{x}_i)^T(\boldsymbol{W}\phi(\boldsymbol{W}_K\boldsymbol{x}_i) + \boldsymbol{b}), \tag{21}$$
*where $\eta$ is the learning rate, $D$ is a constant, and $\boldsymbol{W}_F$ will be determined once the specified pre-trained model, demonstrations and query tokens are given.*

*Proof.* Recalling the proof of Theorem 3.1, we can rewrite Eq (1) as
$$\boldsymbol{h}'_{T+1} = \boldsymbol{W}_0\phi\left(\boldsymbol{W}_Q\boldsymbol{x}'_{T+1}\right) + \left[\sum_{i=1}^{N}\frac{1}{D}\boldsymbol{W}_V\boldsymbol{x}_i \otimes \phi(\boldsymbol{W}_K\boldsymbol{x}_i)\right]\phi\left(\boldsymbol{W}_Q\boldsymbol{x}'_{T+1}\right) \tag{22}$$
where $D = \boldsymbol{1}_N^T\phi(\boldsymbol{W}_K\boldsymbol{X}_D)^T\phi(\boldsymbol{W}_Q\boldsymbol{x}'_{T+1}) + \boldsymbol{1}_T^T\phi(\boldsymbol{W}_K\boldsymbol{X}_T)^T\phi(\boldsymbol{W}_Q\boldsymbol{x}'_{T+1})$ is a constant to normalize the attention scores and $\boldsymbol{W}_0 = \frac{1}{D}(\boldsymbol{W}_V\boldsymbol{X}_T)\phi(\boldsymbol{W}_K\boldsymbol{X}_T)^T$. Furthermore, $\boldsymbol{h}'_{N+1}$ will be taken as input for the FFN sublayer and the Eq (2) can be rewritten as
$$\hat{\boldsymbol{x}}'_{T+1} = \boldsymbol{W}_2\boldsymbol{I}_M(\boldsymbol{W}_1\boldsymbol{h}'_{T+1} + \boldsymbol{b}_1) + \boldsymbol{b}_2 = \boldsymbol{W}_2\boldsymbol{I}_M\boldsymbol{W}_1\boldsymbol{h}'_{T+1} + \boldsymbol{W}_2\boldsymbol{I}_M\boldsymbol{b}_1 + \boldsymbol{b}_2,$$
where $\boldsymbol{I}_M \in \mathbb{R}^{d\times d}$ is a diagonal matrix whose $i$-th diagonal element will be one if $(\boldsymbol{W}_1\boldsymbol{h}'_{T+1} + \boldsymbol{b}_1)_i \geq 0$ otherwise be zero. We need to note that this process is reasonable: for given demonstration and query tokens, once the parameters $\{\boldsymbol{W}_Q, \boldsymbol{W}_K, \boldsymbol{W}_V, \boldsymbol{W}_1, \boldsymbol{b}_1\}$ of the Transformer layer are fixed after training, $\boldsymbol{I}_M$ will be determined implicitly (otherwise, $\boldsymbol{I}_M$ would be a function that varies with these settings). For simplify, we rewrite $\hat{\boldsymbol{x}}'_{T+1}$ as
$$\hat{\boldsymbol{x}}'_{T+1} = \boldsymbol{W}_F\boldsymbol{h}_{T+1} + \boldsymbol{b}_F,$$
where $\boldsymbol{W}_F = \boldsymbol{W}_2\boldsymbol{I}_M\boldsymbol{W}_1$ and $\boldsymbol{b}_F = \boldsymbol{W}_2\boldsymbol{I}_M\boldsymbol{b}_1 + \boldsymbol{b}_2$. Furthermore, expanding $\boldsymbol{h}'_{T+1}$ in the above Equation, we get:
$$\hat{\boldsymbol{x}}'_{T+1} = \boldsymbol{W}_F\boldsymbol{W}_0\phi\left(\boldsymbol{W}_Q\boldsymbol{x}'_{T+1}\right) + \left[\sum_{i=1}^{N}\frac{1}{D}\boldsymbol{W}_F\boldsymbol{W}_V\boldsymbol{x}_i \otimes \phi(\boldsymbol{W}_K\boldsymbol{x}_i)\right]\phi\left(\boldsymbol{W}_Q\boldsymbol{x}'_{T+1}\right) + \boldsymbol{b}_F$$
$$= \left[\boldsymbol{W}_F\boldsymbol{W}_0 + \sum_{i=1}^{N}\frac{1}{D}\boldsymbol{W}_F\boldsymbol{W}_V\boldsymbol{x}_i \otimes \phi(\boldsymbol{W}_K\boldsymbol{x}_i)\right]\phi\left(\boldsymbol{W}_Q\boldsymbol{x}'_{T+1}\right) + \boldsymbol{b}_F. \tag{23}$$

On the other hand, we define a reference model:

$$f^+(\boldsymbol{x}) = \boldsymbol{W}\phi(\boldsymbol{x}) + \boldsymbol{b},$$

where $\phi(\cdot)$ is exactly the mapping function satisfying Eq (7) to approximate the softmax kernel. Given the loss formed in Eq (21), we can note that the right part in $\mathcal{L}$ is exactly the output of this reference model when taking $\boldsymbol{W}_K \boldsymbol{x}_i$ as input, that is,

$$\mathcal{L} = -\frac{1}{\eta D} \sum_{i=1}^{N} (\boldsymbol{W}_F \boldsymbol{W}_V \boldsymbol{x}_i)^T (\boldsymbol{W}\phi(\boldsymbol{W}_K \boldsymbol{x}_i) + \boldsymbol{b}) = -\frac{1}{\eta D} \sum_{i=1}^{N} (\boldsymbol{W}_F \boldsymbol{W}_V \boldsymbol{x}_i)^T f^+(\boldsymbol{W}_K \boldsymbol{x}_i).$$

We can calculate the derivative of $\mathcal{L}$ with respect to $\boldsymbol{W}$ as

$$\frac{\partial \mathcal{L}}{\partial \boldsymbol{W}} = -\frac{1}{\eta D} \left[ \sum_{i=1}^{N} \boldsymbol{W}_F \boldsymbol{W}_V \boldsymbol{x}_i \otimes \phi(\boldsymbol{W}_K \boldsymbol{x}_i) \right].$$

Suppose that the weight matrix $\boldsymbol{W}$ in the reference model $f(\boldsymbol{x})$ is initialized as $\boldsymbol{W}_{init}$, then using one step of stochastic gradient descent (SGD) [Amari, 1993] with learning rate $\eta$, the weight matrix $\boldsymbol{W}$ will be updated as

$$\widehat{\boldsymbol{W}} = \boldsymbol{W}_{init} - \eta \frac{\partial \mathcal{L}}{\partial \boldsymbol{W}} = \boldsymbol{W}_{init} + \left[ \sum_{i=1}^{N} \frac{1}{D} \boldsymbol{W}_F \boldsymbol{W}_V \boldsymbol{x}_i \otimes \phi(\boldsymbol{W}_K \boldsymbol{x}_i) \right].$$

Compared to Eq (23), we can set $\boldsymbol{W}_{init} = \boldsymbol{W}_F \boldsymbol{W}_0$, $\boldsymbol{b} = \boldsymbol{b}_F$ and take $\boldsymbol{W}_Q \boldsymbol{x}'_{T+1}$ as test input. Then, after one step update to $\boldsymbol{W}$, the output of the reference model will be

$$
\begin{aligned}
f^+(\boldsymbol{W}_Q \boldsymbol{x}'_{T+1}) &= \widehat{\boldsymbol{W}}\phi(\boldsymbol{W}_Q \boldsymbol{x}'_{T+1}) + \boldsymbol{b} \\
&= \left[ \boldsymbol{W}_{init} + \sum_{i=1}^{N} \frac{1}{D} \boldsymbol{W}_F \boldsymbol{W}_V \boldsymbol{x}_i \otimes \phi(\boldsymbol{W}_K \boldsymbol{x}_i) \right] \phi(\boldsymbol{W}_Q \boldsymbol{x}'_{T+1}) + \boldsymbol{b} \\
&= \left[ \boldsymbol{W}_F \boldsymbol{W}_0 + \sum_{i=1}^{N} \frac{1}{D} \boldsymbol{W}_F \boldsymbol{W}_V \boldsymbol{x}_i \otimes \phi(\boldsymbol{W}_K \boldsymbol{x}_i) \right] \phi(\boldsymbol{W}_Q \boldsymbol{x}'_{T+1}) + \boldsymbol{b}_F = \hat{\boldsymbol{x}}'_{T+1},
\end{aligned}
$$

which implies that if we initialize the reference model $f^+(\boldsymbol{x}) = \boldsymbol{W}\phi(\boldsymbol{x}) + \boldsymbol{b}$ with $\boldsymbol{W}_{init} = \boldsymbol{W}_F \boldsymbol{W}_0$, $\boldsymbol{b} = \boldsymbol{b}_F$, then after one step of gradient descent for $\boldsymbol{W}$, the test output of $f^+(\boldsymbol{W}_Q \boldsymbol{x}_{N+1})$ will be identical to the ICL result of one Transformer layer. Thus, we call the reference model with setting $\boldsymbol{W}_{init} = \boldsymbol{W}_F \boldsymbol{W}_0$, $\boldsymbol{b} = \boldsymbol{b}_F$ as the dual model corresponding to the ICL inference process. Finally, we complete our proof. □

Now, we discuss Theorem B.1 from the following perspectives:

- **Training set and test input:** In fact, we can observe that the loss function $\mathcal{L}$ can be seen as the sum of inner products of $N$ vector-pairs. In Eq (21), the right vector happens to be the predicted output $\hat{\boldsymbol{y}}_{std}^{(i)} = f^+(\boldsymbol{z}_{std}^{(i)}) = \boldsymbol{W}\phi(\boldsymbol{z}_{std}^{(i)}) + \boldsymbol{b}$ of the dual model for training input $\boldsymbol{z}_{std}^{(i)} = \boldsymbol{W}_K \boldsymbol{x}_i$. Correspondingly, the vector on the left can be regarded as the true label $\boldsymbol{y}_{std}^{(i)} = \boldsymbol{W}_F \boldsymbol{W}_V \boldsymbol{x}_i$. In other words, it can be seen that the dual model performs one step SGD given training set $\{\boldsymbol{z}_{std}^{(i)}, \boldsymbol{y}_{std}^{(i)}\}_{i=1}^{N}$ on $\boldsymbol{W}$ using the loss $\mathcal{L}$:

$$\mathcal{L} = \frac{1}{\eta D} \sum_{i=1}^{N} \left( \boldsymbol{y}_{std}^{(i)} \right)^T \hat{\boldsymbol{y}}_{std}^{(i)}.$$

And then taking $\boldsymbol{z}_{test} = \boldsymbol{W}_Q \boldsymbol{x}_i$ as test input, it finally output the prediction $\boldsymbol{y}_{test}$, which achieves the ICL result $\hat{\boldsymbol{x}}_{N+1}$. Compared to Theorem 3.1, after introducing the FFN layer, the main difference is that the labels of the training data become $\boldsymbol{y}_{std}^{(i)} = \boldsymbol{W}_F \boldsymbol{W}_V \boldsymbol{x}_i$ instead of $\boldsymbol{y}_{std}^{(i)} = \boldsymbol{W}_V \boldsymbol{x}_i$. Additionally, compared to $f(\boldsymbol{x})$, an extra bias $\boldsymbol{b}$ is introduced in the new dual model $f^+(\boldsymbol{x})$, which also have a different initialization $\boldsymbol{W}_{init} = \boldsymbol{W}_F \boldsymbol{W}_0$ rather than $\boldsymbol{W}_0$. We also need to note that in the dual model $f^+(\boldsymbol{x})$, only $\boldsymbol{W}$ is trained, while $\boldsymbol{b}$ remains unchanged after initialization.

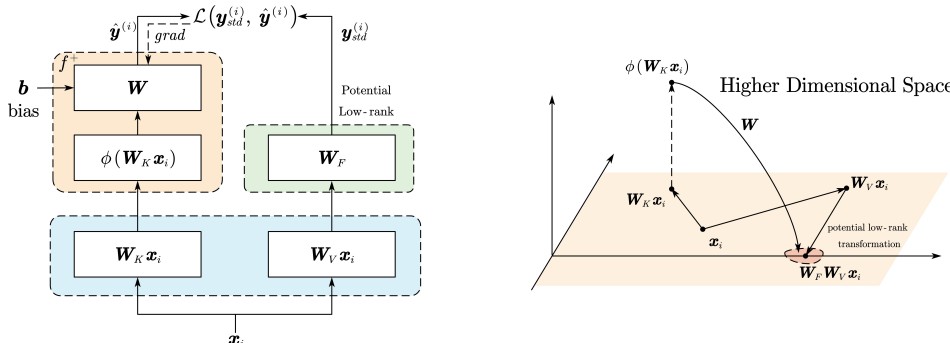

Figure 4: The representation learning process for the ICL inference by one Transformer layer.

- **Potential Low-rankness of $W_F$:** Noting that $W_F = W_2 I_M W_1$ where $W_1 \in \mathbb{R}^{d_h \times d}$, $W_2 \in \mathbb{R}^{d \times d_h}$, $I_M \in \mathbb{R}^{d_h \times d_h}$ (here we assume that $d_i = d_o = d$ for simplify), the rank of $W_F$ will satisfy

$$\text{Rank}(W_F) \leq \min \left\{ \text{Rank}(W_1), \text{Rank}(W_2), \text{Rank}(I_M) \right\}.$$

  We observe that $I_M$ is a diagonal matrix with elements being zero or one, and its rank is determined by the number of non-zero elements. Here, we can make a mild assumption that we can set $\text{Rank}(W_1) = \text{Rank}(W_2) = \min\{d, d_h\}$. This assumption is quite mild as even for any random square matrix as it will be non-singular with probability 1. In addition, we also assume $d_h > d$ which is consistent with settings in practice. Therefore, we get $\text{Rank}(W_1) = \text{Rank}(W_2) = d$, and the upper bound of $\text{Rank}(W_F)$ will be

$$\text{Rank}(W_F) \leq \min \left\{ d, \text{Rank}(I_M) \right\}.$$

  Thus, we can find that if we want to avoid losing information, $W_F$ should strive to maintain $\text{Rank}(I_M) > d$ which will be more easily achieved as $d_h$ becomes larger than $d$. Otherwise, $\text{Rank}(I_M)$ is likely to gradually decrease with an increase in the number of Transformer layers. This explains the necessity of setting $d_h > d$ in practice. In some cases where $\text{Rank}(I_M) < d$, meaning that the number of non-zero elements in $I_M$ or positive elements in $W_1 h_{N+1} + b_1$ is less than $d$, the upper bound of $\text{Rank}(W_F)$ will be $\text{Rank}(I_M)$ and the lower bound of $\text{Rank}(W_F)$ will be given as

$$\text{Rank}(W_F) \geq \text{Rank}(W_2 I_M) + \text{Rank}(I_M W_1) - \text{Rank}(I_M) = \text{Rank}(I_M),$$

  which implies the rank of $W_F$ will exactly equal to $\text{Rank}(I_M)$. We should note that this condition, i.e., $\text{Rank}(I_M) < d$, is easily satisfied when $d_h = d$ or when $d_h$ is slightly larger than $d$ (for example, $d < d_h < 2d$ in an expected sense). Thus, we conclude that $W_F$ has the potential low-rank property.

- **Representation Learning Lens:** For a encoded demonstration representation $x_i$, the key and value projections will generate a pair of feature $W_K x_i$ and $W_V x_i$ to create a certain distance between data representations in space. And then, on the one hand, a potential low-rank transformation $W_F$ is applied to the $W_V x_i$, attempting to compress some information which increases the difficulty of contrastive learning and forces the model to learn better features; on the other hand, $\phi(\cdot)$ projects $W_K x_i$ into a higher-dimensional space to capture deeper-level features. Finally, we need to train the weight matrix $W$, which maps $\phi(W_K x_i)$ back to the original space, aiming to make the mapped vector as close as possible to $W_F W_V x_i$. This interpretation is illustrated in Figure 4.

## B.2 Extension to Multiple Attention Layers

In this part , we extend Theorem 3.1 to multiple attention layers. Here we adopt the attention layer based on PrefixLM [Roberts et al., 2019], where the query tokens can compute attention with all preceding tokens (including itself), while for demonstration ones, attention can be computed between themselves, excluding the query tokens. Existing work [Ding et al., 2023] has theoretically and experimentally explained that PrefixLM achieves better results than CasualLM. In this paper, we

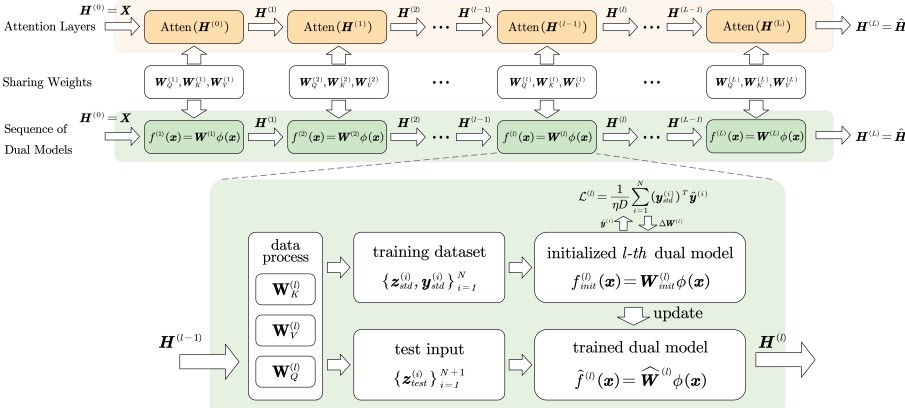

Figure 5: Illustrating the ICL inference process of multiple softmax attention layers from the perspective of dual models. The layer-wise process of ICL can be viewed as a gradual gradient descent on the dual model sequence. The datasets used for each gradient descent, including training data and test input, are obtained from the outputs of the previous dual model before and after training.

assume we have only one query token, that is, there is no query input before the considered query token. With the assumption that $T = 0$, to maintain notational simplicity, we use $\boldsymbol{x}_{N+1}$ to represent the query token here instead of $\boldsymbol{x}'_{T+1}$ and the input will be $\boldsymbol{X} = [\boldsymbol{X}_D, \boldsymbol{x}_{N+1}]$. We assume that there are $L$ attention layers and the output of the $l$-th layer $\boldsymbol{X}^{(l)}$ can be expressed as:

$$\boldsymbol{H}^{(l)} = [\boldsymbol{H}_D^{(l)}, \boldsymbol{h}_{N+1}^{(l)}] = \text{Atten}(\boldsymbol{H}^{(l-1)}; \boldsymbol{W}_Q^{(l)}, \boldsymbol{W}_K^{(l)}, \boldsymbol{W}_V^{(l)}),$$

$$\boldsymbol{H}_D^{(l)} = \boldsymbol{W}_V^{(l)} \boldsymbol{H}_D^{(l-1)} \text{Softmax}\left( \frac{(\boldsymbol{W}_K^{(l)} \boldsymbol{H}_D^{(l-1)})^T \boldsymbol{W}_Q^{(l)} \boldsymbol{H}_D^{(l-1)}}{\sqrt{d}} \right),$$

$$\boldsymbol{h}_{N+1}^{(l)} = \boldsymbol{W}_V^{(l)} \boldsymbol{H}^{(l-1)} \text{Softmax}\left( \frac{(\boldsymbol{W}_K^{(l)} \boldsymbol{H}^{(l-1)})^T \boldsymbol{W}_Q^{(l)} \boldsymbol{h}_{N+1}^{(l-1)}}{\sqrt{d}} \right).$$

where we set $\boldsymbol{H}^{(0)} = \boldsymbol{X} = [\boldsymbol{X}_D, \boldsymbol{x}_{N+1}]$ as the initial input. And the final output of the query token is $\hat{\boldsymbol{h}}_{N+1}^{(L)} = \boldsymbol{h}_{N+1}^{(L)}$. Here, we assume that after training, the parameters $\boldsymbol{W}_Q^{(l)}, \boldsymbol{W}_K^{(l)}, \boldsymbol{W}_V^{(l)} \in \mathbb{R}^{d_o \times d_i}$ are fixed and we set $d_o = d_i = d$.

Next, we extend Theorem 3.1 to the case of multiple softmax attention layers. Formally, we present our result in the following theorem.

**Theorem B.2.** *Given $L$ softmax attention layers whose parameters $\{\boldsymbol{W}_Q^{(l)}, \boldsymbol{W}_K^{(l)}, \boldsymbol{W}_V^{(l)}\}_{l=1}^L$ are fixed after training, the ICL output of these layers is equivalent to sequentially performing one step gradient descent on a sequence of dual models $\left\{ f^{(l)}(\boldsymbol{x}) = \boldsymbol{W}^{(l)} \phi(\boldsymbol{x}) \right\}_{l=1}^L$, where the loss function for the $l$-th dual model is:*

$$\mathcal{L}^{(l)} = -\frac{1}{\eta D^{(l)}} \sum_{i=1}^N \left( \boldsymbol{W}_V^{(l)} \boldsymbol{h}_i^{(l-1)} \right)^T \boldsymbol{W}^{(l)} \phi(\boldsymbol{W}_K^{(l)} \boldsymbol{h}_i^{(l-1)}), \tag{24}$$

*where $\boldsymbol{h}_i^{(l-1)}$ is the output of the $(l-1)$-th attention layer for the $i$-th token, $\eta$ is the learning rate and $D^{(l)}$ is a constant. The input for the $l$-th dual model is generated by the trained $(l-1)$-th dual model.*

*Proof.* Given $\boldsymbol{H}^{(l-1)}$ as the input for the $l$-th attention layer, the inference process of $\boldsymbol{h}_{N+1}^{(l)}$ is

$$\boldsymbol{h}_{N+1}^{(l)} = \boldsymbol{W}_V^{(l)} \boldsymbol{H}^{(l-1)} \text{Softmax}\left(\frac{(\boldsymbol{W}_K^{(l)} \boldsymbol{H}^{(l-1)})^T \boldsymbol{W}_Q^{(l)} \boldsymbol{h}_{N+1}^{(l-1)}}{\sqrt{d}}\right)$$

$$= \boldsymbol{W}_0^{(l)} \phi\left(\boldsymbol{W}_Q^{(l)} \boldsymbol{h}_{N+1}^{(l-1)}\right) + \left[\sum_{i=1}^N \frac{1}{D^{(l)}} \boldsymbol{W}_V^{(l)} \boldsymbol{h}_i^{(l-1)} \otimes \phi(\boldsymbol{W}_K^{(l)} \boldsymbol{h}_i^{(l-1)})\right] \phi\left(\boldsymbol{W}_Q^{(l)} \boldsymbol{h}_{N+1}^{(l-1)}\right),$$

where $D^{(l)} = \boldsymbol{1}_{N+1}^T \phi(\boldsymbol{W}_K^{(l)} \boldsymbol{H}^{(l-1)})^T \phi(\boldsymbol{W}_Q^{(l)} \boldsymbol{h}_{N+1}^{(l-1)})$ is a constant to normalize the attention scores and $\boldsymbol{W}_0^{(l)} = \frac{1}{D^{(l)}}(\boldsymbol{W}_V^{(l)} \boldsymbol{h}_{N+1}^{(l-1)})\phi(\boldsymbol{W}_K^{(l)} \boldsymbol{h}_{N+1}^{(l-1)})^T$. According to Theorem 3.1, we can easily get the dual model $f_{init}^{(l)}(\boldsymbol{h}) = \boldsymbol{W}_{init}^{(l)} \phi(\boldsymbol{h})$ where the initialization is $\boldsymbol{W}_{init}^{(l)} = \boldsymbol{W}_0^{(l)}$. Given the loss function $\mathcal{L}^{(l)}$ formed as Equation 24 and training set $\left\{\boldsymbol{z}_{std}^{(i)}, \boldsymbol{y}_{std}^{(i)}\right\}_{i=1}^N$ where $\boldsymbol{z}_{std}^{(i)} = \boldsymbol{W}_K^{(l)} \boldsymbol{h}_i^{(l-1)}$ and $\boldsymbol{y}_{std}^{(i)} = \boldsymbol{W}_V^{(l)} \boldsymbol{h}_i^{(l)}$, we perform one step SGD with learning rate $\eta$ on weight matrix $\boldsymbol{W}^{(l)}$ and will get trained dual model:

$$\hat{f}^{(l)}(\boldsymbol{x}) = \widehat{\boldsymbol{W}}^{(l)} \phi(\boldsymbol{x}) = (\boldsymbol{W}_{init}^{(l)} + \Delta \boldsymbol{W}^{(l)}) \phi(\boldsymbol{x})$$

$$= \left[\boldsymbol{W}_0^{(l)} + \sum_{i=1}^N \frac{1}{D^{(l)}} \boldsymbol{W}_V^{(l)} \boldsymbol{h}_i^{(l-1)} \otimes \phi(\boldsymbol{W}_K^{(l)} \boldsymbol{h}_i^{(l-1)})\right] \phi(\boldsymbol{x})$$

Taking test input as $\boldsymbol{z}_{test}^{(l)} = \boldsymbol{W}_Q^{(l)} \boldsymbol{h}_{N+1}^{(l-1)}$, the prediction $\hat{f}^{(l)}(\boldsymbol{z}_{test}^{(l)})$ will exactly equal to $\boldsymbol{h}_{N+1}^{(l)}$.

Next, we will show how to obtain $\boldsymbol{H}_D^{(l)}$ through the trained dual model $\hat{f}^{(l)}(\boldsymbol{x})$. And after $\boldsymbol{W}_K^{(l+1)}, \boldsymbol{W}_Q^{(l+1)}, \boldsymbol{W}_V^{(l+1)}$ projections, $\boldsymbol{H}_D^{(l)}$ will constitute the training set as well as the test input for the next dual model $f_{init}^{(l+1)}(\boldsymbol{x})$.

Keeping the initialized dual model $f_{init}^{(l)}(\boldsymbol{x})$ and the trained one $\hat{f}^{(l)}(\boldsymbol{x})$ in mind, we can compute the demonstration token output $\boldsymbol{h}_i^{(l)}$ ($i = 1, 2, ..., N$) of $l$-th attention layer as

$$\boldsymbol{h}_i^{(l)} = \boldsymbol{W}_V^{(l)} \boldsymbol{H}_D^{(l-1)} \text{Softmax}\left(\frac{(\boldsymbol{W}_K^{(l)} \boldsymbol{H}_D^{(l-1)})^T \boldsymbol{W}_Q^{(l)} \boldsymbol{h}_i^{(l-1)}}{\sqrt{d}}\right)$$

$$= \left[\sum_{i=1}^N \frac{1}{D_i^{(l)}} \boldsymbol{W}_V^{(l)} \boldsymbol{h}_i^{(l-1)} \otimes \phi(\boldsymbol{W}_K^{(l)} \boldsymbol{h}_i^{(l-1)})\right] \phi\left(\boldsymbol{W}_Q^{(l)} \boldsymbol{h}_i^{(l-1)}\right)$$

$$= \frac{D^{(l)}}{D_i^{(l)}} \left[\sum_{i=1}^N \frac{1}{D^{(l)}} \boldsymbol{W}_V^{(l)} \boldsymbol{h}_i^{(l-1)} \otimes \phi(\boldsymbol{W}_K^{(l)} \boldsymbol{h}_i^{(l-1)})\right] \phi\left(\boldsymbol{W}_Q^{(l)} \boldsymbol{h}_i^{(l-1)}\right)$$

$$= \frac{D^{(l)}}{D_i^{(l)}} \left[\boldsymbol{W}_{init}^{(l)} + \sum_{i=1}^N \frac{1}{D^{(l)}} \boldsymbol{W}_V^{(l)} \boldsymbol{h}_i^{(l-1)} \otimes \phi(\boldsymbol{W}_K^{(l)} \boldsymbol{h}_i^{(l-1)}) - \boldsymbol{W}_{init}^{(l)}\right] \phi\left(\boldsymbol{W}_Q^{(l)} \boldsymbol{h}_i^{(l-1)}\right)$$

$$= \frac{D^{(l)}}{D_i^{(l)}} \left[\widehat{\boldsymbol{W}}^{(l)} - \boldsymbol{W}_{init}^{(l)}\right] \phi\left(\boldsymbol{W}_Q^{(l)} \boldsymbol{h}_i^{(l-1)}\right) = \frac{D^{(l)}}{D_i^{(l)}} \left[\hat{f}^{(l)}(\boldsymbol{W}_Q^{(l)} \boldsymbol{h}_i^{(l-1)}) - f_{init}^{(l)}(\boldsymbol{W}_Q^{(l)} \boldsymbol{h}_i^{(l-1)})\right]$$

$$(25)$$

where $D_i^{(l)} = \boldsymbol{1}_N^T \phi(\boldsymbol{W}_K^{(l)} \boldsymbol{H}_D^{(l-1)})^T \phi(\boldsymbol{W}_Q^{(l)} \boldsymbol{h}_i^{(l-1)})$ is a constant to normalize the attention scores for $\boldsymbol{h}_i^{(l)}$. Therefore, once we obtain the trained dual model $\hat{f}^{(l)}(\boldsymbol{h})$, we can use the Eq (25) to get the demonstration token output $\boldsymbol{H}_D^{(l)} = [\boldsymbol{h}_1^{(l)}, \boldsymbol{h}_2^{(l)}, ..., \boldsymbol{h}_N^{(l)}]$. These demonstration token outputs, along with the output $\boldsymbol{h}_{N+1}^{(l)}$ for query tokens, will together constitute the training set and test input for the next dual model $f_{init}^{(l+1)}(\boldsymbol{h})$. This process continues layer by layer until we obtain the ultimate ICL output $\boldsymbol{h}_{N+1}^{(L)}$. In summary, the ICL inference process across $L$ attention layers is equivalent to performing gradient descent on $L$ dual models sequentially. Thus, we complete our proof. $\square$

This theorem is a natural extension of Theorem 3.1: when considering the stacking of multiple attention layers, a sequence of dual models is correspondingly generated. Although these dual

models have the same form $f(\boldsymbol{x}) = \boldsymbol{W}\phi(\boldsymbol{x})$, they have different initializations and datasets. As the ICL inference process progresses layer by layer between attention layers, we equivalently perform gradient descent on the dual models one by one. The input $\boldsymbol{H}^{(l)}$ for each attention layer, including demonstration tokens and query tokens, can be obtained from the test output of the dual models. This can be illustrated in Figure 5.

## C  Proof of the Generalization bound

### C.1  Proof of Theorem 3.2

In this part, we provide the proof regarding the generalization boundary in Theorem 3.2. We restate our theorem as follows:

**Theorem C.1.** *Define the function class as $\mathcal{F} := \{f(\boldsymbol{x}) = \boldsymbol{W}\phi(\boldsymbol{W}_K\boldsymbol{x}) \mid \|\boldsymbol{W}\| \le w\}$ and let the loss function defined as Eq (10). Consider the given demonstration set as $\mathcal{S} = \{\boldsymbol{x}_i\}_{i=1}^N$ where $\mathcal{S} \subseteq \mathcal{S}_{\mathcal{T}}$ and $\mathcal{S}_{\mathcal{T}}$ is all possible demonstration tokens for some task $\mathcal{T}$. With the assumption that $\|\boldsymbol{W}_V\boldsymbol{x}_i\|, \|\boldsymbol{W}\phi(\boldsymbol{W}_K\boldsymbol{x}_i)\| \le \rho$, then for any $\delta > 0$, the following statement holds with probability at least $1 - \delta$ for any $f \in \mathcal{F}$*

$$\mathcal{L}(\hat{f}) \le \mathcal{L}(f) + O\left(\frac{w\rho d_o \sqrt{\mathrm{Tr}(\boldsymbol{K}_{\mathcal{S}})}}{N} + \sqrt{\frac{log\frac{1}{\delta}}{N}}\right). \tag{26}$$

*Proof.* Our proof is similar to the Lemma 4.2 in Saunshi et al. [2019], but here we focus on a different function class. Firstly, we consider the classical generalization bound based on the Rademacher complexity of the function class which can refer to Theorem 3.1 in Mohri et al. [2018]. For a real function class $G$ whose functions map from a set $Z$ to $[0, 1]$ and for any $\delta > 0$, if $\mathcal{S}$ is a training set composed by $N$ iid samples $\{\boldsymbol{x}_i\}_{i=1}^N$, then with probability at least $1 - \frac{\delta}{2}$, for all $g \in G$

$$\mathbb{E}\left[g(\boldsymbol{x})\right] \le \frac{1}{N}\sum_{i=1}^N g(\boldsymbol{x}_i) + \frac{2\mathcal{R}_{\mathcal{S}}(G)}{N} + 3\sqrt{\frac{\log\frac{4}{\delta}}{2N}} \tag{27}$$

where $\mathcal{R}_{\mathcal{S}}(G)$ is the traditional Rademacher complexity. By setting $\mathcal{S}$ exactly the demonstration set and $G = \left\{g_f(\boldsymbol{x}) = -(\boldsymbol{W}_V\boldsymbol{x})^T \boldsymbol{W}\phi(\boldsymbol{W}_K\boldsymbol{x}) \big| \|\boldsymbol{W}\| \le w\right\}$, we can apply this bound to our case.

Then, we construct a function class $\tilde{\mathcal{F}} = \left\{\tilde{f}(\boldsymbol{x}) = [f(\boldsymbol{x}); \boldsymbol{W}_V\boldsymbol{x}] = [\boldsymbol{W}\phi(\boldsymbol{W}_K\boldsymbol{x}); \boldsymbol{W}_V\boldsymbol{x}] \big| \|\boldsymbol{W}\| \le w\right\}$ whose functions map from $\mathcal{S}$ to $\mathbb{R}^{2d_o}$. Next, we will first prove $\mathcal{R}_{\mathcal{S}}(G) \le 2\rho\mathcal{R}_{\mathcal{S}}(\tilde{\mathcal{F}})$ and to do this, we need to use the following Lemma:

**Lemma C.2** (Corollary 4 in Maurer [2016]). *Let $Z$ be any set, and $\mathcal{S} = \{\boldsymbol{z}_i\}_{i=1}^M \in Z^M$. Let $\tilde{\mathcal{F}}$ be a class of functions $\tilde{f} : Z \to \mathbb{R}^n$ and $h : \mathbb{R}^n \to \mathbb{R}$ be $L$-Lipschitz. For all $\tilde{f} \in \tilde{\mathcal{F}}$, let $g_{\tilde{f}} = h \circ \tilde{f}$. Then*

$$\mathbb{E}_{\sigma\sim\{\pm1\}^M}\left[\sup_{\tilde{f}\in\tilde{F}}\langle\sigma, (g_{\tilde{f}|\mathcal{S}})\rangle\right] \le \sqrt{2}L \mathbb{E}_{\sigma\sim\{\pm1\}^{nM}}\left[\sup_{\tilde{f}\in\tilde{F}}\langle\sigma, (\tilde{f}_{|\mathcal{S}})\rangle\right] \tag{28}$$

*where $\tilde{f}_{|\mathcal{S}} = \left(\tilde{f}_t(\boldsymbol{z}_j)\right)_{t\in[n],j\in[M]}$.*

We apply Lemma C.2 to our case by setting $Z = \mathbb{R}^{d_i}$, $\mathcal{S}$ to be exactly the demonstration set, $\tilde{\mathcal{F}}$ to be the function class we constructed and $n = 2d$. We also use $h : \mathbb{R}^{2d_o} \to \mathbb{R}$ where $h(\boldsymbol{x}) = -\langle\boldsymbol{x}_{1:d_o}, \boldsymbol{x}_{d_o+1:2d_o}\rangle$ and thus we have $g_{\tilde{f}}(\boldsymbol{x}) = h(\tilde{f}(\boldsymbol{x})) = h([f(\boldsymbol{x}); \boldsymbol{W}_V\boldsymbol{x}]) = -(\boldsymbol{W}_V\boldsymbol{x})^T \boldsymbol{W}\phi(\boldsymbol{W}_K\boldsymbol{x})$. We can find that $g_f(\boldsymbol{x}) = g_{\tilde{f}}(\boldsymbol{x})$ and the left side of inequality (28) is exactly $\mathcal{R}_{\mathcal{S}}(G)$.

Then we can see that $h$ is $\sqrt{2}\rho$-Lipschitz with the assumption that $\|\boldsymbol{W}_V\boldsymbol{x}_i\|, \|\boldsymbol{W}\phi(\boldsymbol{W}_K\boldsymbol{x}_i)\| \le \rho$ and we have $\mathcal{R}_{\mathcal{S}}(G) \le 2\rho\mathcal{R}_{\mathcal{S}}(\tilde{\mathcal{F}})$. Now using Lemma C.2 and the classical generalization bound (27),

we have that with probability at least $1 - \frac{\delta}{2}$

$$\mathcal{L}(\hat{f}) \leq \hat{\mathcal{L}}(\hat{f}) + O\left(\frac{\rho \mathcal{R}_{\mathcal{S}}(\tilde{F})}{N} + \sqrt{\frac{\log \frac{1}{\delta}}{N}}\right), \tag{29}$$

Let $f^* \in \arg\min_{f \in \mathcal{F}} \mathcal{L}(f)$. According to Hoeffding's inequality, with probability at least $1 - \frac{\delta}{2}$, we have that $\hat{\mathcal{L}}(f^*) \leq \mathcal{L}(f^*) + 3\sqrt{\frac{\log \frac{2}{\delta}}{2N}}$. Combining this with (29), the fact that $\hat{\mathcal{L}}(\hat{f}) \leq \hat{\mathcal{L}}(f^*)$ and applying a union bound, we can get that

$$\mathcal{L}(\hat{f}) \leq \mathcal{L}(f) + O\left(\frac{\rho \mathcal{R}_{\mathcal{S}}(\tilde{F})}{N} + \sqrt{\frac{\log \frac{1}{\delta}}{N}}\right). \tag{30}$$

Next, we give the upper bound for $\mathcal{R}_{\mathcal{S}}(\tilde{\mathcal{F}})$.

$$
\begin{aligned}
\mathcal{R}_{\mathcal{S}}(\tilde{\mathcal{F}}) &= \mathbb{E}_{\sigma \sim \{\pm 1\}^{2N d_o}} \left[ \sup_{\|\boldsymbol{W}_j\| \leq w} \sum_{t=1}^{2N d_o} \sigma_t (\tilde{f}_{|\mathcal{S}})_t \right] && \text{(Definition of Rademacher complexity)} \\
&= \mathbb{E}_{\sigma \sim \{\pm 1\}^{N d_o}} \left[ \sup_{\|\boldsymbol{W}\| \leq w} \sum_{j=1}^{d_o} \boldsymbol{W}_j \sum_{i=1}^{N} \sigma_{i,j} \phi(\boldsymbol{W}_K \boldsymbol{x}_i) \right] && (\boldsymbol{W}_V \boldsymbol{x}_i \text{ is independent of } \boldsymbol{W}_j) \\
&\leq \mathbb{E}_{\sigma \sim \{\pm 1\}^{N d_o}} \left[ \sup_{\|\boldsymbol{W}\| \leq w} \sum_{j=1}^{d_o} \|\boldsymbol{W}_j\| \left\| \sum_{i=1}^{N} \sigma_{i,j} \phi(\boldsymbol{W}_K \boldsymbol{x}_i) \right\| \right] && \text{(By Cauchy-Schwartz inequality)} \\
&\leq w d_o \mathbb{E}_{\sigma \sim \{\pm 1\}^{N}} \left[ \left\| \sum_{i=1}^{N} \sigma_i \phi(\boldsymbol{W}_K \boldsymbol{x}_i) \right\| \right] && \text{(Using the fact that } \|\boldsymbol{W}_j\| \leq w) \\
&\leq w d_o \sqrt{ \mathbb{E}_{\sigma \sim \{\pm 1\}^{N}} \left[ \left\| \sum_{i=1}^{N} \sigma_i \phi(\boldsymbol{W}_K \boldsymbol{x}_i) \right\|^2 \right] } && \text{(By Jensen's inequality)} \\
&= w d_o \mathrm{Tr}(\boldsymbol{K}_{\mathcal{S}})
\end{aligned}
$$

Substituting the upper bound of $\mathcal{R}_{\mathcal{S}}(\tilde{\mathcal{F}})$ into (30), we will get that

$$\mathcal{L}(\hat{f}) \leq \mathcal{L}(f) + O\left(\frac{w \rho d_o \sqrt{\mathrm{Tr}(\boldsymbol{K}_{\mathcal{S}})}}{N} + \sqrt{\frac{\log \frac{1}{\delta}}{N}}\right). \tag{31}$$

Thus we finish our proof. □

## C.2 Extension to negative models:

One may also wonder whether the ratio of negative samples mentioned in Section 4 will affect the generalization bounds. In fact, after introducing negative samples and ignoring constant term in Eq (9), we consider the following representation loss:

$$\mathcal{L}(f) = \mathbb{E}_{x \sim \mathcal{D}_{\mathcal{T}}} \left[ -\frac{1}{K} \sum_{j=1}^{K} (\boldsymbol{W} \phi(\boldsymbol{W}_K x))^T (\boldsymbol{W}_V x - \boldsymbol{W}_V x_j^-) \right],$$

where we consider sampling $K$ negative samples for each $x_i$ and $x_j^-$ denotes the $j$-th negative sample for token $x$. Correspondingly, the empirical loss will be considered as $\hat{\mathcal{L}}(f) = -\frac{1}{N} \sum_{i=1}^{N} \frac{1}{K} \sum_{j=1}^{K} (\boldsymbol{W} \phi(\boldsymbol{W}_K x_i))^T (\boldsymbol{W}_V x_i - \boldsymbol{W}_V x_{ij}^-)$ where $x_{ij}^-$ is the $j$-th negative sample for $x_i$. Then, by retaining the other definitions in Section 3.3, corresponding to Theorem 3.2, we can obtain the generalization bound as

$$\mathcal{L}(\hat{f}) \leq \mathcal{L}(f) + O\left(w \rho d_o \sqrt{\mathrm{Tr}(K_S)\left(\frac{5}{N^2} + \frac{1}{rN^3}\right)} + \sqrt{\frac{\log \frac{1}{\delta}}{N}}\right),$$

where $r = \frac{K}{N}$ is excatly the the ratio of the number of negative samples. It can be observed that as the ratio of negative samples increases, the generalization error decreases. However, we also notice that $\frac{5}{N^2} > \frac{1}{rN^3}$ thus the former term dominates, which means the reduction in generalization error due to an increased proportion of negative samples is limited. Nevertheless, we do not rule out the possibility of a tighter generalization bound, which is a promising direction for future research.

*Proof Sketch.* The proof process is similar to that of Theorem 3.2. The main difference lies in the fact that we should firstly define the function class $G = \left\{ -\frac{1}{K} \sum_{j=1}^{K} (\boldsymbol{W}\phi(\boldsymbol{W}_K x_i))^T (\boldsymbol{W}_V x_i - \boldsymbol{W}_V x_j^-) \middle| \|\boldsymbol{W}\| \le w \right\}$ to use the classical bound. In addition, we define $\tilde{F} = \left\{ \tilde{f}(x) = [f(x); \boldsymbol{W}_V x; \boldsymbol{W}_V x_1^-; ...; \boldsymbol{W}_V x_K^-] \middle| \|\boldsymbol{W}\| \le w \right\}$ whose functions map from $\mathcal{S}$ to $\mathbb{R}^{(K+2)d_o}$. Similarly, when using Lemma C.2, we set $Z = \mathbb{R}^{d_i}$, $\tilde{F}$ be the above function class and $n = (K+2)d_o$. We also use $h : \mathbb{R}^{(K+2)d_o} \to \mathbb{R}$ defined as $h(x) = -\frac{1}{K} \sum_{j=1}^{K} x_{1:d_o}^T (x_{d_o+1:2d_o} - x_{(j+1)d_o+1:(j+2)d_o})$. Then we notice that

$$
\frac{\partial h}{\partial x_{1:d_0}} = -\frac{1}{K} \sum_{j=1}^{K} (x_{d_0+1:2d_o} - x_{(j+1)d_o+1:(j+2)d_o}),
$$
$$
\frac{\partial h}{\partial x_{d_o+1:2d_o}} = -x_{1:d_o}, \quad \frac{\partial h}{\partial x_{(j+1)d_o+1:(j+2)d_o}} = \frac{1}{K} x_{1:d_o}. \tag{32}
$$

With the assumption that $\|W_V x\|, \|W\phi(W_K x)\| \le \rho$, we can get that the Frobenius norm of the Jocabian $J$ of $h$ has $\|J\|_F^2 \le 4\rho^2 + \rho^2 + \frac{K}{K^2}\rho^2 = (5 + \frac{1}{K})\rho^2$. Thus we get that $h$ is $\sqrt{5 + \frac{1}{K}}\rho$-Lipschitz. The rest of the proof process is similar to that of Theorem 3.2. Ultimately, we will obtain the aforementioned generalization error. $\square$

# D  Details and More Discussions for Section 4

In this section, we provide a more detailed discussion on improving the model structure from the perspective of representation learning especially contrastive learning, which is presented in Section 4 of the main body. And we also point out the corresponding modifications in the self-attention mechanism, which are adopted in our experiments.

## D.1  More Discussion on the Contrastive Loss

Although we have figured out the representation learning loss of the implicit gradient updates, it can be observed that this loss function has a flaw: due to the lack of normalization for $\boldsymbol{y}_{std}^{(i)}$ and $\hat{\boldsymbol{y}}^{(i)}$ when calculating the cosine distance, the loss can theoretically be optimized to negative infinity. To address this issue, we introduce regularization to constrain the norm of $\boldsymbol{W}$, that is,

$$
\mathcal{L} = -\frac{1}{\eta D} \sum_{i=1}^{N} (\boldsymbol{W}_V \boldsymbol{x}_i)^T \boldsymbol{W}\phi(\boldsymbol{W}_K \boldsymbol{x}_i) + \frac{\alpha}{2\eta} \|\boldsymbol{W}\|_F^2,
$$

where $\alpha$ is a hyperparameter to balance the two parts. As a result, we can see that the gradient update for $\boldsymbol{W}$ will be in an exponentially smoothed manner meaning that a portion of the initial part will be discarded at every step, that is,

$$
\boldsymbol{W}^{(t)} = \boldsymbol{W}^{(t-1)} - \eta \frac{\partial \mathcal{L}}{\partial \boldsymbol{W}} = (1 - \alpha)\boldsymbol{W}^{(t-1)} + \sum_{i=1}^{N} D^{-1} \boldsymbol{W}_V \boldsymbol{h}_i \otimes \phi(\boldsymbol{W}_K \boldsymbol{h}_i).
$$

Equivalently, the inference process of ICL can be seen as the first step of the aforementioned update, and the attention mechanism will be correspondingly adjusted as,

$$
\boldsymbol{h}'_{T+1} = (1 - \alpha)\boldsymbol{W}_0 \phi(\boldsymbol{q}) + D^{-1} \left[ \sum_{i=1}^{N} \boldsymbol{V}_D^{(i)} \otimes \phi(\boldsymbol{K}_D^{(i)}) \right] \phi(\boldsymbol{q}),
$$

which means more demonstration information will be attended to. This will directly result in Eq 13.

This result can be easily extended to self-attention mechanism. As for a self-attention layer, if all other tokens adopt the same modification, the self-attention layer will become

$$\boldsymbol{H} = \boldsymbol{W}_V \boldsymbol{X} \operatorname{softmax}\left(\frac{(\boldsymbol{W}_K \boldsymbol{X})^T \boldsymbol{W}_Q \boldsymbol{X}}{\sqrt{d_o}}\right) - \alpha \boldsymbol{W}_V \boldsymbol{X}$$

$$= \boldsymbol{W}_V \boldsymbol{X} \left[\operatorname{softmax}\left(\frac{(\boldsymbol{W}_K \boldsymbol{X})^T \boldsymbol{W}_Q \boldsymbol{X}}{\sqrt{d_o}}\right) - \alpha \boldsymbol{I}\right],$$

which leads to the model structure incorporating an operation similar to skip connections. Furthermore, to ensure numerical stability, we normalize the attention scores yielding:

$$\boldsymbol{H} = \boldsymbol{W}_V \boldsymbol{X} \cdot \operatorname{Norm}\left(\operatorname{softmax}\left(\frac{(\boldsymbol{W}_K \boldsymbol{X})^T \boldsymbol{W}_Q \boldsymbol{X}}{\sqrt{d_o}}\right) - \alpha \boldsymbol{I}\right),$$

where $\operatorname{Norm}(\cdot)$ is performed column-wise to ensure that the attention scores sum to 1. The above modification reduce the attention score of each token to its own information during aggregation. It is worth noting that, although our initial intention is to impose regularization on the contrastive loss where $\alpha > 0$ to prevent it from diverging to negative infinity, we find in experiments that this modification remains effective even when $\alpha$ is less than 0. We interpret this as possibly stemming from the fact that an appropriate $\alpha$ helps the attention block become full-rank, thereby better preserving information, which can be illustrated by Lemma D.1:

**Lemma D.1.** *Let the attention block $\boldsymbol{A} \in \mathbb{R}^{n \times n}$. There exists some $\delta > 0$ such that, for any $0 < |\alpha| < \delta$, the attention block $\boldsymbol{A} + \alpha \boldsymbol{I}_n$ will become full-rank.*

*Proof.* Define $f(\alpha) = \det(\alpha \boldsymbol{I}_n + \boldsymbol{A})$, which is a polynomial of degree $n$ in $\alpha$. Then, $f(\alpha)$ has only finitely roots. Let $\alpha_1, \alpha_2, \ldots, \alpha_r$ be the non-zero roots of $f(t)$. Now, consider $\delta = \min\{|\alpha_1|, |\alpha_2|, \ldots, |\alpha_r|\}$. For $0 < |\alpha| < \delta$, we can claim that $f(\alpha) = \det(\alpha \boldsymbol{I}_n + \boldsymbol{A}) \neq 0$. Thus, $\boldsymbol{A} + \alpha \boldsymbol{I}_n$ becomes non-singular (full-rank) and we complete the proof. $\square$

Lemma D.1 provides one possible case for appropriate $\alpha$. In fact, the selection of $\alpha$ can be quite flexible; for instance, similarly, when $\delta = \max\{|\alpha_1|, |\alpha_2|, \ldots, |\alpha_r|\}$ and $|\alpha| > \delta$ holds, $\boldsymbol{A} + \alpha \boldsymbol{I}_n$ also remains full-rank. Our experimental results related to regularized models will further illustrate the effectiveness of an appropriate $\alpha$ in enhancing model performance.

We also acknowledge that our modification is relatively straightforward and may not be optimal. However, we believe that it may be a good choice to make structural improvements to the model from the perspective of the loss function, or more generally, from an optimization standpoint. For example, to address the issue of non-normalized $\boldsymbol{y}_{std}^{(i)}$ and $\hat{\boldsymbol{y}}^{(i)}$, we can also modify the loss function from the perspective of ridge regression as:

$$\mathcal{L} = \frac{1}{2\eta D} \sum_{i=1}^{N} \|\boldsymbol{W}_V \boldsymbol{x}_i - \boldsymbol{W}\phi(\boldsymbol{W}_K \boldsymbol{x}_i)\|_F^2 + \frac{\alpha}{2\eta}\|\boldsymbol{W}\|_F^2.$$

And the optimal $\boldsymbol{W}^*$ will be

$$\boldsymbol{W}^* = \left[\phi(\boldsymbol{W}_K \boldsymbol{X})\phi(\boldsymbol{W}_K \boldsymbol{X})^T + \alpha D \boldsymbol{I}\right]^{-1} \boldsymbol{W}_V \boldsymbol{X}\phi(\boldsymbol{W}_K \boldsymbol{X}).$$

Correspondingly, the attention mechanism will be modified to

$$\boldsymbol{H} = \boldsymbol{W}^*\phi(\boldsymbol{W}_Q \boldsymbol{X}) = \left[\phi(\boldsymbol{W}_K \boldsymbol{X})\phi(\boldsymbol{W}_K \boldsymbol{X})^T + \alpha D \boldsymbol{I}\right]^{-1} \boldsymbol{W}_V \boldsymbol{X}\phi(\boldsymbol{W}_K \boldsymbol{X})\phi(\boldsymbol{W}_Q \boldsymbol{X}), \quad (33)$$

where we neglect the normalization operation. This result is very similar to the mesa-layer proposed by Von Oswald et al. [2023b], which optimizes linear attention layers under the auto-regressive setting. Here, we presented its form on softmax self-attention setting using kernel methods and explained it from the perspectives of contrastive loss and ridge regression. Although the matrix inversion calculation in Eq (33) can be computationally expensive, effective methods for computing Eq (33), including both forward computation and backward propagation, have been thoroughly researched in Von Oswald et al. [2023b], which contributes to making the above modification practically applicable.

## D.2 More Discussion on the Data Augmentation

In addition to discussing the loss function, the contrastive learning paradigm also offers our some insights. In the corresponding representation learning process of ICL, we can easily notice that "data augmentation" is performed using a simple linear mapping, which may be not sufficient for learning deeper-level features. To address this, we can employ more complicated nonlinear functions for more complex augmentations. Denoting these two augmentations as $g_1$ and $g_2$, consequently, the process of contrastive learning will be modified as follows

$$\mathcal{L} = -\frac{1}{\eta D} \sum_{i=1}^{N} [g_1(\boldsymbol{W}_V \boldsymbol{x}_i)]^T \boldsymbol{W} \phi(g_2(\boldsymbol{W}_K \boldsymbol{x}_i)).$$

Correspondingly, the gradient update for $\boldsymbol{W}$ will become

$$\boldsymbol{W}^{(t)} = \boldsymbol{W}^{(t-1)} - \eta \frac{\partial \mathcal{L}}{\partial \boldsymbol{W}} = \boldsymbol{W}^{(t-1)} + \sum_{i=1}^{N} D^{-1} g_1(\boldsymbol{W}_V \boldsymbol{x}_i) \otimes \phi(g_2(\boldsymbol{W}_K \boldsymbol{x}_i)).$$

And from the perspective of ICL, correspondingly, the last token will be updated as

$$\boldsymbol{h}'_{T+1} = \boldsymbol{W}_0 \phi(\boldsymbol{q}) + D^{-1} \left[ \sum_{i=1}^{N} g_1(\boldsymbol{V}_D^{(i)}) \otimes \phi(g_2(\boldsymbol{K}_D^{(i)})) \right] \phi(\boldsymbol{q}).$$

And by reformulating the above equation we will get Eq (14) in the main body.

Correspondingly, the modification for self-attention layer can be adjusted as,

$$\boldsymbol{H} = g_1(\boldsymbol{W}_V \boldsymbol{X}) \text{softmax} \left( \frac{g_2(\boldsymbol{W}_K \boldsymbol{X})^T \boldsymbol{W}_Q \boldsymbol{X}}{\sqrt{d_o}} \right),$$

where $g_1(\cdot)$ and $g_2(\cdot)$ will be column-wise here. It is worth noting that here we have only presented the framework of using nonlinear functions as data augmentations to modify the self-attention layer and in the simplest case, we can set $g_1(x)$ and $g_2(x)$ as MLPs (Multi-Layer Perceptrons). However, in practice, it is encouraged to use data augmentation functions that are tailored to specific data structures. For example, in the case of CMT [Guo et al., 2022], the used Convolutional Neural Networks (CNNs) can be considered as a form of "strong data augmentations" suitable for image datas within our framework. We consider the exploration of various augmentation methods tailored to different types of data as an open question for future research.

## D.3 More discussion on the Negative Samples

Although the gradient descent process corresponding to ICL exhibits some similarities with traditional contrastive learning approaches without negative samples, there are also significant differences: In traditional Siamese networks, the augmented representations as positive pairs are further learned through target and online network that share weights (or at least influence each other using EMA). The output of the target network is then passed through a predictor to compute the contrastive loss. In contrast, the representation learning pattern corresponding to ICL indeed performs more simply, which may potentially limit the ability of the dual model to learn representations fully without negative samples. To address this, similar to most contrastive learning approaches, we can introduce negative samples forcing the model to separate the distances between positive and negative samples at the same time, that is,

$$\mathcal{L} = -\frac{1}{\eta D} \sum_{i=1}^{N} (\boldsymbol{W}_V \boldsymbol{x}_i)^T \boldsymbol{W} \phi(\boldsymbol{W}_K \boldsymbol{x}_i) + \frac{\beta}{\eta D} \sum_{i=1}^{N} \frac{1}{|\mathcal{N}(i)|} \sum_{j \in \mathcal{N}(i)} (\boldsymbol{W}_V \boldsymbol{x}_j)^T \boldsymbol{W} \phi(\boldsymbol{W}_K \boldsymbol{x}_i)$$

$$= -\frac{1}{\eta D} \sum_{i=1}^{N} \left( \boldsymbol{W}_V \left( \boldsymbol{x}_i - \frac{\beta}{|\mathcal{N}(i)|} \sum_{j \in \mathcal{N}(i)} \boldsymbol{x}_j \right) \right)^T \boldsymbol{W} \phi(\boldsymbol{W}_K \boldsymbol{x}_i)$$

$$= -\frac{1}{\eta D} \sum_{i=1}^{N} (\boldsymbol{W}_V \tilde{\boldsymbol{x}}_i)^T \boldsymbol{W} \phi(\boldsymbol{W}_K \boldsymbol{x}_i),$$

where $\tilde{\boldsymbol{x}}_i = \boldsymbol{x}_i - \frac{\beta}{|\mathcal{N}(i)|} \sum_{j \in \mathcal{N}(i)} \boldsymbol{x}_j$, $\mathcal{N}(i)$ is the set of the negative samples for $\boldsymbol{x}_i$ and $\beta$ is a hyperparameter. As a result, the gradient descent on $\boldsymbol{W}$ will be modified as

$$\boldsymbol{W}^{(t)} = \boldsymbol{W}^{(t-1)} - \eta \frac{\partial \mathcal{L}}{\partial \boldsymbol{W}} = \boldsymbol{W}^{(t-1)} + \sum_{i=1}^{N} D^{-1} \boldsymbol{W}_V \tilde{\boldsymbol{x}}_i \otimes \phi(\boldsymbol{W}_K \boldsymbol{x}_i).$$

Correspondingly, the ICL process for $\hat{\boldsymbol{h}}_{N+1}$ will be

$$\boldsymbol{h}'_{T+1} = \boldsymbol{W}_0 \phi(\boldsymbol{W}_Q \boldsymbol{x}'_{T+1}) + D^{-1} \left[ \sum_{i=1}^{N} \boldsymbol{W}_V \tilde{\boldsymbol{x}}_i \otimes \phi(\boldsymbol{W}_K \boldsymbol{x}_i) \right] \phi(\boldsymbol{W}_Q \boldsymbol{x}'_{T+1}).$$

And this will directly result in Eq (15) in the main body.

As for a self-attention layer, similarly, we can get the corresponding modification as

$$\boldsymbol{H} = \boldsymbol{W}_V \tilde{\boldsymbol{X}} \mathrm{softmax} \left( \frac{(\boldsymbol{W}_K \boldsymbol{X})^T \boldsymbol{W}_Q \boldsymbol{X}}{\sqrt{d_o}} \right), \tag{34}$$

where $\tilde{\boldsymbol{X}}^{(i)} = \tilde{\boldsymbol{x}}_i$. In corresponding experiments, for each token, we simply choose other the $k$ least relevant tokens as its negative samples, i.e., the $k$ tokens with the lowest attention scores. Noting that here we simply use other token representations as negative samples for $\boldsymbol{x}_i$. However, there are more ways to construct negative samples that are worth exploring (for instance, using noise vectors or tokens with low semantic similarity as negative samples). For specific data structures and application scenarios, customizing the selection or construction of negative samples may be more effective.

## E   More Experiments

### E.1   More details of Experiments on Linear Task

In this part, we will discuss our experimental setup in more details and provide more results on linear regression task.

Inspired by Garg et al. [2022] and Von Oswald et al. [2023a], we choose to pretrain a softmax attention layer before exploring the equivalence proposed by Theorem 3.1. In fact, pretraining is not mandatory since our theoretical analysis does not depend on any specific weight construction. In other words, the inference results of ICL and the test prediction of the dual model will still remain consistent for an attention layer with arbitrary weights or even random initialization. However, for the convenience of further investigating the impact of subsequent modifications to the model structure and to better align with real-world scenarios, we still opted for pretraining to let the model acquire some task-specific knowledge. Additionally, our experiments are conducted in a self-attention setting. When we focus only on the last token, this is equivalent to considering the case with only one query token ($T = 0$) in Section 2.1. The experiments are completed on a single 24GB NVIDIA GeForce RTX 3090 and the experiments can be completed within one day.

For the linear regression task, we generate the task by $\boldsymbol{s} = \boldsymbol{W}\boldsymbol{t}$ where every element of $\boldsymbol{W} \in \mathbb{R}^{d_s \times d_t}$ is sampled from a normal distribution $\boldsymbol{W}_{ij} \sim \mathcal{N}(0, 1)$ and $\boldsymbol{t}$ is sampled from a Gaussian distribution $\boldsymbol{x} \sim U(-1, 1)^{d_t}$. To facilitate more accurate estimation of attention matrices using random features and considering the limited learning capacity of a single attention layer, we only set a small value for $d_t = 11$ and $d_s = 1$. Then, at each step, we use generated $\{\boldsymbol{x}_i = [\boldsymbol{t}_i; s_i]\}_{i=1}^{N+1}$ to form the input matrix $\boldsymbol{X}$ while the label part of the query token is masked to be zero, that is, $\boldsymbol{x}_{N+1} = [\boldsymbol{t}_i; 0]$ where we consider only one query token and we denote $\boldsymbol{x}'_{T+1} = \boldsymbol{x}_{N+1}$ to maintain consistency of notation in Section 2.1. The softmax attention layer is expected to predict $\hat{s}_{N+1}$ to approximate the ground truth value $\boldsymbol{s}_{N+1}$. We use mean square error (MSE) as the loss function, that is, for each epoch,

$$\mathcal{L} = \frac{1}{N_{step}} \sum_{j=1}^{N_{step}} \|\hat{\boldsymbol{s}}_{N+1}^{(j)} - \boldsymbol{s}_{N+1}^{(j)}\|^2,$$

where $\hat{\boldsymbol{s}}_{N+1}^{(j)}$ and $\boldsymbol{s}_{N+1}^{(j)}$ are the prediction and ground truth value at $j$-th step and $N_{step}$ is the number of steps. We set $N_{step} = 1024$ for $N + 1 = 16$ which means the total number of tokens remains

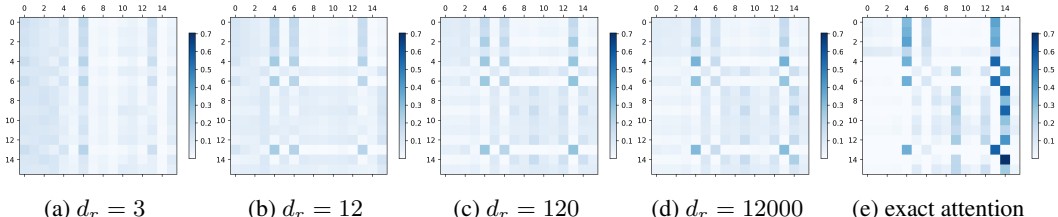

(a) $d_r = 3$      (b) $d_r = 12$      (c) $d_r = 120$      (d) $d_r = 12000$      (e) exact attention

Figure 6: The estimation of the attention matrix by positive random features when varying $d_r$

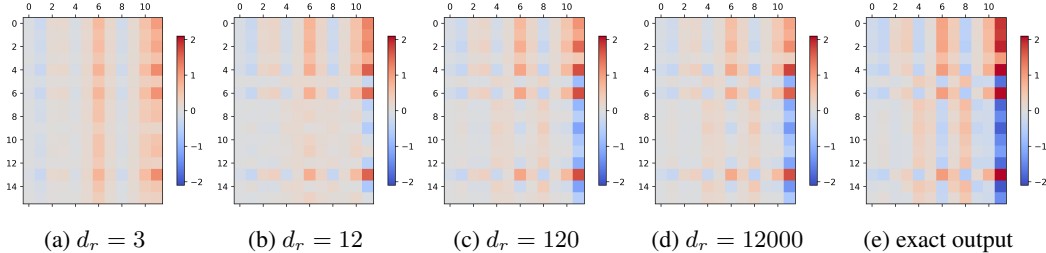

(a) $d_r = 3$      (b) $d_r = 12$      (c) $d_r = 120$      (d) $d_r = 12000$      (e) exact output

Figure 7: The estimation of the output matrix by positive random features when varying $d_r$

16384. We choose stochastic gradient descent (SGD) [Amari, 1993] as the optimizer and we set the learning rate to 0.003 for normal and regularized models, while the remaining experiments to 0.005. We also attempt the multi-task scenario, where the input token at each step is generated from a different task. However, we find it challenging for a single attention layer to effectively learn in this setting, resulting in disordered predictions. Therefore, our experiments are currently limited to single-task settings, and the multi-task scenario is worth further investigation in the future.

It is worth noting that we approximate the attention matrix calculation using random features as kernel mapping function instead of using the traditional softmax function in the self-attention layer [Choromanski et al., 2020]. The mapping function $\phi : \mathbb{R}^{d_o} \to \mathbb{R}^{d_r}$ has the form of $\phi(\boldsymbol{x}) = e^{\boldsymbol{w}^T \boldsymbol{x} - \|\boldsymbol{x}\|^2/2}$ where $\boldsymbol{w} \sim \mathcal{N}(0, I)$. Orthogonal random features [Yu et al., 2016, Choromanski et al., 2020] or simplex random features [Reid et al., 2023] can be chosen to achieve better performance theoretically. We investigate the impact of changing the dimension of random features $d_r$ on the approximation of attention matrices and output, using Mean Squared Error (MSE) and Mean Absolute Error (MAE) as evaluation metrics, where we conduct 50 repeated experiments and calculated the average values for each value of $d_r$, as shown in Figure 8. It can be observed that as the dimension of random features increases, the approximation performance gradually improves, with both errors reaching a low level in the end. We visualize the exact attention matrix and compare it with the estimated attention matrices obtained using different values of $d_r$, as shown in Figure 6. Again, it can be seen that as $d_r$ increases, the approximation of the true attention matrix improves gradually and similar results can be observed for the analysis of output matrices in Figure 7.

To obtain a more accurate estimation of the attention matrix, we set the output dimension of the mapping function to be 100 times the input dimension, that is, $d_r = 100(d_s + d_t) = 1200$. Furthermore, we visualize the exact attention matrix and the output with the approximation results, which are shown in the Figure 9. As we can see, although some larger values are not estimated accurately due to the limited dimension of the random features we select, the majority of the information is still estimated comprehensively well. These findings indicate that our choice of using positive random features as mapping functions to estimate the true softmax attention and conduct experiments is relatively feasible.

After the weights $\boldsymbol{W}_Q, \boldsymbol{W}_K, \boldsymbol{W}_V$ of the attention layer have been determined, we generate test $N + 1$ tokens in the same way where the $s$ part of the $(N + 1)$-th token is also set to be zero and finally input the test tokens into the attention layer to obtain the corresponding predicted $\hat{\boldsymbol{h}}_{N+1} = [\hat{\boldsymbol{t}}_{N+1}^{(1)}, \hat{s}_{N+1}^{(1)}]$. Here, we also use $\boldsymbol{h}'_{T+1} = \hat{\boldsymbol{h}}_{N+1}$ to maintain the notation consistency in Section 2.1.

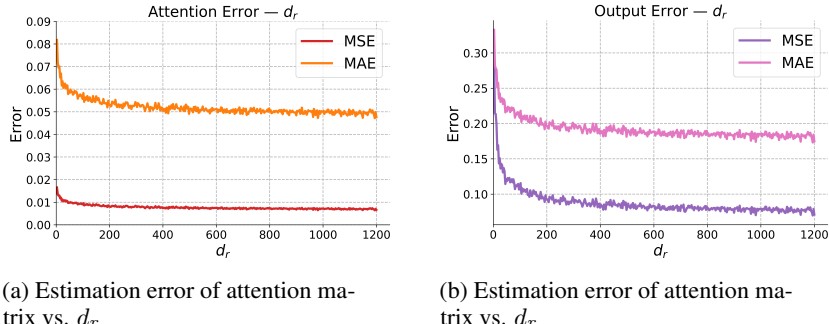

(a) Estimation error of attention matrix vs. $d_r$

(b) Estimation error of attention matrix vs. $d_r$

Figure 8: The error of positive random features in estimating the attention and output matrices as $d_r$ varies.

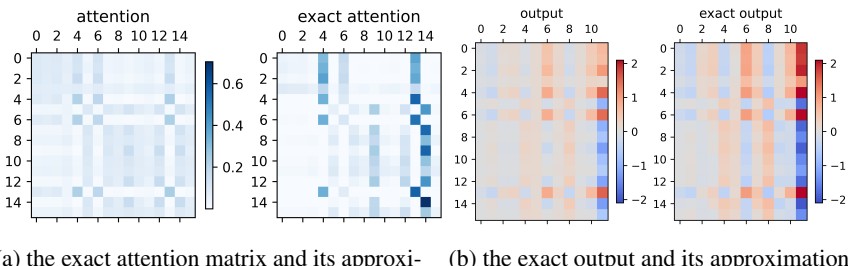

(a) the exact attention matrix and its approximation

(b) the exact output and its approximation

Figure 9: The comparison between the exact attention matrix, output and their estimated approximations using random features under setting $N = 16$ and $d_r = 1200$.

On the other hand, we construct a dual model $f(\boldsymbol{x}) = \boldsymbol{W}\phi(\boldsymbol{x})$ where $\phi(\cdot)$ is strictly equivalent to the kernel mapping function used in the attention layer. We transform the first $N$ tokens as the training set according to Theorem 3.1 and train the dual model using the loss formed by Eq (9). In fact, according to Theorem 3.1, after we perform one step of gradient descent on this training set, the test prediction $\hat{\boldsymbol{y}}_{test} = [\hat{\boldsymbol{t}}_{N+1}^{(2)}, \hat{s}_{N+1}^{(2)}]$ of the dual model will strictly equal $\boldsymbol{h}'_{T+1}$.

We conduct experiments under the same setup using different random seeds to explore the effects of various model modifications. The data for all three experiments are generated under identical conditions. One set of experimental results is presented in the main text, while the results of the other two sets are shown in the Figure 10. Similar to the discussion in the main text, we can achieve better performance than the normal model with appropriate parameter settings.

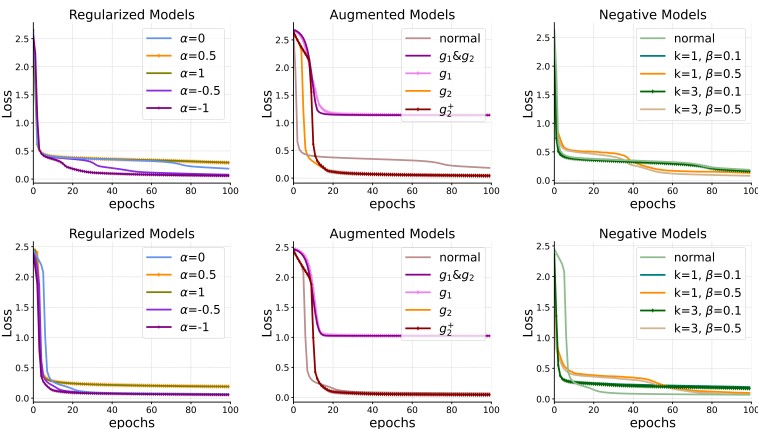

Figure 10: The performance for regularized models (Center Left), augmented models (Center Right) and negative models (Right) with different settings for different random seeds.

## E.2  More details of Experiments on Different Tasks

In addition to conducting experiments on linear regression tasks, we also extended our experiments to involve trigonometric and exponential tasks.

### E.2.1  More details of Experiments on Trigonometric Tasks

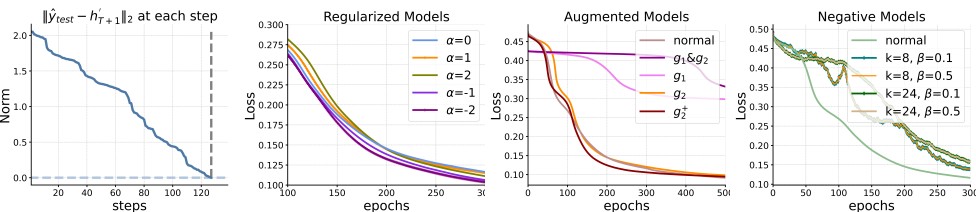

Figure 11: The equivalence between ICL of one softmax attention layer and gradient descent, along with analysis on different model modifications for trigonometric tasks. **Left Part:** $\|\hat{\boldsymbol{y}}_{test} - \boldsymbol{h}'_{T+1}\|_2$ as the gradient descent proceeds under setting $N = 127$; **Remaining Part:** the performance for regularized models (Center Left), augmented models (Center Right) and negative models (Right) with different settings.

For trigonometric task, we generate the task by $\boldsymbol{s} = \cos(\boldsymbol{W}\boldsymbol{t})$ where $\cos(\cdot)$ is element-wise, $\boldsymbol{W} \in \mathbb{R}^{d_s \times d_t}$ is sampled from the normal distribution $\boldsymbol{W}_{ij} \sim \mathcal{N}(0, 1)$ while $\boldsymbol{t}$ is sampled from the uniform distribution $\boldsymbol{x} \sim U(0, \pi)^{d_t}$. In experiments, we found that for one softmax attention layer, learning higher-dimensional tasks is challenging. Therefore, we only set $d_t = 7$ and $d_s = 1$. At each step, we use $N + 1 = 128$ tokens $\{\boldsymbol{x}_i = [\boldsymbol{t}_i; \boldsymbol{s}_i]\}_{i=1}^{N+1}$ and the total number of tokens remains unchanged at 16384. Compared to the setting $N + 1 = 16$ of linear tasks, we observed that for more complex tasks, the attention layer needs to use more tokens to provide information at each training step. Similarly, we mask the label part of the last token, that is, $\boldsymbol{s}_{N+1} = \boldsymbol{0}$ and use mean square error (MSE) loss to train the attention layer. We choose SGD as the optimizer and the learning rate is set as 0.005. The rest of the settings remain consistent with those used in the linear task. The result for trigonometric regression task is shown in Figure 11.

Firstly, as shown in the left part of Figure 11, the inference results is of ICL is strictly equivalent to the prediction of the dual model, that is, $\hat{\boldsymbol{h}}_{N+1} = \hat{\boldsymbol{y}}_{test}$ as well as the label part $\hat{s}_{N+1}^{(1)} = \hat{s}_{N+1}^{(2)}$, aligning with our analysis in Theorem 3.1.

The performance of modified model during training process can be seen in the remaining parts of Figure 11. For regularized models, as seen in the center left part of figure 11, the models when $\alpha < 0$ converge slightly faster and reach better final results compared to the normal model ($\alpha = 0$). For augmented models, we use as the same augmentation functions $g_1$ and $g_2$ as the ones in the linear regression task, that is, $g_1(\boldsymbol{x}) = g_2(\boldsymbol{x}) = \sigma(\boldsymbol{W}\boldsymbol{x})$ where $\sigma(\cdot)$ is GELU activation function. However, for $g_2^+$, we use ELU as the activation function. We can find from the center right part of Figure 11 that, compared to the normal model, using $g_1$ alone and using $g_1$ and $g_2$ simultaneously as data augmentations significantly degrade the model's performance, including convergence speed and final results. However, using $g_2$ alone yields comparable result with the normal model. Particularly, when using $g_2^+$, the model accelerates its convergence speed. However, for negative models, the performance with the selected number of negative samples $k$ and the parameter $\beta$ is worse than the normal model, which suggests that our simple approach of selecting those tokens low attention scores as negative samples is not a reasonable method. Just as we discussed in Section 4, for different tasks, a more refined strategy for selecting negative samples should be considered.

### E.2.2  More details of Experiments on Exponential Tasks

For exponential task, we generate the task by $\boldsymbol{s} = \exp(\boldsymbol{W}\boldsymbol{t})$ where $\exp(\cdot)$ is also element-wise, $\boldsymbol{W} \in \mathbb{R}^{d_s \times d_t}$ is sampled from the normal distribution $\boldsymbol{W}_{ij} \sim \mathcal{N}(0, 1)$ while $\boldsymbol{t}$ is sampled from the uniform distribution $\boldsymbol{x} \sim U(-1, 1)^{d_t}$. We only set $d_t = 6$ and $d_s = 1$ considering the limited learning capacity of one softmax attention layer. At each training step, we use $N + 1 = 512$ tokens $\{\boldsymbol{x}_i = [\boldsymbol{t}_i; \boldsymbol{s}_i]\}_{i=1}^{N+1}$ and the total number of tokens remains unchanged at 16384. Compared to the

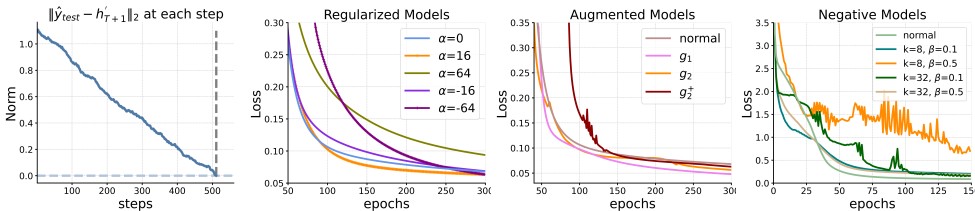

Figure 12: The equivalence between ICL of one softmax attention layer and gradient descent, along with analysis on different model modifications for exponential tasks. **Left Part:** $\|\hat{\boldsymbol{y}}_{test} - \boldsymbol{h}'_{T+1}\|_2$ as the gradient descent proceeds under setting $N = 511$; **Remaining Part:** the performance for regularized models (Center Left), augmented models (Center Right) and negative models (Right) with different settings.

setting $N + 1 = 16$ of linear tasks and $N + 1 = 128$ of trigonometric tasks, we also find that for exponential tasks, the attention layer needs more tokens to provide in-context information at each training step. The rest of the settings remain consistent with those used in the trigonometric task. The result for exponential regression task is shown in Figure 12.

Similarly, as shown in the left part of Figure 12, the result $\hat{\boldsymbol{h}}_{N+1}$ of ICL inference is equivalent to the test prediction $\hat{\boldsymbol{y}}_{test}$ of the dual model after training, just as stated in Theorem 3.1. For regularized models, it can be observed that when $\alpha = 16$, the model converges faster and achieves better result. For augmented models, using $g_1$ or $g_2$ alone as data augmentations results in better performance. However, when both $g_1$ and $g_2$ are used simultaneously, the training process becomes unstable, so we did not show it in the center right part of Figure 12. For negative model, similar to the case in the trigonometric task, the different combinations of negative samples' number $k$ and parameter $\beta$ do not show a significant improvement over the normal model, highlighting the importance of the strategy for selecting negative samples. We leave the exploration of a more refined negative sample selection strategy when facing various tasks for future consideration.

### E.3 More Experiments on Combinations

In addition, we also conduct experiments with their combinations on linear tasks, trigonometric tasks , and exponential tasks. The results are shown in Figure 13. For linear tasks, a combination of regularized and augmented modifications is sufficient. However, for the other two tasks, the results are actually worse than using regularized or augmented modification individually (compared to Figures 11 and 12). We think this may be due to the ineffective selection of negative samples, which is amplified when combined. Therefore, when the design of augmentation or negative sample improvement methods is not effective, we recommend using a single modification method.

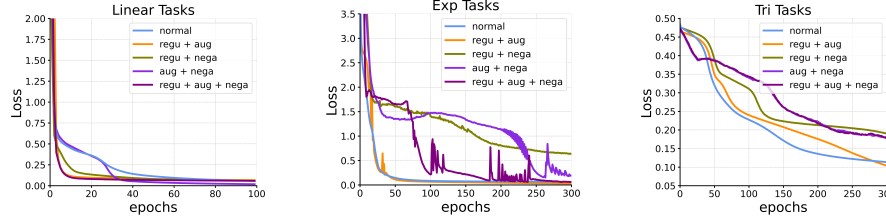

Figure 13: Performance of different combinations on linear (Left), exponential (Center), and trigonometric (Right) tasks.

### E.4 More Experiments on One Transformer Layer

Similar to the experiments with one softmax attention layer, we also conduct experiments on a Transformer layer (introducing one FFN layer after the attention layer) and trained its dual model based on Theorem B.1. As shown in Figure 14, the inference result $\hat{\boldsymbol{h}}_{N+1}$ of ICL remains equivalent to the test prediction $\hat{\boldsymbol{y}}_{test}$ of the trained dual model. Furthermore, to validate the potential low-rank

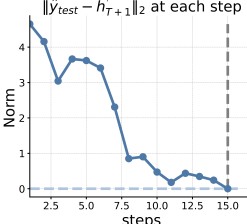
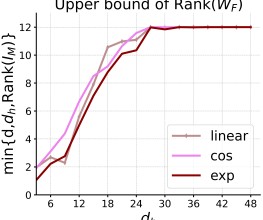

Figure 14: The equivalence between ICL of one Transformer layer and gradient descent, along with analysis on upper bound of Rank($\boldsymbol{W}_F$). **Left:** $\|\hat{\boldsymbol{y}}_{test} - \boldsymbol{h}'_{T+1}\|_2$ as the gradient descent proceeds under setting $N = 15$; **Right:** the upper bound of Rank($\boldsymbol{W}_F$) when setting $d = 12$ and varying $d_h$.

property of matrix $\boldsymbol{W}_F$, we explore its upper bound of rank. Noting that $\boldsymbol{W}_F = \boldsymbol{W}_2 \boldsymbol{I}_M \boldsymbol{W}_1$ where $\boldsymbol{W}_1 \in \mathbb{R}^{d_h \times d}$, $\boldsymbol{W}_2 \in \mathbb{R}^{d \times d_h}$, $\boldsymbol{I}_M \in \mathbb{R}^{d_h \times d_h}$, the upper bound of Rank($\boldsymbol{W}_F$) is

$$\text{Rank}(\boldsymbol{W}_F) \leq \min\{d, d_h, \text{Rank}(\boldsymbol{I}_M)\},$$

where $\text{Rank}(\boldsymbol{I}_M)$ is equivalent to the number of non-zero elements in $\boldsymbol{I}_M$. We fix $d = 12$ while varying the values of $d_h$. We generate 1024 sets of $\boldsymbol{X}_{test}$ for different tasks and repeat the experiments 5 times. Finally, we calculate the average upper bound of the rank of $\boldsymbol{W}_F$. The results are shown in the right part of Figure 14, indicating that when $d_h \geq 2.75d = 33$, the upper bound remains stable and equals $d = 12$. Otherwise, when $d_h$ is set to a smaller value, $\boldsymbol{W}_F$ exhibits clear low-rank property.

### E.5 More Experiments on More Realistic NLP Tasks

We supplement our experiments on more more realistic NLP tasks. We choose the BERT-base-uncased model (can be downloaded from Huggingface library[Wolf, 2019], hereafter referred to as BERT[Kenton and Toutanova, 2019]) to validate the effectiveness of modifications to the attention mechanism and select four relatively smaller GLUE datasets (CoLA, MRPC, STS-B, RTE) [Wang, 2018]. We load the checkpoint of the pre-trained BERT model, where 'classifier.bias' and 'classifier.weight' are newly initialized, and then we fine-tune the model to explore the performance of three attention modifications as well as their combinations. In terms of more detailed experiment settings, we set the batch size to 32, the learning rate to 2e-5, and the number of epochs to 5 for all datasets. All experiments are conducted on a single 24GB NVIDIA GeForce RTX 3090. All experimental results are presented in Table 1. Below, we discuss the various modifications and their performance.

**For the regularized modification**, we consider different values of $\alpha$, specifically selected from $\{-0.5, -0.1, 0.1, 0.5\}$. As can be observed in Table 1, except for RTE, the best regularized models outperform the original model on the other three datasets. However, we also note that when the absolute value of $\alpha$ is too large, the model's performance declines significantly, so we recommend using smaller absolute values for $\alpha$.

**For the augmented modification**, we also consider applying more complex "augmentation" functions to the linear key/value mappings. However, unlike the previous methods used in simulation tasks, we do not simply select $g_1$ and $g_2$ as MLPs, i.e., $g_1(\boldsymbol{W}_V\boldsymbol{x}) = \boldsymbol{W}_2\sigma(\boldsymbol{W}_1\boldsymbol{W}_V\boldsymbol{x})$. This design is avoided because it could undermine the effort made during pre-training to learn the weights $\boldsymbol{W}_V$ and $\boldsymbol{W}_K$, leading to difficulties in training and challenges in comparison. Instead, we adopt a parallel approach, i.e., $g_1(\boldsymbol{W}_V x) = \boldsymbol{W}_V x + c\boldsymbol{W}_2\sigma(\boldsymbol{W}_1 x)$, where $c$ is a hyperparameter to control the influence of the new branch, $\sigma$ is the GELU activation function and the hidden layer dimension is set to twice the original size of $\boldsymbol{W}_V x$. $g_2(\boldsymbol{W}_K x) = \boldsymbol{W}_K x + c\boldsymbol{W}_2\sigma(\boldsymbol{W}_1 x)$ follows the same format.

Experimental results show that the best augmented models achieve better performance than the original model across all four datasets. Notably, augmentation on the value mapping (i.e., using $g_1$ alone) proves to be more effective than other methods, both in terms of performance and the amount of additional parameters introduced. Using both $g_1$ and $g_2$ introduces more parameters, which is particularly undesirable for larger models. Thus, under the augmentation methods and experimental settings we selected, using $g_1$ alone is recommended.

In addition, we do not rule out the possibility of more powerful and efficient augmentation methods. Our choice of $g_1$ and $g_2$ as parallel MLPs is primarily motivated by the desire to make better use of

| Model Types | Dataset | CoLA | MRPC | STS-B | RTE |
|---|---|---|---|---|---|
| Normal | Bert-base-uncased | 56.82 | 90.24/86.27 | 88.29/87.96 | 68.23 |
| Regularized Models | $\alpha = -1.0$ | 0.0 | 79.01/68.87 | 57.23/60.16 | 52.71 |
| | $\alpha = -0.5$ | 61.42 | 83.17/74.02 | 85.28/85.22 | 57.04 |
| | $\alpha = -0.1$ | 58.06 | 89.50/85.05 | 88.71/88.27 | 65.70 |
| | $\alpha = 0.1$ | 58.34 | 90.59/86.76 | 88.12/87.81 | 64.98 |
| | $\alpha = 0.5$ | 27.01 | 83.56/73.28 | 85.25/85.03 | 59.93 |
| | $\alpha = 1.0$ | 0.0 | 81.22/68.38 | 52.07/55.60 | 47.29 |
| | Local Best | **61.42** | **90.59/86.76** | **88.71/88.27** | 65.70 |
| Augmented Models | $g_1$ / $c = 0.2$ | 59.85 | 88.11/83.33 | 88.56/88.22 | 68.59 |
| | $g_1$ / $c = 1$ | 56.51 | 90.88/87.01 | 88.96/88.60 | 71.12 |
| | $g_2$ / $c = 0.2$ | 56.29 | 87.65/82.60 | 88.60/88.24 | 68.59 |
| | $g_2$ / $c = 1$ | 58.85 | 87.74/82.60 | 88.68/88.32 | 70.40 |
| | $g_1$ & $g_2$ / $c = 0.2$ | 57.32 | 89.62/85.29 | 88.48/88.19 | 71.12 |
| | $g_1$ & $g_2$ / $c = 1$ | 58.30 | 90.40/86.52 | 88.83/88.45 | 68.95 |
| | Local Best | **59.85** | **90.88/87.01** | **88.96/88.60** | **71.12** |
| Negative Models | $r = 0.1$ / $\beta = 0.1$ | 56.22 | 88.54/83.82 | 88.25/87.91 | 65.34 |
| | $r = 0.2$ / $\beta = 0.1$ | 57.92 | 90.00/85.78 | 88.22/87.84 | 66.06 |
| | $r = 0.3$ / $\beta = 0.1$ | 57.92 | 89.31/84.80 | 88.26/87.90 | 67.15 |
| | $r = 0.1$ / $\beta = 0.2$ | 58.92 | 87.90/83.33 | 88.34/88.11 | 63.54 |
| | $r = 0.2$ / $\beta = 0.2$ | 57.13 | 87.87/83.09 | 88.59/88.27 | 64.98 |
| | $r = 0.3$ / $\beta = 0.2$ | 58.14 | 88.97/84.56 | 88.64/88.33 | 66.79 |
| | Local Best | **58.92** | **90.00/85.78** | **88.64/88.33** | 67.15 |
| Combined Models | Reg & Aug | 56.56 | 88.54/83.82 | 88.86/88.60 | 68.59 |
| | Reg & Neg | 58.11 | 88.19/83.33 | 88.41/88.17 | 69.31 |
| | Aug & Neg | 59.07 | 90.49/86.76 | 88.59/88.21 | 70.76 |
| | Reg & Aug & Neg | 58.92 | 88.39/83.58 | 88.32/88.01 | 67.87 |
| | Local Best | **59.07** | **90.49/86.76** | **88.86/88.60** | **70.76** |
| | Global Best | **61.42** | **90.88/87.01** | **88.96/88.60** | **71.12** |

Table 1: Partial GLUE test results of different modifications. "Local Best" is used to display the best results for each modification type, where bolded results indicate the performance superior to the original model. "Global Best" is used to showcase the best results among all modifications. Matthews correlation, F1 scores/accuracy, Pearson/Spearman correlation, accuracy are reported for CoLA, MRPC, STS-B, RTE respectively.

the pre-trained weights $W_K$ and $W_V$. We have also noticed that this specific augmentation function design is structurally similar to the Parallel Adapter [He et al., 2021]. However, we would like to emphasize that our parallel design is just a specific case within this broader augmented modification framework and this is a new perspective for understanding the Parallel Adapter. As for practical implementation, the Parallel Adapter method focuses more on efficient training, so it uses fewer parameters, and the original $W_V$ and $W_K$ are freezed—only the newly introduced parameters are trained. In contrast, our approach aims to validate the benefits of introducing stronger nonlinear augmentation functions into the linear value/key mappings. Therefore, we set a higher hidden layer dimension (twice that of $W_V x$ or $W_K x$) and also train $W_V$ and $W_K$ simultaneously. This design is relatively general and does not take into account the specific characteristics of individual tasks. We still encourage the development of more task-specific augmentation strategies tailored to different tasks.

**For the negative modification**, we continue to select tokens with lower attention scores as negative samples. The parameter $r$ represents the proportion of tokens used as negative samples, while $\beta$ indicates the overall reduction in attention scores. We choose $r$ from $\{0.1, 0.2, 0.3\}$ and $\beta$ from $\{0.1, 0.2\}$. Under these combinations, the best negative models only outperform the original model

on CoLA and STS-B, whereas their performance on MRPC and RTE is worse than the original one. This suggests that our simple approach of considering tokens with low attention scores as negative samples might be too coarse. A more effective method for constructing negative samples should be designed, which is a direction worth exploring in the future.

We also consider **combining different modification methods**. Specifically, we choose $\alpha = 0.1$, $g_1/c = 1$ and $r = 0.2/\beta = 0.1$ respectively as the basis for combining the three types of modifications, considering their overall performance across all datasets. The results indicate that under our settings, the combination of augmented and negative modification achieves the best performance on CoLA, MRPC, and RTE, while the combination of regularized and augmented modification achieves the best performance on STS-B. However, their optimal performance is slightly inferior to the best performance achieved with augmented models alone. Therefore, we conclude that using all three modifications simultaneously is not necessary. With appropriate hyperparameter choices, using augmented modification alone or in combination with one other modification is sufficient.

Overall, the experimental results show that our modifications inspired by the representation learning process are helpful in enhancing performance. This further validates the potential of our approach of thinking about and improving the attention mechanism from a representation learning perspective. In addition, we would like to reiterate that more validation across additional tasks and models, and the development of task-specific augmentation and negative sampling methods are all interesting directions worth exploring in the future.

# F More Details about Related Work

In this section, we provide additional details about the related work in Section 6, especially those that involve formalization. Dai et al. [2022] interpret ICL as implicit fine-tuning: More specifically, let $\boldsymbol{X} = [\boldsymbol{X}_D, \boldsymbol{X}_T]$ where $\boldsymbol{X}_D = [\boldsymbol{x}_1, \boldsymbol{x}_2, \ldots, \boldsymbol{x}_N]$ denotes the demonstration tokens and $\boldsymbol{X}_T = [\boldsymbol{x}'_1, \boldsymbol{x}'_2, \ldots, \boldsymbol{x}'_T]$ be query tokens. On the one hand, for ICL, they consider the output of $\boldsymbol{q} = \boldsymbol{W}_Q \boldsymbol{x}'_{T+1}$ under the linear attention setting as

$$
\begin{aligned}
\tilde{F}_{\text{ICL}}(\boldsymbol{q}) &= \boldsymbol{W}_V[\boldsymbol{X}_D, \boldsymbol{X}_T](\boldsymbol{W}_K[\boldsymbol{X}_D; \boldsymbol{X}_T])^T \boldsymbol{q} \\
&= \boldsymbol{W}_V \boldsymbol{X}_T(\boldsymbol{W}_K \boldsymbol{X}_T)^T q + \boldsymbol{W}_V \boldsymbol{X}_D(\boldsymbol{W}_K \boldsymbol{X}_D)^T \boldsymbol{q} \\
&= \boldsymbol{W}_{\text{ZSL}} \boldsymbol{q} + \text{LinearAtten}(\boldsymbol{W}_V \boldsymbol{X}_D, \boldsymbol{W}_K \boldsymbol{X}_D, \boldsymbol{q}) \\
&= \boldsymbol{W}_{\text{ZSL}} \boldsymbol{q} + \sum_i ((\boldsymbol{W}_V \boldsymbol{x}_i) \odot (\boldsymbol{W}_K \boldsymbol{x}_i))^T \boldsymbol{q} \\
&= \boldsymbol{W}_{\text{ZSL}} \boldsymbol{q} + \Delta \boldsymbol{W}_{\text{ICL}} \boldsymbol{q},
\end{aligned}
$$

where $\boldsymbol{W}_{\text{ZSL}} \boldsymbol{q}$ is interpreted as the output in the zero-shot learning (ZSL) where no demonstrations are given. On the other hand, they consider a specific fine-tuning setting, which updates only the parameters for the key and value projection, that is,

$$
\begin{aligned}
\tilde{F}_{\text{FT}}(\boldsymbol{q}) &= (\boldsymbol{W}_V + \Delta \boldsymbol{W}_V) \boldsymbol{X} \boldsymbol{X}^T (\boldsymbol{W}_K + \Delta \boldsymbol{W}_K)^T \boldsymbol{q} \\
&= (\boldsymbol{W}_{\text{ZSL}} + \Delta \boldsymbol{W}_{\text{FT}}) \boldsymbol{q}
\end{aligned}
$$

where $\Delta \boldsymbol{W}_K$ and $\Delta \boldsymbol{W}_V$ denote the parameter updates and they are acquired by back-propagation from task-specific training objectives [Dai et al., 2022], which is a supervised learning process of the original model. Considering the similarity in form between $\tilde{F}_{\text{ICL}}$ and $\tilde{F}_{FT}$, their focus is on establishing a connection between ICL and implicit fine-tuning on the original model.

As a comparison, we turn our attention to establish a connection between ICL and the gradient descent process of the dual model, rather than the original model. More specifically, we consider the dual model $f(\boldsymbol{x}) = \boldsymbol{W}\phi(\boldsymbol{x})$ of the nonlinear attention layer, where the weight $\boldsymbol{W}$ are updated according to the following loss (presented as Eq (9) in Section 3.2):

$$
\mathcal{L} = -\frac{1}{\eta D} \sum_{i=1}^{N} (\boldsymbol{W}_V \boldsymbol{x}_i)^T \boldsymbol{W} \phi(\boldsymbol{W}_K \boldsymbol{x}_i),
$$

where $\boldsymbol{x}_i$ is the $i$-th demonstration token. The prediction output of the trained dual model will be consistent with the ICL output of the attention layer. The gradient descent process of the dual model

using this loss can be viewed from a self-supervised learning lens: unlike in supervised fine-tuning, where the original model is instructed to perform gradient descent using a given objective (loss), this loss formed as Eq (9) is determined (derived) by the attention mechanism itself and it also does not require additional "true label" to supervise each token $\boldsymbol{x}_i$ (so called self-supervised). Therefore, modifications to this self-supervised learning loss will in turn cause modifications in the attention mechanism correspondingly, as we discussed in our work in Section 4. We believe this perspective offers several benefits:

- By analyzing from the dual perspective, we can transform the forward inference process into an optimization process. Since optimization processes are well-known and have established theoretical tools (for example, generalization error as mentioned in Section 3.3), this transformation can provide reverse insights into analyzing the model mechanisms.

- It can clearly observed that the dual model involves a self-supervised representation learning process from the dual perspective. Considering that there are lots of mature works in this area, we can draw on these works to reflect on the attention mechanism, which has also inspired attention modifications as illustrated in Section 4.

- Intuitively, this explanation might be also reasonable as the original model is not explicitly instructed to provide the answer under some given objective (e.g., minimizing cross-entropy) during ICL inference process. Instead, the underlying criterion should be determined by the model's own structure (self-supervised) as we mentioned above.

In addition, although we do not target specific tasks like linear regression as previous works mentioned in Section 6, we would like to point out that under those specific weight and input settings, an intuitive explanation can also be provided from a representation learning perspective. Here, we take the linear regression task as well as the weight constructions considered by Von Oswald et al. [2023a] as an example. Specifically, it assumes that the structured input is $\boldsymbol{H} = [\boldsymbol{h}_i]_{i=1}^N \in \mathbb{R}^{(d+1)\times(N)}$ where $\boldsymbol{h}_i = [\boldsymbol{x}_i, y_i]$ is sampled from some linear task $y = \boldsymbol{w}^T\boldsymbol{x}$ and the query token will be $\boldsymbol{h}_{N+1} = [\boldsymbol{x}_{N+1}, -\boldsymbol{w}_0^T\boldsymbol{x}_{N+1}]$. And the considered linear self-attention layer will take the constructed weights and query output as:

$$\boldsymbol{W}_K = \boldsymbol{W}_Q = \begin{bmatrix} \boldsymbol{I}_{d\times d} & 0 \\ 0 & 0 \end{bmatrix}, \boldsymbol{W}_V = \begin{bmatrix} 0_{d\times d} & 0 \\ \boldsymbol{w}_0^T & -1 \end{bmatrix}, \boldsymbol{P} = \frac{\eta}{N}\boldsymbol{I},$$
$$\tilde{\boldsymbol{h}}_{N+1} = \boldsymbol{h}_{N+1} + \boldsymbol{P}(\boldsymbol{W}_V\boldsymbol{H})(\boldsymbol{W}_K\boldsymbol{H})^T\boldsymbol{W}_Q\boldsymbol{h}_{N+1},$$
(35)

where $\boldsymbol{w}_0$ is the underlying initial matrix. Then the label part of $\tilde{\boldsymbol{h}}_{N+1}$ will has the form as $\tilde{y}_{N+1} = -\boldsymbol{w}_0^T\boldsymbol{x}_{N+1} + \Delta\boldsymbol{w}^T\boldsymbol{x}_{N+1} = -(\boldsymbol{w}_0^T - \frac{\eta}{N}\sum_{i=1}^N(\boldsymbol{w}_0^T\boldsymbol{x}_i - y_i)\boldsymbol{x}_i^T)\boldsymbol{x}_{N+1} = -\hat{y}_{N+1}$, which is equivalent to the output $-\hat{y}_{N+1}$ (multiplied by $-1$) of the linear layer $y = \boldsymbol{w}^T\boldsymbol{x}$ where $\boldsymbol{w}$ is initialized as $\boldsymbol{w}_0$ after performing one step of gradient descent under mean squared loss $\mathcal{L} = \frac{1}{2N}\sum_{i=1}^N \|\boldsymbol{w}_0^T\boldsymbol{x}_i - y_i\|^2$.

In practice, the underlying initial weight matrix $\boldsymbol{w}_0$ is set to be approximately $\boldsymbol{0}$ thus the test input can be formed as $\boldsymbol{h}_{N+1} = [\boldsymbol{x}_i, \boldsymbol{0}]$ [Von Oswald et al., 2023a]. In addition, when reading out the label $\hat{y}_{N+1}$, the test prediction $\tilde{y}_{N+1}$ will be multiplied again by $-1$, which can be done by a final projection matrix (or equivalently, $\boldsymbol{P} = -\frac{\eta}{N}\boldsymbol{I}$). In this case, we first note that the dual model of the linear attention layer can be written as $f(\boldsymbol{z}) = \boldsymbol{W}\boldsymbol{z}$ where $\boldsymbol{W} \in \mathbb{R}^{(d+1)\times(d+1)}$ and similar to Eq (9), it will be trained under the loss below:

$$\min_{\boldsymbol{W}} \mathcal{L} = -\frac{1}{\eta}\sum_{i=1}^N (\boldsymbol{P}\boldsymbol{W}_V\boldsymbol{h}_i)^T \boldsymbol{W}\boldsymbol{W}_K\boldsymbol{h}_i.$$
(36)

By substituting the corresponding weights in Eq (35) where we replace $\boldsymbol{P} = -\frac{\eta}{N}\boldsymbol{I}$ for the readout, the loss can be reformulated as:

$$\min_{\boldsymbol{W}} \mathcal{L} = -\frac{1}{N}\sum_{i=1}^N [0, \ y_i] \boldsymbol{W} \begin{bmatrix} \boldsymbol{x}_i \\ 0 \end{bmatrix}.$$
(37)

Recalling that $\boldsymbol{h}_i = [\boldsymbol{x}_i, y_i]$ is sampled from some linear task $y = \boldsymbol{w}^T\boldsymbol{x}$, we assume that $\|\boldsymbol{W}\|_F \leq \|\boldsymbol{w}\|_2$, it can then be easily seen that the optimal solution for Eq (37) will be

$$\boldsymbol{W}^* = \begin{bmatrix} \boldsymbol{0} & \boldsymbol{0} \\ \boldsymbol{w}^T & \boldsymbol{0} \end{bmatrix}.$$
(38)

Furthermore, similar to Section 3.2, we take $\boldsymbol{W}_Q \boldsymbol{h}_{N+1}$ as the input where $\boldsymbol{W}_Q$ is constructed as Eq (35) and $\boldsymbol{h}_{N+1} = [\boldsymbol{x}_{N+1}, 0]$, the optimal dual model will output the result $f(\boldsymbol{W}_Q \boldsymbol{h}_{N+1}) = \boldsymbol{W}^* \boldsymbol{W}_Q \boldsymbol{h}_{N+1} = [\boldsymbol{0}, \boldsymbol{w}^T \boldsymbol{x}_{N+1}] = [\boldsymbol{0}, y_{N+1}]$ where the label part will be just the answer for the test query. Additionally, it would also be interesting to explore how these weights converge to the constructed form in Eq (35) or other forms under this special setting as previous works illustrated from the perspective of the dual model. Investigating this issue goes beyond the scope of this paper, and we will leave it for future exploration.

