# OpenReview forum: "Towards Understanding How Transformers Learn In-context Through a Representation Learning Lens"
_NeurIPS.cc/2024/Conference — NeurIPS 2024 poster_

### Official Review · Reviewer_8J5b · 2024-07-12

**Soundness:** 3
**Presentation:** 2
**Contribution:** 3
**Rating:** 5
**Confidence:** 4

**Summary:**

The paper explores the ICL capabilities of LLMs, focusing on understanding the mechanisms underlying ICL. The authors aim to investigate the ICL process using representation learning principles. They mainly use kernel methods to develop a dual model for one softmax attention layer, demonstrating that the ICL inference process of this layer aligns with the training procedure of its dual model.

* Extend theoretical analysis to more complex scenarios with multiple attention layers
* Propose potential modifications to the attention layer to enhance ICL capabilities

Experiments also support the theoretical findings and improvements on ICL. This paper may give a deeper understanding of ICL and suggests practical approaches for improving the ICL capabilities of LLMs in real-world applications.

**Strengths:**

* While previous studies mainly on the linear attention, this work extends it to a more realistic case -> softmax attention, including its training dynamics.
* Based on the dual gradient descent process, the authors also propose potential attention modifications, which could be benificial to real-world downstream application.

**Weaknesses:**

* The work may ignore the format of task input/output. And some of recent work such as (Theoretical Understanding of In-Context Learning in Shallow Transformers with Unstructured Data, Xing et al.) the success of ICL may be also because of the alignment of attention mechanism, especially in real-world nosiy cases.
* The experimental section is not well-organized and self-explained, the readers may need to refer extra papers to get the protocols.

**Questions:**

Theoretical Section
* Will the ratio of the number of negative samples have an impact on generalization bound?

Experimental Section

* As mentioned in weeknesses part, the authors may need to add more experimental details to make it more self-explained.
* Since the authors propose three attention modifications, which one could be the best, and do we need add these three together? why or why don't need?
* Does the conclusion hold the same for both one-layer transformer and multi-layer transoformer?
* The dynamics of token interactions may not be well supported in the experiments.

Conclusion Section?

* seems missing a conclusion section due to the restriction of page sizes.

**Limitations:**

The authors have clearly stated the limitations of this work, the selected transformer components of this work, and selected task settings.

---

> ### Author Rebuttal · Authors · 2024-08-06
>
> We sincerely appreciate your recognition of the novelty of our paper. We have carefully addressed each of your concerns and considering the word limits of rebuttal, we notice that some issues might overlap, so we have organized our responses as follows.
>
> > **Weakness 1** (The work may ignore the format of task input/output. And some of recent work...)
> > **& Question 5** (The dynamics of token interactions may not be well supported in the experiments.)
>
> The mentioned work [1] extends the original structured input format [2,3] to the unstructured setting and then explores the impact of various factors (one/two layers, look-ahead attention mask, PE) when learning linear regression tasks both theoretically and experimentally. The explanation for ICL derived from these interesting findings is more aligned with real-world scenarios (where the examples are usually stored in different tokens).
>
> Although our work also relates ICL to GD, our focus is on how we can view the ICL inference process as a GD process on a dual model from a representation learning perspective. We do not make assumptions about the input format, such as $[x; y]$ or $[x; 0], [0; y]$, as we do not target specific tasks (e.g. linear regression tasks) but aim to provide more general explanations. To simplify the analysis, our input tokens can be viewed as embeddings of demonstration sentences (or intermediate states among multi layers).
>
> Correspondingly, in the experimental section, our focus is not on the dynamics of token interactions as specific tasks are not being targeted. Instead, we are more concerned with whether the result of ICL inference $h'\_{T+1}$ is equivalent to the test prediction $\hat{y}_{test}$​ of the trained dual model, as illustrated in the left part of Figure 3. This result aligns with our analysis in Theorem 3.1.
>
> [1] Theoretical Understanding of In-Context Learning in Shallow Transformers with Unstructured Data
>
> [2] Transformers learn in-context by gradient descent
>
> [3] Trained transformers learn linear models in-context
>
> > **Weakness 2** (The experimental section is not well-organized and self-explained...)
> > **& Question 2** (As mentioned in weeknesses part...)
>
> Thank you for your careful suggestions! Due to the resrtiction of page sizes, we simplified the description of the experiments. In fact, more details of our experiments are presented in the Appendix E. We will reorganize our experimental part and provide more details in our future revision.
>
> > **Question 1** (Will the ratio of the number of negative samples have an impact on generalization bound?)
>
> The answer is yes. After introducing negative samples, we consider the following representation loss:
> $$
> \mathcal{L}(f) = \mathbb{E}_{x\sim\mathcal{D}\_{\mathcal{T}}} \left[- \frac{1}{K} \sum\_{j = 1}^{K}  \left( W\phi(W_Kx) \right)^T\left(W_Vx - W_Vx^{-}\_{j} \right)\right],
> $$
> where $K$ is the number of negative samples for each $x_i$ and  $x\_{j}^-$ denotes the $j$-th negative sample for token $x$. The empirical loss is modified correspondingly. Then, by the other definitions in Section 3.3, we can obtain the generalization bound as
> $$
> \mathcal{L}(\hat{f})\le \mathcal{L}(f) + O\left(w\rho d\_o \sqrt{\mathrm{Tr}(K\_S)\left(\frac{5}{N^2} + \frac{1}{rN^3}\right)} + \sqrt{\frac{log\frac{1}{\delta}}{N}}\right),
> $$
> where $r = \frac{K}{N}$ is the ratio of the number of negative samples. It can be observed that as the ratio increases, the generalization error will decrease. However, we also notice that $\frac{5}{N^2} > \frac{1}{rN^3}$​​ thus the reduction in generalization error due to an increased proportion of negative samples is limited. The proof is similar to that of Theorem 3.2 (see Appendix C). The main difference is that we need to define the new function class $G$ and $F$ according to above definition. Nevertheless, we do not rule out the possibility of a tighter generalization bound using better theoretical tools.
>
> > **Question 3** (Since the authors ... which one could be the best ... need add these three together? why...?)
>
> The answer is related to the specific tasks but generally speaking, using all three methods together is not necessary, especially when the design of  augmentated or negative modification is not effective enough. We conduct experiments with their combinations  on linear, trigonometric, and exponential tasks. **The results are shown in Figure 1 of the  PDF** in  `To AC and All Reviewers`. For the latter two tasks, the results are actually worse than using regularized or augmented modification individually (compared to Figures 11 and 12 in Appendix E). Therefore, when the design of augmented or negative mofification is not effective, we recommend using a single modification method.
>
> > **Question 4** (Does the conclusion hold the same for both one-layer transformer and multi-layer transformer?)
>
> For more scenarios, the conclusion still holds: using all three methods simultaneously is not necessary. We supplemented our experiments using BERT (multi-layer Transformer), see ` (To AC and All Reviewers)`. From Table 1 in the PDF, the combined models do not outperform the augmented version alone. We also observed that negative sample improvements remain quite limited, indicating that our method for selecting negative samples is not effective enough. In these cases, using augmented modification alone will be a better choice.
>
> > **Question 6** (seems missing a conclusion section)
>
> We will streamline our paper content and supplement the conclusion section based on the valuable questions raised by all you reviewers in the future revision. Thank you very much for your reminder!
>
> **Final Note:** We want to thank you again for all the questions you have provided. If there are any remaining questions, please do not hesitate to let us know.

---

> > ### Author Response · Authors · 2024-08-13
> > **Looking forward to your reply**
> >
> > We sincerely appreciate the time and effort you have dedicated to reviewing our manuscript and providing us with your valuable feedback！
> >
> > As the author-reviewer discussion phase is drawing to a close, we kindly wish to confirm whether our response has adequately addressed your concerns. A few days ago, we submitted a detailed response addressing your concerns and hope that they have adequately resolved any issues. If there are any remaining questions, please do not hesitate to let us know.
> >
> > We would greatly appreciate any additional feedback you may have!

---

### Official Review · Reviewer_k2vC · 2024-07-12

**Soundness:** 3
**Presentation:** 3
**Contribution:** 3
**Rating:** 5
**Confidence:** 1

**Summary:**

The paper explores the in-context learning (ICL) abilities of Transformer-based models. The authors propose an interpretation of ICL through the lens of representation learning. They establish a connection between the inference process of softmax attention layers in Transformers and the gradient descent process of a dual model, providing theoretical insights and generalization bounds. The paper also suggests potential modifications to the attention mechanism inspired by contrastive learning techniques and supports its findings with experiments.

**Strengths:**

(Reminder) I'm not a researcher working in the theory field, so I couldn't recognize this paper's theoretical contributions well.

- The paper provides a fresh perspective on understanding ICL by connecting it to gradient descent and representation learning, which is a novel and insightful contribution to the field.
- The authors extend their theoretical findings to various Transformer settings, including single and multiple attention layers, enhancing their conclusions' generalizability.

**Weaknesses:**

(Reminder) I'm not a researcher in the theory field, so I couldn't also recognize this paper's limitation from a perspective.

- The paper relies on several assumptions and simplifications, such as ignoring the impact of layer normalization and residual connections. These might limit the applicability of the findings in more complex Transformer architectures.
- Lack of discussions towards some related works that argue against the equivalence between ICL and gradient descent [1,2].

[1] In-context Learning and Gradient Descent Revisited, 2023
[2] Do pretrained Transformers Learn In-Context by Gradient Descent, 2023

**Questions:**

- How would the inclusion of layer normalization and residual connections affect the theoretical framework and findings?
- How do the proposed methods perform on more complex and diverse tasks beyond the linear regression tasks used in the experiments?

**Limitations:**

The experiments are primarily focused on linear regression tasks, and it remains to be seen how well the proposed methods generalize to a wider range of tasks and datasets.

---

> ### Author Rebuttal · Authors · 2024-08-06
>
> We thank the reviewer for acknowledging the novel perspective on understanding ICL of our paper. Considering the word limits of rebuttal, we notice that some issues might overlap, so we have organized our responses as follows.
>
> > **Weakness 1** (The paper relies on several assumptions and simplifications...)
> > **& Question 1** (How would the inclusion of layer normalization and residual connections affect the theoretical framework and findings?)
>
> As for residual connection, intuitively, our conclusions can be naturally extended to versions with residual connections, which are analogous to introducing a residual connection structure in dual models, i.e., $\mathrm{Atten}(x) + x = \hat{W}\phi(W_Qx) + x$, where $\hat{W}$ is the weight in the dual model $f(x) = W\phi(x)$ trained under the original representation loss in Eq (9).
> This modification does not have a significant impact on the specific representation loss or generalization bound, but its effect on the GD process of the dual model still requires more detailed analysis.
>
> As for layer normalization, from the representation learning perspective of the dual model, it normalizes $x$ to a scaled unit sphere to make the process more stable intuitively. However, we acknowledge the analytical challenges posed by the nonlinearity of layer normalization, and the more detailed effects still warrant further investigation. This may require more powerful theoretical tools, e.g. [1],  which is also a direction for our future research.
>
> [1] On the Nonlinearity of Layer Normalization.
>
> > **Weakness 2** (Lack of discussions towards some related works that argue against the equivalence between ICL and gradient descent...)
>
> Following the previous work [1], the first mentioned work [2] further explores potential shortcomings in the previously used evaluation metrics and further points out that the ICL output is more dependent on previous lower layers, which is different from the finetuning that relies on all model parameters. Inspired by this, it introduces a new finetuning method called LCGD and validates the claim with new metrics.
>
> Unlike previous works including [3,4], which studies ICL and Gradient Descent (GD) under strong constraints (specific regression tasks and weight constructions, etc.), the second mentioned work [5] highlights significant gaps between these strong constraints and real-world LLMs. They observe that there are differences in how ICL and GD modify the output distribution and emphasize the distinction between naturally emergent ICL (pretrained on natural text data) and task-specific ICL (such as in [3,4]). It provides a clearer definition and experimental results in more realistic settings.
>
> Although our work also relates ICL to GD, our focus is on how we can view the ICL inference process as a GD process on a dual model from a representation learning perspective:
> - To simplify theoretical analysis, we concentrate on analyzing the core component of the Transformer architecture—the attention layer—while neglecting residual connections and layer normalization. This does introduce some divergence from real-world large language models;
> - In terms of research approach, we focus on the GD process of dual model under representation learning loss, rather than analyzing GD directly on the original model as in [2,5]. This is also a significant departure from the mentioned works.
> - Additionally, the representation learning process of the dual model resembles a self-supervised process, which we believe is reasonable: unlike fine-tuning, ICL inference is not explicitly provided specific metrics to adapt to the target task and this also aligns with certain aspects of previous work [5]. And we also compare this process with existing self-supervised learning methods in our work.
> - Inspired by this representation learning perspective, we propose modifications to the attention mechanism, which were not sufficiently explored in previous work.
>
> Thank you for your suggestion and we will add more supplementary information on the relevance between our work and the mentioned works in our future revision.
>
> [1] Why Can GPT Learn In-Context? Language Models Implicitly Perform Gradient Descent as Meta-Optimizers
>
> [2] In-context Learning and Gradient Descent Revisited
>
> [3] Transformers Learn In-context by Gradient Descent
>
> [4] Trained Transformers Learn Linear Models In-context
>
> [5] Do pretrained Transformers Learn In-Context by Gradient Descent?
>
>
> > **Question 2:** How do the proposed methods perform on more complex and diverse tasks beyond the linear regression tasks used in the experiments?
>
> For more complex tasks, considering the limited computational resources and time constraints, we select the pre-trained BERT-base-uncased model to apply attention modifications, and validate the results on part of GLUE datasets.  **More experimental settings are detailed in the (global) author rebuttal and the results are presented in the Table 1 of PDF** `(To AC and All Reviewers)`.
>
> **For the regularized models**, we select different values $\alpha$ and conclude that smaller absolute values of $\alpha$ are recommended. **For the augmented models**, we adopt a parallel MLP approach for data augmentation $g_1/g_2$, that is, $g_1(W_Vx) =  W_Vx  +  c W_2\sigma(W_1x)$​. The results suggest that using $g_1$ alone may be a more effective choice under our settings. **For the negative sample models**, we continued to select tokens with lower attention scores as negative samples. The final results showed limited improvement in model performance. We think that this may be due to the ineffective selection of negative sample tokens, and better methods for selecting negative samples would be interesting to explore in the future.
> In conclusion, these results validate the potential of our proposed methods.
>
> **Final Note:** We want to thank you again for all the questions you have provided. If there are any remaining questions, please do not hesitate to let us know.

---

> > ### Author Response · Authors · 2024-08-13
> > **Looking forward to your reply**
> >
> > We are truly grateful for the time and effort you have invested in reviewing our manuscript and offering your valuable feedback.
> >
> > As the author-reviewer discussion phase is drawing to a close, we kindly wish to ensure that our recent detailed response, submitted a few days ago, has effectively addressed all of your concerns. If any issues remain or if further clarification is needed, please do not hesitate to reach out.
> >
> > We would greatly appreciate any additional feedback you may have!

---

### Official Review · Reviewer_cRiK · 2024-07-13

**Soundness:** 3
**Presentation:** 3
**Contribution:** 4
**Rating:** 8
**Confidence:** 4

**Summary:**

The author's present a new way of linking in-context-learning (ICL) to gradient descent.

The author's are able to demonstrate that indeed a (simplified) transformer decoder layer ICL is equivalent to "representation learning".

Using the theoretical findings the authors are also able to propose extensions to the attention mechanism.

The paper shows theoretical proofs for these claims as well as additional experiments to demonstrate the claim.

**Strengths:**

Strenghts:
1. Excellent theoretical contribution to understanding in-context-learning
2. Adding relevant proofs
3. Using theoretical results to motivate better attention mechanisms

**Weaknesses:**

Weakness:
1. Slightly simplified transformer architecture is used (it would be interesting to see the skip-connection version). As recent work, such as "mechanistic interpretability" often relies on these skip connections.
2. The experimental setup and results are a bit limited. Slightly more description of the task would be good, as well as a somewhat "realistic" task would be interesting to see - and what the results mean qualitatively.
3. A small section on how these results can be applied for bigger models in practice.

**Questions:**

1. How could a "realistic" task look like for your evaluation?
2. How could one practically extend your augementations to e.g. modern LLMs (even of `moderate' size, e.g. 7B)
- I.e. what would these "data augmentations" or "negative samples" really look like?

**Limitations:**

Yes.

---

> ### Author Rebuttal · Authors · 2024-08-06
>
> We sincerely thank the reviewer for acknowledging the theoretical contribution of our paper. We have carefully addressed each of your concerns and considering the word limits of rebuttal, we notice that some issues might overlap, so we have organized our responses as follows.
>
> > **Weakness 1**: Slightly simplified transformer architecture is used (it would be interesting to see the skip-connection version) ...
>
> The attention module, as one of the core components of the transformer, is crucial for extracting contextual information. Therefore, for the sake of analytical simplicity, we focus on comparing pure attention modules/mechanisms. Intuitively, our conclusions can be naturally extended to versions with residual connections, which are analogous to introducing a residual connection structure in dual models, i.e., $\mathrm{Atten}(x) + x = \hat{W}\phi(W_Qx) + x$, where $\hat{W}$ is the weight in the dual model $f(x) = W\phi(x)$​ trained under the original representation loss in Eq (9). Although it is relatively straightforward to connect residual connections with dual models in form, the role of such residual connections in the learning process of the dual model's weight from a representation learning perspective still requires further investigation. Therefore, a more reasonable explanation is currently lacking. We believe this is an interesting direction for future research.
>
> > **Weakness 2** (The experimental setup and results are a bit limited ... as well as a somewhat "realistic" task would be interesting to see - and what the results mean qualitatively. )
> > **& Question 1** （How could a "realistic" task look like for your evaluation?）
>
> Due to constraints of page sizes, we have simplified the description of the experiments for simulation tasks in the main body. In fact, more detailed experimental setups are presented in the Appendix E. We will provide more detailed supplementary information on the experimental setups if space permits in future revisions.
>
> For the evaluation on more realistic tasks, we have supplemented our work with additional experiments. **More experimental settings are detailed in the (global) author rebuttal and the results is presented in the Table 1 of PDF** `(To AC and All Reviewers)`.
>
> Regarding the qualitative results: **For the regularized models**, smaller absolute values of $\alpha$ are recommended. We interpret this as adding a small scaled identity matrix to the attention score matrix helps achieve full rank, thereby better preserving information. **For the augmented models**, to better retain the information from the pre-trained $W_V$ and $W_K$ weights, we adopted a parallel MLP approach for data augmentation $g_1$ and $g_2$. For example, $g_1(W_Vx) =  W_Vx  +  c W_2\sigma(W_1x)$. The results suggest that enhancing the value mapping may be a more effective choice. **For the negative sample models**, we continued to select tokens with lower attention scores as negative samples while the final results showed limited improvement in model performance. We believe this may be due to the ineffective selection of negative sample tokens, and better methods for selecting negative samples would be interesting to explore in the future.
>
> > **Weakness 3** (A small section on how these results can be applied for bigger models in practice.)
> > **& Question 2**(How could one practically extend your augementations to e.g. modern LLMs ...)
>
> We believe that our supplementary experiments on pre-trained BERT provide more insights into this issue (see global rebuttal `(To AC and All Reviewers)`). Although the scale of the model and datasets selected is relatively small due to our limited computational resource and time constraints,  we believe similar approaches would naturally extend to larger-scale experiments.
>
> Take the augmented models as the example: we also considered using more complex data augmentations for the linear key/value mapping ($g_1(W_Vx)$ and $g_2(W_Kx)$). However, unlike previous methods used in simulation tasks, we do not choose $g_1/g_2$ as $g_1(W_Vx)=W_2σ(W_1W_Vx)$, because our experiments showed that this design struggles to fully leverage the pre-trained weights $W_V$ and $W_K$. Thus, we adopted a parallel approach, that is,  $g_1(W_Vx) =  W_Vx  +  c W_2\sigma(W_1x)$, where $c$ is a hyperparameter to control the influence of the new branch ($g_2$ is similar). This approach introduces nonlinear "augmentation" while preserving the knowledge from the original pre-trained weights, making it easier to train. The results in Table 1 also prove the effectiveness of this method especially using $g_1$ alone, which leads to the most significant improvement in model performance.
>
> Additionally, we do not rule out the possibility of more powerful and efficient augmentation methods. The parallel MLP approach is chosen primarily to make better use of the pre-trained weights $W_V$/ $W_K$ and this design is quite general and does not consider specific characteristics of particular tasks. We still encourage the design of more task-specific augmentation format for different tasks. For example, in CV tasks, it might be natural to incorporate CNN-extracted features into the $g_1$/$g_2$ in ViTs.
>
> Similarly, our discussion on the negative sample models further supports the need for more task-specific designs. As shown in Table 1, selecting tokens with low attention scores as negative samples is a rather crude approach, leading to relatively limited performance improvements. Exploring how to select or construct higher-quality negative samples is also an interesting direction. These detailed considerations go beyond the scope of this paper, and these interesting directions will be the focus of our future work.
>
> **Final Note**: We are excited that you acknowledge our theoretical contribution. If there are any remaining questions, please do not hesitate to let us know. Thank you once again for your insightful comments and for your encouraging feedback!

---

> > ### Comment · Reviewer_cRiK · 2024-08-08
> >
> > Thank you for your rebuttal and additional comments.
> >
> > At this stage there are no questions on your rebuttal to my questions.
> >
> > With regards to a response to another reviewer (k2cV).
> >
> > > Weakness 2 (Lack of discussions towards some related works that argue against the equivalence between ICL and gradient descent...)
> >
> > The answer is not very clear. Could you elaborate on:
> > 1. A 1-1 (quick) comparison against each of the five papers. [1-5].
> > 2. A more detailed explanation why using the dual is important. (As the dual is simply a mathematical tool).

---

> > > ### Author Response · Authors · 2024-08-09
> > > **Replying to Official Comment by Reviewer cRiK**
> > >
> > > Thank you sincerely for your feedback. Here is our response:
> > >
> > > > **Question 1: ** A 1-1 (quick) comparison against each of the five papers. [1-5].
> > >
> > > - Oswald et al. [1] focus on the ability of the linear attention layer to perform gradient descent during ICL when faced with  linear regression tasks. Their observations are based on specific assumptions, including the constructed forms for $W_Q, W_K, W_V$ and the input tokens (concatenated $[x,y]$). Our work does not target specific tasks like linear regression; therefore, we do not make detailed assumptions about the model weights (simply treated as weights after pre-training) or construct specific input forms. Instead, we aim to view the ICL inference process from the perspective of representation learning in the dual model.
> > >
> > > - Zhang et al. [2] also theoretically analyze the gradient flow of linear self-attention modules under certain initialization conditions when dealing with linear regression tasks and provide the forms of weights at the global minimum as well as the prediction error. Similar to the comparison above, the aim of our work is not to investigate the expressive capabilities of the model's structure for these specific tasks but to interpret the ICL inference process from a representation learning lens in a more general setting, thus we do not use these assumptions.
> > >
> > > - Shen et al. [3] experimentally illustrates that the assumptions used in previous works including Oswald et al. [1] and Zhang et al. [2] may be too strong in real LLMs. They analyze the differences between ICL inference in LLMs and the fine-tuned models in real-world scenarios from various perspectives, including weight sparsity, order stability, and output distribution. In comparison, our work: (1) for the sake of theoretical analysis, considers not a "complete" real model but a simplified one that omits structures like residual connections; and (2) our interpretation of the ICL inference is not linked to fine-tuning on the original model but rather to training on the dual model.
> > >
> > > - Dai et al. [4] use the dual form to interpret ICL as an implicit fine-tuning (gradient descent) of the original model under the linear attention setting and this alignment is ambiguous as the specific details of the gradient descent process is not clear. Thus, our work extends this analysis to the nonlinear attention setting and delve deeper into more details of this process (exploring the specific form of the loss function). The main difference is that we consider this as training the dual model under a self-supervised representation learning loss, rather than performing supervised fine-tuning on the original model.
> > >
> > > - Natan et al. [5]  investigate potential shortcomings in the evaluation metrics used by Dai et al.[4] in real model assessments and propose a layer-causal GD variant that performs better in simulating ICL. However, similar to the comparison above [3], their study discusses complex gradients in the original model in real scenarios, whereas we turn our attention to the gradient descent of the dual model of the attention layer.
> > >
> > > > **Question 2: **A more detailed explanation why using the dual is important. (As the dual is simply a mathematical tool).
> > >
> > > By using the dual, or rather, by analyzing the potential gradient descent process of the dual model, we can gain new insights into the model mechanisms in reverse. Specifically, through the dual perspective,
> > >
> > > - we transform the forward inference process into an optimization process. Since optimization processes are well-known and have established theoretical tools (for example, generalization error as mentioned in our work), this transformation can provide reverse insights into analyzing the model mechanisms.
> > >
> > > - we can clearly observe that the dual model involves a self-supervised representation learning process. Considering that there are lots of mature works in this area, we can draw on these works to reflect on the attention mechanism, which has also inspired attention modifications as illustrated in our work.
> > >
> > > Finally, as for "the dual is simply a mathematical tool", we want to clarify that the term "dual" we use is different from the one in  optimization within the mathematical field. Instead, it follows the terminology used in previous work [4], where the forward process of the attention layer and backward process on some model are referred to as a form of "dual". We are not sure if this has also led to some misunderstandings. If there are any remaining questions, please do not hesitate to let us know!
> > >
> > > [1] Von Oswald J, et al. Transformers learn in-context by gradient descent.
> > >
> > > [2] Zhang R, et al. Trained transformers learn linear models in-context.
> > >
> > > [3] Shen L, et al. Do pretrained Transformers Really Learn In-context by Gradient Descent?
> > >
> > > [4] Dai D, et al. Why can gpt learn in-context? language models implicitly perform gradient descent as meta-optimizers
> > >
> > > [5] Nathan T B, et al. In-context learning and gradient descent revisited

---

> > > > ### Comment · Reviewer_cRiK · 2024-08-11
> > > >
> > > > Thank you for clarifying the points and taking the time to do a comparison against each paper.
> > > >
> > > > The “dual” has indeed led to slight confusion and perhaps the link to previous should be made clearer.
> > > >
> > > > Also, generally, having this concise summary of comparisions would be handy in the paper as well. (As it helps understand the contribution very precisely.)
> > > >
> > > > —-
> > > > Also, as a follow-up to the comparison to [4]
> > > > 1. Could you elaborate on the exact difference between “supervised learning of the original model” and “self-representation learning of the ‘dual’ model”?
> > > >
> > > > Thank you.

---

> > > > > ### Author Response · Authors · 2024-08-12
> > > > > **Replying to New Official Comment by Reviewer cRiK**
> > > > >
> > > > > We sincerely appreciate your suggestions, which are greatly beneficial in improving our work！
> > > > >
> > > > > - We will further clarify in Section 2.3 that the term "dual" we use is linked to prior work [1], but differs from the "dual" concept in mathematical optimization, to avoid causing any potential misunderstanding for readers.
> > > > >
> > > > > - Additionally, we will also include comparisons with the aforementioned related works in the main body of our future revision to make our contributions more precisely.
> > > > >
> > > > > > **Question:** Could you elaborate on the exact difference between “supervised learning of the original model” and “self-representation learning of the ‘dual’ model”?
> > > > >
> > > > > Dai et al. [1] interpret ICL as implicit fine-tuning: More specifically, let $X = [X_D;X_T]$ where $X_D = [x_1;...;x_N]$ denotes the demonstration tokens and $X_T= [x'\_1;...;x'\_{T}]$ be query tokens. On the one hand, for ICL, they consider the output of $q = W_Qx’\_{T+1}$ under the linear attention setting as
> > > > > $$
> > > > > \begin{aligned}
> > > > > 	\tilde{F}\_{\mathrm{ICL}} (q) &= W\_{V}[X_D; X_T](W_K[X_D; X_T])^T q \\\\
> > > > > 	&= W_{V}X_{T}(W_KX_T)^Tq + W_{V}X_{D}(W_KX_D)^Tq \\\\
> > > > > 	&= W_{\mathrm{ZSL}}q +  \mathrm{LinearAtten}(W_VX_{D}, W_{K}X_{D}, q) \\\\
> > > > > 	&= W_{\mathrm{ZSL}}q + \sum_{i}\left( (W_{V}x_{i}) \odot (W_{K}x_{i}) \right)q \\\\
> > > > > 	&= W_{\mathrm{ZSL}}q + \Delta W_{\mathrm{ICL}}q ,
> > > > > \end{aligned} \\
> > > > > $$
> > > > >
> > > > > where $W\_{\mathrm{ZSL}}q$ is interpreted as the output in the zero-shot learning (ZSL) where no demonstrations are given.
> > > > >
> > > > > On the other hand, they consider a specific fine-tuning setting, which updates only the parameters for the key and value projection, that is,
> > > > > $$
> > > > > \begin{align}
> > > > > \tilde{F}\_{\mathrm{FT}}(q) &= (W_{V} + \Delta W_{V})XX^T(W_K + \Delta W_K)^Tq \\\\
> > > > > &= (W_{\mathrm{ZSL}} + \Delta W_{\mathrm{FT}})q,
> > > > > \end{align}
> > > > > $$
> > > > > where $\Delta W_K$ and $\Delta W_V$ denote the parameter updates and **they are "acquired by back-propagation from task-specific training objectives" [1], which is a supervised learning process of the original model**. Considering the similarity in form between $\tilde{F}\_{\mathrm{ICL}}$ and $\tilde{F}\_{\mathrm{FT}}$​, their focus is on establishing a connection between ICL and implicit fine-tuning on the original model.
> > > > >
> > > > > ------
> > > > >
> > > > > As a comparison, we turn our attention to establish a connection between ICL and the gradient descent process of the dual model, rather than the original model. More specifically, we consider the dual model $f(x)=Wϕ(x)$ of the nonlinear attention layer, where the weight $W$ are updated according to the following loss (presented as Eq (9) in Section 3.2):
> > > > > $$
> > > > > \mathcal{L} = -\frac{1}{\eta D}\sum_{i=1}^{N}(W_Vx_i)^TW\phi(W_Kx_i),
> > > > > $$
> > > > > where $x_i$ is the i-th demonstration token. The prediction output of the trained dual model will be consistent with the ICL output of the attention layer. The gradient descent process of the dual model using this loss can be viewed from a self-supervised learning lens:  unlike in supervised fine-tuning where the original model is instructed to perform gradient descent using a given objective (loss), **this loss is determined (derived) by the attention mechanism itself and it also does not require additional "true label" to supervise each token $x_i$​ (so called self-supervised)**. Therefore, modifications to this self-supervised learning loss will in turn cause modifications in the attention mechanism correspondingly, as we discussed in our work in Section 4.
> > > > >
> > > > > We believe this perspective offers several benefits:
> > > > >
> > > > > - By analyzing from the dual perspective, our analysis shifts to an optimization process, which makes it easier for us to use existing theoretical tools, as mentioned in our response to previous question ("A more detailed explanation why using the dual is important").
> > > > >
> > > > > - This self-supervised learning process of the dual model can also inspire us to draw on existing works to modify the attention mechanism, as also mentioned in our previous response.
> > > > >
> > > > > - Intuitively, this explanation might be more reasonable as the original model is not explicitly instructed to provide the answer under some given objective (e.g., minimizing cross-entropy) during ICL inference process. Instead, the underlying criterion should be determined by the model's own structure (self-supervised) as we mentioned above.
> > > > >
> > > > > Thank you once again for your valuable suggestions!
> > > > >
> > > > >
> > > > > [1] Dai D, et al. Why can gpt learn in-context? language models implicitly perform gradient descent as meta-optimizers

---

> > > > > > ### Comment · Reviewer_cRiK · 2024-08-12
> > > > > >
> > > > > > Thank you for all the clarifications and aiming for a clarified version.
> > > > > >
> > > > > > The confidence score is now updated from 3 to 4.

---

> > > > > > > ### Author Response · Authors · 2024-08-12
> > > > > > > **Replying to Official Comment by Reviewer cRiK**
> > > > > > >
> > > > > > > We are deeply encouraged by your acknowledgement, and we promise to improve our work in the future version as discussed previously.
> > > > > > >
> > > > > > > Once again, we sincerely thank you for your suggestions and contributions to the discussion process!

---

### Official Review · Reviewer_HnPE · 2024-07-20

**Soundness:** 3
**Presentation:** 3
**Contribution:** 3
**Rating:** 6
**Confidence:** 4

**Summary:**

The paper investigates the in-context learning (ICL) capabilities of Transformers, explaining it through a representation learning lens. It establishes a theoretical connection between ICL and gradient descent, deriving a generalization error bound tied to demonstration tokens. The authors also suggest modifications to the attention mechanism inspired by theory and support their findings with experiments.

**Strengths:**

* **Theoretical Depth**: It provides a rigorous theoretical analysis, including the derivation of a generalization error bound, which contributes to the theoretical foundation of Transformer models.
* **Insights into Attention Mechanism**: The paper provides potential modifications to the attention layer, inspired by theory, which could potentially improve the learning capabilities of Transformers.
* **Good Writing**: Clear Structure and Presentation: The paper is well-structured, with a clear abstract and a good introduction. It is easy for readers to understand.

**Weaknesses:**

* **Generalization to Other Tasks**: The paper's findings are based on specific tasks. It's unclear how well these insights would generalize to general language tasks.
* **Empirical Validation**: While the paper includes experiments, the extent of empirical validation is limited. We may need more experiments on real-world language tasks to verify the modification of the attention mechanism.

**Questions:**

Please see the weakness section.

**Limitations:**

The paper has discussed the limitations sufficiently.

---

> ### Author Rebuttal · Authors · 2024-08-06
>
> We thank the reviewer for the encouraging feedback, especially for recognizing the theoretical contribution and insights of our paper. Our response is detailed below.
>
> > **Weakness 1**: Generalization to Other Tasks: The paper's findings are based on specific tasks. It's unclear how well these insights would generalize to general language tasks.
>
> In fact, our findings can be naturally extended to more realistic tasks. We supplement our experiments on more general language tasks. **More experimental settings are detailed in the (global) author rebuttal and the results is presented in the Table 1 of PDF** `(To AC and All Reviewers)`. Considering the limited computational resources and time constraints, we choose the BERT-base-uncased model to and select part of GLUE datasets to validate the effectiveness of modifications to the attention mechanism. More discussions on the results are provided in detail in our response to Weakness 2.
>
> > **Weakness 2**:  Empirical Validation: While the paper includes experiments, the extent of empirical validation is limited. We may need more experiments on real-world language tasks to verify the modification of the attention mechanism.
>
> To answer the questions, below we discuss the main experimental results presented in the Table 1 of PDF `(To AC and All Reviewers)`.
>
> **For the regularized modification**, we consider different values of $\alpha$, and as can be observed in Table 1, except for RTE, the best regularized models outperform the original model on the other three datasets. while we also note that when the absolute value of $\alpha$  is too large, the model's performance declines significantly, so smaller absolute values for $\alpha$ is recommended.
>
> **For the augmented modification**, we also consider applying more complex ''augmentation'' functions to the linear key/value mappings. Specifically, we adopt a parallel approach, i.e., $g_1(W_Vx) =  W_Vx  +  c W_2\sigma(W_1x)$, where $c$ is a hyperparameter to control the influence of the new branch, $\sigma$ is the GELU activation function and the hidden layer dimension is set to twice the original size of $W_Vx$. And $g_2(W_Kx) =  W_Kx  +  c W_2\sigma(W_1x)$ follows the same format. Experimental results show that the best augmented models achieve better performance than the original model across all four datasets. Notably, using $g_1$ alone proves to be more effective than other methods and using both $g_1$ and $g_2$ introduces more parameters, which is particularly significant for larger models. Thus, under the augmentation methods and experimental settings we selected, using $g_1$​ alone is recommended.
>
> **For the negative modification**, we continue to select tokens with lower attention scores as negative samples. The parameter $r$ represents the ratio of tokens used as negative samples, while $\beta$ indicates the overall reduction in attention scores. And the results show that the best negative models only outperform the original model on CoLA and STS-B, whereas their performance on MRPC and RTE is worse than the original one. This suggests that our simple approach of considering tokens with low attention scores as negative samples might be too coarse. A more effective method for constructing negative samples should be designed, which is a direction worth exploring in the future.
>
> We also consider **combining different modification methods**. The results indicate that under our settings, the combination of augmented and negative modification achieves the best performance on CoLA, MRPC, and RTE, while the combination of regularized and augmented modification achieves the best performance on STS-B. However, their optimal performance is slightly inferior to the best performance achieved with augmented models alone. Therefore, we conclude that using all three modifications simultaneously is not necessary. With appropriate hyperparameter choices, using augmented modification alone or in combination with one other modification is sufficient.
>
> In conclusion, the experimental results show the potential of our approach of improving the attention mechanism from a representation learning perspective. However, due to time and computational resource limitations, our experiments are conducted on a limited tasks and model size. More detailed parameter searches, validation across additional tasks and models, and the development of task-specific augmentation and negative sampling methods are all interesting directions worth exploring in the future.
>
> **Final Note**: Thank you for your detailed review. We are excited that you found our novel insights into attention mechanism. If there are any remaining questions, please do not hesitate to let us know.

---

> ### Author Response · Authors · 2024-08-13
> **Looking forward to your reply**
>
> We sincerely appreciate your time and effort in reviewing our manuscript and providing valuable feedback!
>
> As the author-reviewer discussion phase is nearing its end, we would like to confirm whether our response has effectively addressed your concerns. We provided a detailed response to your concerns a few days ago and hope that they have adequately addressed your concerns. If there are any remaining questions, please do not hesitate to let us know.
>
> We would greatly appreciate any additional feedback you may have!

---

> > ### Comment · Reviewer_HnPE · 2024-08-13
> >
> > Thank you for your detailed responses, my concerns have been addressed and I will maintain my score.

---

> > > ### Author Response · Authors · 2024-08-14
> > > **Replying to Official Comment by Reviewer HnPE**
> > >
> > > We have received your feedback and would like to express our sincere gratitude for the time you dedicated to the review and for the valuable suggestions you shared with us!

---

### Author Rebuttal · Authors · 2024-08-06

### **To AC and All Reviewers**

We thank the reviewers for providing valuable suggestions that help us improve our paper.

We are particularly encouraged that the reviewers have found that (i) the fresh perspective on understanding attention mechanisms `(HnPE, cRiK, k2vC,8J5b) ` ,  (ii) thorough theoretical analysis `(HnPE, cRiK, 8J5b)`,  (iii) the potential of proposed attention modifications `(8J5b, HnPE, cRiK, k2vC)`, and (iv) good writing `(HnPE)` of our work.

In response to the feedback, we've done our best to address each concern and have added new experiments and theoretical results.

We notice that the reviewers share a common concern regarding the generalizability of our findings to broader NLP tasks. Below, we provide a detailed response to the additional experiments we've conducted:

- **Basic Experiment Setting**: We supplement our experiments on more realistic NLP tasks. Considering the limited computational resources and time constraints, we choose the BERT-base-uncased model (hereafter referred to as BERT) to validate the effectiveness of modifications to the attention mechanism. As for datasets, we select part of GLUE (CoLA, MRPC, STS-B, RTE). We load the checkpoint of the pre-trained BERT model (where the classifier are newly initialized) and then fine-tune the model to explore the performance of three modifications as well as their combinations. We set the batch size to 32, the learning rate to 2e-5, and the number of epochs to 5 for all datasets. All experiments are conducted on a single 24GB NVIDIA GeForce RTX 3090. **All experimental results are presented in Table 1 of PDF**. Below, we discuss the setting of various modifications and their performance.

- **For the regularized modification**, we consider different values of $\alpha$. As can be observed in Table 1, except for RTE, the best regularized models outperform the original model on the other three datasets. However, we also note that when $|\alpha|$  is too large, the model's performance declines significantly, so we recommend using smaller $|\alpha|$.

- **For the augmented modification**, we also consider applying more complex ''augmentation'' functions to the linear key/value mappings. However, unlike the previous methods used in simulation tasks, we do not simply select $g_1$ and $g_2$ as MLPs, i.e., $g_1(W_Vx)=W_2σ(W_1W_Vx)$ because it could undermine the effort made during pre-training to learn the weights $W_V$ and $W_K$, leading to difficulties in training and challenges in comparison. Instead, we adopt a parallel approach, i.e., $g_1(W_Vx) =  W_Vx  +  c W_2\sigma(W_1x)$, where $c$ is a hyperparameter to control the influence of the new branch, $\sigma$ is GELU and the hidden dimension is set to twice the original size of $W_Vx$. And $g_2(W_Kx) =  W_Kx  +  c W_2\sigma(W_1x)$ follows the same format.

  Experimental results show that the best augmented models achieve better performance than the original model across all four datasets. Notably, augmentation on the value mapping (i.e., using $g_1$ alone) proves to be more effective than other methods, both in terms of performance and the amount of additional parameters introduced. Using both $g_1$ and $g_2$ introduces more parameters, which is particularly significant for larger models. Thus, under the augmentation methods and experimental settings we selected, using $u_1$ alone is recommended.

  In addition, we do not rule out the possibility of more powerful and efficient augmentation methods. Our choice of $g_1$ and $g_2$ is primarily motivated by the desire to make better use of the pre-trained weights $W_K$ and $W_V$. This design is relatively general and does not take into account the specific characteristics of individual tasks. We still encourage the development of more task-specific augmentation strategies tailored to different tasks.


- **For the negative modification**, we continue to select tokens with lower attention scores as negative samples. The parameter $r$ represents the proportion of tokens used as negative samples, while $\beta$ indicates the overall reduction in attention scores. The best negative models only outperform the original model on CoLA and STS-B, whereas their performance on MRPC and RTE is worse than the original one. This suggests that our simple approach of considering tokens with low attention scores as negative samples might be too coarse. A more effective method for constructing negative samples should be designed, which is a direction worth exploring in the future.

- We also consider **combining different modification methods**. The results indicate that under our settings, the combination of augmented and negative modification achieves the best performance on CoLA, MRPC, and RTE, while the combination of regularized and augmented modification achieves the best performance on STS-B. However, their optimal performance is slightly inferior to the best performance achieved with augmented models alone. Therefore, we conclude that using all three modifications simultaneously is not necessary. With appropriate settings, using augmented modification alone or in combination with one other modification is sufficient.

Overall, the experimental results show that our modifications inspired by the representation learning process are helpful in enhancing performance even with rough parameter selections. This further validates the potential of our approach of thinking about and improving the attention mechanism from a representation learning perspective. However, due to time and computational resource limitations, our experiments are conducted on a limited set of datasets and model size. More detailed parameter searches, validation across additional tasks and models, and the development of task-specific augmentation and negative sampling methods are all interesting directions worth exploring in the future.

---

### Decision · Program_Chairs · 2024-09-25

**Decision:**

Accept (poster)

**Comment:**

The paper seeks to understand in-context learning from a gradient descent perspective on a dual model. In addition to theoretical analysis, the authors also conduct empirical studies to verify the theoretical results.

While some reviewers raised concerns regarding to the assumptions made for the analysis and the generalization ability, the authors' response  generally resolved most of the concerns.  Considering the contributions to understanding the mechanism of LLMs, I prefer an acceptance.